# Limiting fluctuation and trajectorial stability of multilayer neural networks with mean field training

**Huy Tuan Pham**
Department of Mathematics, Stanford University

**Phan-Minh Nguyen** *
The Voleon Group

## Abstract

The mean field theory of multilayer neural networks centers around a particular infinite-width scaling, in which the learning dynamics is shown to be closely tracked by the mean field limit. A random fluctuation around this infinite-width limit is expected from a large-width expansion to the next order. This fluctuation has been studied only in the case of shallow networks, where previous works employ heavily technical notions or additional formulation ideas amenable only to that case. Treatment of the multilayer case has been missing, with the chief difficulty in finding a formulation that must capture the stochastic dependency across not only time but also depth.

In this work, we initiate the study of the fluctuation in the case of multilayer networks, at any network depth. Leveraging on the neuronal embedding framework recently introduced by Nguyen and Pham [17], we systematically derive a system of dynamical equations, called the *second-order mean field limit*, that captures the limiting fluctuation distribution. We demonstrate through the framework the complex interaction among neurons in this second-order mean field limit, the stochasticity with cross-layer dependency and the nonlinear time evolution inherent in the limiting fluctuation. A limit theorem is proven to relate quantitatively this limit to the fluctuation realized by large-width networks.

We apply the result to show a stability property of gradient descent mean field training: in the large-width regime, along the training trajectory, it progressively biases towards a solution with "minimal fluctuation" (in fact, vanishing fluctuation) in the learned output function, even after the network has been initialized at or has converged (sufficiently fast) to a global optimum. This extends a similar phenomenon previously shown only for shallow networks with a squared loss in the empirical risk minimization setting, to multilayer networks with a loss function that is not necessarily convex in a more general setting.

## 1 Introduction

Recent literature has witnessed much interest and progresses in the mean field theory of neural networks. In particular, it is shown that under a suitable scaling, as the widths tend to infinity, the neural network's learning dynamics converges to a nonlinear deterministic limit, known as the mean field (MF) limit [14, 17]. This line of works starts with analyses of the shallow case under various settings and has led to a number of nontrivial exciting results [18, 14, 5, 23, 25, 9, 22, 19, 29, 24, 30, 12, 1, 16]. The generalization to multilayer neural networks, already much more conceptually and technically challenging, has also been met with serious efforts from different groups of authors, with various novel ideas and insights [15, 17, 20, 2, 26, 6].

Since the MF limit is basically a first-order infinite-width approximation of the neural network, it is natural to consider the next order term in the expansion for a more faithful finite-width description.

---

*The author ordering is randomized.

35th Conference on Neural Information Processing Systems (NeurIPS 2021).

This leads to the consideration of the fluctuation, up-scaled appropriately with the widths, around the MF limit. On one hand, this fluctuation should display MF interactions among neurons. On the other hand, it is random due to the inherent stochasticity in the finite-width network which, for instance, is randomly initialized and hence induces randomness in the interactions among neurons.

For shallow networks, [27, 23, 3] has identified the limiting fluctuation in the form of a time-evolving signed measure over weights, leading to a central limit theorem (CLT) in the space of measures. In particular, [27] pinpoints the fluctuation via a compactness argument in an appropriate measure space. Avoiding the heavy technicality, [3] realizes a neat trick, specific to shallow networks, where quantities of interest are described via push-forward of the initialized values of the weights.

There has been no similar attempt for the multilayer case, which faces major obstacles in formulating the limiting fluctuation. Firstly, existing formulations of the MF limit already move away from working with measures over weights [15, 17, 26, 6], unless restricted conditions are assumed [2]. This is due to the complexity of MF interactions: the presence of middle layers brings about simultaneous actions of multiple symmetry groups. We face the same complexity when formulating the fluctuation. Secondly, unlike shallow networks, the fluctuation in multilayer networks displays stochastic dependency not only through time but also across layers. Specifically the cross-layer dependency is propagated by both forward and backward propagation signals, at any time instance.

**Contributions.** We tackle the challenge by leveraging on the neuronal embedding framework, recently proposed by [17]. An important concept that is supported by the neuronal embedding is the sampling of neurons. We use this concept to formulate the limiting fluctuation via a decomposition into two components with different roles in Section 3. One component is a random process, which encodes CLT-like stochasticity in the fluctuation of the sampled neurons around the MF ensemble. The other component, named the *second-order mean field limit*, is a system of ordinary differential equations (ODEs), which displays MF interactions in the deviation of the neural network under gradient descent (GD) training from the sampled neurons.

This decomposition is an innovation of the work. Our formulation enjoys the generality brought about by the neuronal embedding framework without restrictive assumptions (e.g. i.i.d. initialization assumption made in [2, 26]) and faithfully describes the expected stochastic dependency across time and depth. We prove a limit theorem that establishes the connection with the fluctuation in finite-sized networks **(Theorems 5 and 6)**. Unlike [27, 3], this result is quantitative.

Using this formulation, in Section 4, we show a trajectorial stability property in a large-width multilayer network, particularly a variance reduction effect: GD training traverses a solution trajectory that reduces and eventually eliminates the (width-corrected) variance of the learned output function. That is, a bias towards "minimal fluctuation". This holds even if the network is initialized at, or if it is trained to converge sufficiently fast to, a global optimum **(Theorems 9 and 10)**. The same phenomenon has been shown in the shallow case [3], which requires a squared loss in the empirical risk minimization setup. As demonstrated by our result, it is not an isolated phenomenon and can indeed hold for a neural architecture where MF interactions are more complex, a loss function that is not necessarily convex and a training setup unrestricted to finitely many training data points.

**Limitations and potential extensions.** Let us finally mention a few limitations. Our work considers fully-connected multilayer networks trained with GD, a setup much less general than [17]. We expect certain extensions in this direction are doable. We also do not treat stochastic GD and hence disregard the stochasticity of data sampling, unlike [27]. Given the broader literature on this subject (e.g. [11]), this extension is foreseeable. Here we focus instead on the stochasticity that is inherent in the interactions among neurons, which is the more interesting aspect of neural networks.

Our result on the output variance is specific to unregularized training, unlike [3], and also requires a sufficient global convergence rate. While there have been several proven global convergence guarantees for multilayer networks [17, 21, 20], understanding of the convergence rate is still lacking. Even in the shallow case, global convergence has been studied only for a type of sparsity-inducing regularization [5, 4]. Unless the convergence rate for multilayer networks is generally perilous (an unlikely scenario in light of the experiments in [15]), our result is expected to be relevant.

Our development is specific to the MF scaling. This scaling allows for nonlinear feature learning, unlike the NTK scaling [8]. While there are other scalings that also admit a certain sense of feature learning [7, 31], the standard parameterization in practice – in the infinite-width limit – is known to degenerate into NTK-like behaviors, which are not expected of practical finite-but-large-width

neural networks [13, 31]. In other words, *all* infinite-width scalings that display feature learning are only proxies of practical networks. This limitation motivates finite-width studies as we pursue here.

**Notations.** We use $K$ to denote a generic constant that may change from line to line, and similarly $K_u$ for a finite constant that may depend on some constant $u$. For simplicity, for $L$ the network depth, we write $K$ in place of $K_L$. For $\mathbb{E}$ (resp. $\mathbf{E}$) being the expectation, we use $\mathbb{V}$ and $\mathbb{C}$ (resp. $\mathbf{V}$ and $\mathbf{C}$) for the variance and covariance. We write $\mathbb{E}_Z$ for the expectation w.r.t. data $Z = (X, Y) \sim \mathcal{P}$. We write $\partial_i f$ to denote the partial derivative w.r.t. the $i$-th variable of $f$.

## 2 Background on the MF limit for multilayer networks and assumptions

We describe the necessary backgrounds based on (and simplifying) the work [17]. This section also introduces several important notations, as well as the assumptions for our study of the fluctuation.

**Multilayer network.** Following [17], we consider a $L$-layer fully-connected neural network with widths $\{N_i\}_{i \leq L}$ (for $N_L = 1$):

$$
\begin{aligned}
\hat{\mathbf{y}}(t, x) &= \varphi_L(\mathbf{H}_L(t, 1, x)), \\
\mathbf{H}_i(t, j_i, x) &= \mathbb{E}_J[\mathbf{w}_i(t, J_{i-1}, j_i) \varphi_{i-1}(\mathbf{H}_{i-1}(t, J_{i-1}, x))], \quad j_i \in [N_i], \ i = L, ..., 2, \\
\mathbf{H}_1(t, j_1, x) &= \langle \mathbf{w}_1(t, 1, j_1), x \rangle, \quad j_1 \in [N_1],
\end{aligned}
\tag{1}
$$

in which $x \in \mathbb{X} = \mathbb{R}^d$ is the input, the weights are $\mathbf{w}_1(t, 1, j_1) \in \mathbb{W}_1 = \mathbb{R}^d$ and $\mathbf{w}_i(t, j_{i-1}, j_i) \in \mathbb{W}_i = \mathbb{R}$, $\varphi_i : \mathbb{R} \to \mathbb{R}$ is the activation and $t \in \mathbb{T} = \mathbb{R}_{\geq 0}$ the continuous time. Here we reserve the notation $J_i$ for the random variable $J_i \sim \text{Unif}([N_i])$ and we write $\mathbb{E}_J$ to denote the expectation w.r.t. these random variables. For convenience, we take $j_0 \in \{1\}$ and $N_0 = 1$.

Given an initialization $\mathbf{w}_i(0, \cdot, \cdot)$, we train the network with gradient descent (GD) w.r.t. the loss $\mathcal{L} : \mathbb{R} \times \mathbb{R} \to \mathbb{R}_{\geq 0}$ and the data $Z = (X, Y) \in \mathbb{X} \times \mathbb{R}$ drawn from a training distribution $\mathcal{P}$:

$$
\partial_t \mathbf{w}_i(t, j_{i-1}, j_i) = -\mathbb{E}_Z\left[\partial_2 \mathcal{L}(Y, \hat{\mathbf{y}}(t, X)) \frac{\partial \hat{\mathbf{y}}(t, X)}{\partial \mathbf{w}_i(j_{i-1}, j_i)}\right], \quad i \in [L],
$$

in which we define

$$
\frac{\partial \hat{\mathbf{y}}(t, x)}{\partial \mathbf{H}_i(j_i)} = \begin{cases} \varphi_L'(\mathbf{H}_L(t, 1, x)), & i = L, \\ \mathbb{E}_J\left[\dfrac{\partial \hat{\mathbf{y}}(t, x)}{\partial \mathbf{H}_{i+1}(J_{i+1})} \mathbf{w}_{i+1}(t, j_i, J_{i+1}) \varphi_i'(\mathbf{H}_i(t, j_i, x))\right], & i < L, \end{cases}
$$

$$
\frac{\partial \hat{\mathbf{y}}(t, x)}{\partial \mathbf{w}_i(j_{i-1}, j_i)} = \begin{cases} \dfrac{\partial \hat{\mathbf{y}}(t, x)}{\partial \mathbf{H}_i(j_i)} \varphi_{i-1}(\mathbf{H}_{i-1}(t, j_{i-1}, x)), & i > 1, \\ \dfrac{\partial \hat{\mathbf{y}}(t, x)}{\partial \mathbf{H}_1(j_1)} x, & i = 1. \end{cases}
$$

One may recognize that these "derivative" quantities are defined in a perturbative fashion; for instance, $\frac{\partial \hat{\mathbf{y}}(t,x)}{\partial \mathbf{H}_i(j_i)}$ represents how $\hat{\mathbf{y}}(t, x)$ changes (rescaled by widths) as one perturbs $\mathbf{H}_i(t, j_i, x)$.

**Mean field limit.** The MF limit is defined upon a given neuronal ensemble $(\Omega, \mathcal{F}, P) = \prod_{i=0}^{L}(\Omega_i, \mathcal{F}_i, P_i)$ (in which $\Omega_0 = \Omega_L = \{1\}$), which is a product probability space. We reserve the notation $C_i$ for the random variable $C_i \sim P_i$ and we write $\mathbb{E}_C$ to denote the expectation w.r.t. these random variables. The MF limit that is associated with the network (1) is described by the evolution of $\{w_i(t, \cdot, \cdot)\}_{i \in [L]}$, given by the following MF ODEs:

$$
\partial_t w_i(t, c_{i-1}, c_i) = -\mathbb{E}_Z\left[\partial_2 \mathcal{L}(Y, \hat{y}(t, X)) \frac{\partial \hat{y}(t, X)}{\partial w_i(c_{i-1}, c_i)}\right], \quad i \in [L], \ c_i \in \Omega_i, \ c_{i-1} \in \Omega_{i-1},
$$

where $w_i : \mathbb{T} \times \Omega_{i-1} \times \Omega_i \to \mathbb{W}_i$ and $t \in \mathbb{T}$. Here we define the forward quantities:

$$
\begin{aligned}
\hat{y}(t, x) &= \varphi_L(H_L(t, 1, x)), \\
H_i(t, c_i, x) &= \mathbb{E}_C[w_i(t, C_{i-1}, c_i) \varphi_{i-1}(H_{i-1}(t, C_{i-1}, x))], \quad i = L, ..., 2, \\
H_1(t, c_1, x) &= \langle w_1(t, c_1), x \rangle,
\end{aligned}
$$

and the backward quantities:

$$\frac{\partial \hat{y}(t,x)}{\partial H_i(c_i)} = \begin{cases} \varphi'_L\left(H_L\left(t,1,x\right)\right), & i = L, \\ \mathbb{E}_C\left[\dfrac{\partial \hat{y}(t,x)}{\partial H_{i+1}(C_{i+1})} w_{i+1}\left(t,c_i,C_{i+1}\right)\varphi'_i\left(H_i\left(t,c_i,x\right)\right)\right], & i < L, \end{cases}$$

$$\frac{\partial \hat{y}(t,x)}{\partial w_i(c_{i-1},c_i)} = \begin{cases} \dfrac{\partial \hat{y}(t,x)}{\partial H_i(c_i)}\varphi_{i-1}\left(H_{i-1}\left(t,c_{i-1},x\right)\right), & i > 1, \\ \dfrac{\partial \hat{y}(t,x)}{\partial H_1(c_1)}x, & i = 1. \end{cases}$$

The evolution with time $t$ of the MF limit is generally complex due to the nonlinear activations.

**Sampling of neurons, neuronal embedding, and the neural net-MF limit connection.** A priori there is no connection between the MF limit and the neural network (1); they are two separate self-contained time-evolving systems, despite the similarity in their definitions. To formalize their connection as in [17], we describe the neuron sampling procedure. In particular, we independently sample $\{C_i(j_i)\}_{j_i \in [N_i]} \sim_{\text{i.i.d.}} P_i$ for $i = 1, ..., L$. The samples $C_i(j_i)$ should be thought of as "sampled neurons". Let us use $\mathbf{E}$ to denote the expectation w.r.t. the sampling randomness. Although [17] considers more general sampling rule, this suffices for our purposes.

Now suppose that we are given functions $w_i^0 : \Omega_{i-1} \times \Omega_i \to \mathbb{W}_i$ and set the initializations for the neural network and the MF limit:

$$\mathbf{w}_i\left(0,j_{i-1},j_i\right) = w_i^0\left(C_{i-1}\left(j_{i-1}\right),C_i\left(j_i\right)\right), \quad w_i\left(0,\cdot,\cdot\right) = w_i^0\left(\cdot,\cdot\right) \quad \forall i \in [L].$$

Then one lets them evolve according to their own respective dynamics. The sampling of neurons hence allows to couple them on the basis of the tuple $\left(\Omega, \mathcal{F}, P, \{w_i^0\}_{i\in[L]}\right)$ – known as *the neuronal embedding*, the key idea in [17]. Note that this does not pose a major limitation on the type of initializations for the neural network (1); indeed it allows both i.i.d. and non-i.i.d. initializations since we have freedom to choose the neuronal embedding, as studied in [17, 20].

Once the coupling is done, [17] obtains the following result, which realizes the connection in mathematical terms. For any finite terminal time $T \in \mathbb{T}$, we have [2]:

$$\max_{i\in[L]}\mathbb{E}_J\left[\sup_{t\leq T}\left|\mathbf{w}_i\left(t,J_{i-1},J_i\right) - w_i\left(t,C_{i-1}\left(J_{i-1}\right),C_i\left(J_i\right)\right)\right|^2\right]^{1/2} = \tilde{O}(N_{\min}^{-c_*}) \qquad (2)$$

with high probability, where $c_* > 0$ is a universal constant, $N_{\min} = \min_{i\in[L-1]} N_i$, $N_{\max} = \max_{i\in[L]} N_i$ and $\tilde{O}$ hides the dependency on $T$, $L$ and $\log N_{\max}$. This result is akin to the law of large numbers (LLN). In words, it says that for large (ideally infinite) widths, the evolution of the network (1) can be tracked closely by the MF limit. This result is fundamental and gives a useful suggestion that is to study the neural network via analyzing the MF limit. For example, [17] uses it to study the optimization efficiency of the neural network.

The sampling of neurons is inspired by the propagation of chaos argument [28], but is not a mere proof device. Indeed, in this work, we again make a crucial use of this sampling in the formulation of the limiting fluctuation (Section 3.3).

**Set of assumptions.** We consider the following assumptions for the rest of the paper:

**Assumption 1.** *We make the following assumptions:*

- *(Regularity) $\varphi_i$ is $K$-bounded for $i \in [L-1]$; $\varphi'_i$ and $\varphi''_i$ are $K$-bounded and $K$-Lipschitz for $i \in [L]$; $\partial_2 \mathcal{L}$ and $\partial_2^2 \mathcal{L}$ are $K$-bounded and $K$-Lipschitz in the second variable; $|X| \leq K$ almost surely (a.s.).*
- *(Sub-Gaussian initialization) $\sup_{m\geq 1} m^{-1/2}\mathbb{E}_C\left[\left|w_i^0\left(C_{i-1},C_i\right)\right|^m\right]^{1/m} \leq K$ for any $i \in [L]$.*
- *(Measurability) $L^2(P_i)$ is separable for any $i \in [L]$.*
- *(Constant hidden widths) $N_1 = ... = N_{L-1} = N$ (with ideally $N \to \infty$).*

An example for the regularity assumption is $\varphi_L(u) = u$ the identity, $\varphi_i = \tanh$ for $i \in [L-1]$ and $\mathcal{L}(u_1, u_2)$ is a smooth version of the Huber loss. It is noteworthy that $\mathcal{L}$ does not need to be

---

[2]While [17] treats stochastic GD in discrete time, this result is implicit in the proof.

convex. The initialization assumption is also mild; it allows most common i.i.d. initializations as well as non-i.i.d. schemes in [17, 20]. These assumptions satisfy the conditions in these works. Measurability assumption is a technical condition needed for well-defined-ness of the fluctuation; it is not conceptually restrictive and can accommodate results in [17, 20].

The constant width assumption aligns with our interest in the infinite-width regime. Our development can be extended easily to the proportional scaling, where $N_i = \lfloor \rho_i N \rfloor$ for some constants $\rho_1, ..., \rho_{L-1} > 0$ with $N \to \infty$. These scalings are relevant to practical setups and also to yield interesting results. Specifically once some widths are much larger than others, we expect the fluctuation to be dominated by a subset of layers.

## 3 Limit system for the fluctuation around the MF limit

We describe our formulation for the fluctuation. As introduced, it is composed of two components: a stochastic process that induces Gaussian CLT-like behavior, and another process – called the second-order MF limit – that signifies the MF interactions at the fluctuation level. This is done on the basis of the sampling of neurons, on top of the neuronal embedding, introduced in Section 2.

We introduce the first component in Section 3.1 and then the second-oder MF limit in 3.2. Together the two components are connected with the fluctuation around the MF limit realized by a large-width network in Section 3.3, to give firstly the analog of (2) and secondly a limit theorem on the fluctuation distribution. We end with a discussion in Section 3.4. Proofs are deferred to the appendix.

### 3.1 Gaussian component $\tilde{G}$

Recall the sampling of neurons described in Section 2. Given the MF limit, we define the following random quantities:

$$\tilde{y}(t, x) = \varphi_L(\tilde{H}_L(t, 1, x)),$$

$$\tilde{H}_i(t, c_i, x) = \mathbb{E}_J \left[ w_i(t, C_{i-1}(J_{i-1}), c_i) \varphi_{i-1}(\tilde{H}_{i-1}(t, C_{i-1}(J_{i-1}), x)) \right], \qquad i = L, ..., 2,$$

$$\tilde{H}_1(t, c_1, x) = \langle w_1(t, c_1), x \rangle,$$

$$\frac{\partial \tilde{y}(t, x)}{\partial \tilde{H}_i(c_i)} = \begin{cases} \varphi_L'(\tilde{H}_L(t, 1, x)), & i = L, \\ \mathbb{E}_J \left[ \dfrac{\partial \tilde{y}(t, x)}{\partial \tilde{H}_{i+1}(C_{i+1}(J_{i+1}))} w_{i+1}(t, c_i, C_{i+1}(J_{i+1})) \varphi_i'(\tilde{H}_i(t, c_i, x)) \right], & i < L, \end{cases}$$

$$\frac{\partial \tilde{y}(t, x)}{\partial w_i(c_{i-1}, c_i)} = \begin{cases} \dfrac{\partial \tilde{y}(t, x)}{\partial \tilde{H}_i(c_i)} \varphi_{i-1}(\tilde{H}_{i-1}(t, c_{i-1}, x)), & i > 1, \\ \dfrac{\partial \tilde{y}(t, x)}{\partial \tilde{H}_1(c_1)} x, & i = 1. \end{cases}$$

These quantities are analogues of the forward and backward MF quantities, but with $\mathbb{E}_{C_i}$ being replaced by an empirical average over $\{C_i(j_i)\}_{j_i \in [N_i]}$.

Now we define

$$\tilde{G}^y(t, x) = \sqrt{N}(\tilde{y}(t, x) - \hat{y}(t, x)),$$

$$\tilde{G}_i^w(t, c_{i-1}, c_i, x) = \sqrt{N} \left( \frac{\partial \tilde{y}(t, x)}{\partial w_i(c_{i-1}, c_i)} - \frac{\partial \hat{y}(t, x)}{\partial w_i(c_{i-1}, c_i)} \right), \quad i = 1, ..., L,$$

and let $\tilde{G}$ denote the collection of these functions. Notice that the randomness of $\tilde{G}$ is induced by the samples $C_i(j_i)$; thus we write the expectation w.r.t. $\tilde{G}$ via $\mathbf{E}$. Intuitively $\tilde{G}$ is the fluctuation of the sampled neurons around the MF limit and thus should converge to a Gaussian process. We show that this is indeed the case in the following mode of convergence.

**Definition 1.** Given joint distributions over a sequence (in $N$) of functions $\tilde{G}_{i,N} : \mathbb{T} \times \Omega_{i-1} \times \Omega_i \times \mathbb{X} \to \mathbb{W}_i$ for $i \in [L]$, we say that $\tilde{G}_N = \{\tilde{G}_{i,N}\}_{i \in [L]}$ *converges $G$-polynomially in moment* (as $N \to \infty$) to $\underline{G}$ if for any finite collection in $\ell$ of square-integrable $f_\ell : \mathbb{T} \times \Omega_{i(\ell)-1} \times \Omega_{i(\ell)} \times \mathbb{X} \to \mathbb{W}_{i(\ell)}$ that is continuous in time and integers $\alpha_\ell, \beta_\ell \geq 1$,

$$\sup_{t(\ell) \leq T \, \forall \ell} \left| \mathbb{E} \left[ \prod_\ell \langle f_\ell, \tilde{G}_{i(\ell),N}^{\alpha_\ell} \rangle_{t(\ell)}^{\beta_\ell} \right] - \mathbb{E} \left[ \prod_\ell \langle f_\ell, \underline{G}_{i(\ell)}^{\alpha_\ell} \rangle_{t(\ell)}^{\beta_\ell} \right] \right| = O_{D,T} \left( \sup_{t \leq T} \max_\ell \|f_\ell\|_t^D \right) N^{-1/8},$$

for $D = \sum_\ell \alpha_\ell \beta_\ell$ and a constant terminal time $T$. Here we define:

$$\langle f, h \rangle_t = \mathbb{E}_{Z,C} \left[ f \left( t, C_{i-1}, C_i, X \right) h \left( t, C_{i-1}, C_i, X \right) \right]$$

for any two such functions $f$ and $h$, and $\|f\|_t^2 = \langle f, f \rangle_t$. As a convenient convention, if $f :$ $\mathbb{T} \times \Omega_0 \times \Omega_1 \times \mathbb{X} \to \mathbb{W}_1$ is tested against $\tilde{G}_{1,N}$, we define $\langle f, \tilde{G}_{1,N} \rangle_t$ and $\|f\|_t$ using Euclidean inner product and the Euclidean norm.

Note that this mode of convergence gives a quantitative rate in $N$.

**Theorem 2** (Gaussian component $\tilde{G}$). *For any terminal time $T \geq 0$, $\tilde{G}$ converges $G$-polynomially in moment to $\underline{G} = \{\underline{G}^y, \ \underline{G}_i^w, \ i \in [L]\}$, which is a collection of centered Gaussian processes.*

The covariance structure of $\underline{G}$ in Theorem 2 is explicit but lengthy; we refer to the appendix for full details. Here let us examine the variance of $\underline{G}^y$, given recursively by:

$$v_2 \left( t, c_2, x \right) = \mathbb{V}_C \left[ w_2 \left( t, C_1, c_2 \right) \varphi_1 \left( H_1 \left( t, C_1, x \right) \right) \right],$$
$$v_i \left( t, c_i, x \right) = \mathbb{E}_C \left[ \left| w_i \left( t, C_{i-1}, c_i \right) \varphi'_{i-1} \left( H_{i-1} \left( t, C_{i-1}, x \right) \right) \right|^2 v_{i-1} \left( t, C_{i-1}, x \right) \right]$$
$$+ \mathbb{V}_C \left[ w_i \left( t, C_{i-1}, c_i \right) \varphi_{i-1} \left( H_{i-1} \left( t, C_{i-1}, x \right) \right) \right], \quad i = 3, ..., L,$$
$$\mathbf{E} \left[ \left| \underline{G}^y \left( t, x \right) \right|^2 \right] = \left| \varphi'_L \left( H_L \left( t, 1, x \right) \right) \right|^2 v_L \left( t, 1, x \right).$$

The recursive structure shows that $\underline{G}^y$ compounds the stochasticity propagated forwardly through the depth of the multilayer network. From the appendix, a similar observation can be made in the backward propagation direction.

## 3.2 Second-order MF limit

We introduce a system of ODE's for a dynamics that represents MF interactions at the fluctuation level. As we shall see later, this system gives a limiting description of the the deviation of the neural network from the sampled neurons under GD training.

Let us denote by $G$ a collection of functions $G^y : \mathbb{T} \times \mathbb{X} \to \mathbb{R}$, $G_i^w : \mathbb{T} \times \Omega_{i-1} \times \Omega_i \times \mathbb{X} \to \mathbb{W}_i$ for $i \in [L]$. For $p \geq 1$, let

$$\mathcal{G} = \left\{ G : \ \|G\|_T < \infty \ \forall T \geq 0 \right\}, \quad \|G\|_{T,2p}^{2p} = \sup_{t \leq T} \mathbb{E}_{Z,C} \left[ \left| G^y \left( t, X \right) \right|^{2p} + \sum_{i=1}^{L} \left| G_i^w (t, C_{i-1}, C_i, X) \right|^{2p} \right],$$

with $\|G\|_T \equiv \|G\|_{T,2}$. We define processes $R_i : \mathcal{G} \times \mathbb{T} \times \Omega_{i-1} \times \Omega_i \to \mathbb{W}_i$ for $i \in [L]$ as follows. The processes $R_i$ are initialized at $R_i \left( \cdot, 0, \cdot, \cdot \right) = 0$ for all $i$, and for each fixed $G \in \mathcal{G}$, solve the differential equations:

$$\partial_t R_i \left( G, t, c_{i-1}, c_i \right) =$$
$$- \mathbb{E}_Z \left[ \frac{\partial \hat{y} \left( t, X \right)}{\partial w_i \left( c_{i-1}, c_i \right)} \partial_2^2 \mathcal{L} \left( Y, \hat{y} \left( t, X \right) \right) \left( G^y \left( t, X \right) + \sum_{r=1}^{L} \mathbb{E}_C \left[ R_r \left( G, t, C_{r-1}, C_r \right) \frac{\partial \hat{y} \left( t, X \right)}{\partial w_r \left( C_{r-1}, C_r \right)} \right] \right) \right]$$
$$- \mathbb{E}_Z \left[ \partial_2 \mathcal{L} \left( Y, \hat{y} \left( t, X \right) \right) \mathbb{E}_C \left[ \sum_{r=1}^{L} \left[ R_r \left( G, t, a, b \right) \frac{\partial^2 \hat{y} \left( t, X \right)}{\partial w_r \left( a, b \right) \partial w_i \left( c_{i-1}, c_i \right)} \right]_{a := C_{r-1}, \ b := C_r} \right] \right]$$
$$- \mathbb{E}_Z \left[ \partial_2 \mathcal{L} \left( Y, \hat{y} \left( t, X \right) \right) G_i^w \left( t, c_{i-1}, c_i, X \right) \right]. \tag{3}$$

Here $\frac{\partial^2 \hat{y}(t,X)}{\partial w_r(a,b) \partial w_i(c_{i-1}, c_i)}$ is defined in a perturbative fashion similar to those in Section 2 and is hence self-explanatory; we give the full explicit definitions in the appendix. We shall write $R_i \left( G, t \right) = R_i \left( G, t, \cdot, \cdot \right)$ for brevity. Let $R$ denote the collection $\{R_i\}_{i \in [L]}$.

**Theorem 3.** *Under Assumption 1, for any $\epsilon > 0$ and $G \in \mathcal{G}$ with $\|G\|_{T,2+\epsilon} < \infty$, there exists a unique solution $t \mapsto R_i \left( G, t, \cdot, \cdot \right) \in L^2 \left( P_{i-1} \times P_i \right)$ which is continuous in time. Furthermore, for each $t$, $R \left( G, t \right)$ is a continuous linear functional in $G$.*

The process $R$ is called the *second-order MF limit* for a reason we shall see in Section 3.4.

*Remark* 4. Care should be taken in the derivation of the "second derivative" quantity in Eq. (3). To illustrate, consider a simplified problem of taking a perturbation w.r.t. $w_r(a, b)$ of the quantity

$$g\left(w_i\left(c_{i-1}, c_i\right), \mathbb{E}_C\left[f_1\left(w_i\left(C_{i-1}, c_i\right)\right)\right], \mathbb{E}_C\left[f_2\left(w_i\left(c_{i-1}, C_i\right)\right)\right], \mathbb{E}_C\left[f_3\left(w_i\left(C_{i-1}, C_i\right)\right)\right]\right) \equiv g\left(c_{i-1}, c_i\right).$$

For instance, when $r = i$, we have:

$$\left[R_i\left(G, t, a, b\right) \frac{\partial g\left(c_{i-1}, c_i\right)}{\partial w_r\left(a, b\right)}\right]_{a:=C_{i-1},\, b:=C_i}$$
$$= R_i\left(G, t, c_{i-1}, c_i\right) \partial_1 g\left(c_{i-1}, c_i\right) + R_i\left(G, t, C_{i-1}, c_i\right) \partial_2 g\left(c_{i-1}, c_i\right) \partial f_1\left(w_i\left(C_{i-1}, c_i\right)\right)$$
$$+ R_i\left(G, t, c_{i-1}, C_i\right) \partial_3 g\left(c_{i-1}, c_i\right) \partial f_2\left(w_i\left(c_{i-1}, C_i\right)\right) + R_i\left(G, t, C_{i-1}, C_i\right) \partial_4 g\left(c_{i-1}, c_i\right) \partial f_3\left(w_i\left(C_{i-1}, C_i\right)\right).$$

### 3.3  Fluctuation around the MF limit via sampling of neurons

We are interested in a characterization of the following quantity, which represents the deviation of the weights of the neural network under GD training from the sampled neurons:

$$\mathbf{R}_i\left(t, j_{i-1}, j_i\right) = \sqrt{N}\left(\mathbf{w}_i\left(t, j_{i-1}, j_i\right) - w_i\left(t, C_{i-1}\left(j_{i-1}\right), C_i\left(j_i\right)\right)\right).$$

Let $\mathbf{R}$ denote the collection $\{\mathbf{R}_i\}_{i\in[L]}$. We now state our first main result of the section.

**Theorem 5** (Second-order MF). *Under Assumption 1, we have, for any $t \leq T$,*

$$\mathbf{E}\mathbb{E}_J\left[\left|\left|\mathbf{R}_i\left(t, J_{i-1}, J_i\right) - R_i(\tilde{G}, t, C_{i-1}\left(J_{i-1}\right), C_i\left(J_i\right))\right|\right|^2\right] \leq K_T/N.$$

Upon the correct choice $\tilde{G}$, one deduces from the theorem the second-order expansion:

$$\mathbf{w}_i\left(t, j_{i-1}, j_i\right) \approx w_i\left(t, C_{i-1}\left(j_{i-1}\right), C_i\left(j_i\right)\right) + N^{-1/2}R_i(\tilde{G}, t, C_{i-1}\left(j_{i-1}\right), C_i\left(j_i\right)),$$

where the first-order term is directly from Eq. (2). Importantly this relation is on the basis of the sampled neurons $\{C_i\left(j_i\right):\, j_i \in [N_i],\, i \in [L]\}$. The dynamics of $R_i$ at the $i$-th layer is dependent on all $\{R_k\}_{k\in[L]}$ and $G^y$, exemplifying the cross-layer interaction of fluctuations in multilayer networks.

Although Theorem 5 involves $\tilde{G}$, as shown in the derivation of Section 3.4, we actually do not utilize the specific structure of $\tilde{G}$. Incidentally we study the limiting structure of $\tilde{G}$ separately in Theorem 2. In other words, our decomposition of the fluctuation via $\tilde{G}$ and $R$ allows relatively independent treatments of the two components (which have different natures, to be discussed in Section 3.4).

Combining the two theorems, we obtain the following CLT for the limiting output fluctuation.

**Theorem 6** (CLT for output function). *Under Assumption 1, the fluctuation $\sqrt{N}(\hat{\mathbf{y}}(t, x) - \hat{y}(t, x))$ converges weakly to the Gaussian process $\hat{G}$ indexed by $\mathbb{T} \times \mathbb{X}$:*

$$\hat{G}\left(t, x\right) = \sum_{i=1}^{L} \mathbb{E}_C\left[R_i(\underline{G}, t, C_{i-1}, C_i)\frac{\partial \hat{y}(t, x)}{\partial w_i(C_{i-1}, C_i)}\right] + \underline{G}^y(t, x),$$

*where $\underline{G}$ is the Gaussian process described in Theorem 2. Specifically, for any integer $m \geq 1$, $t \leq T$, and 1-bounded $h : \mathbb{X} \to \mathbb{R}$,*

$$\left|\mathbf{E}\mathbb{E}_Z\left[h(X)\left(\sqrt{N}(\hat{\mathbf{y}}(t, X) - \hat{y}(t, X))\right)^m\right] - \mathbf{E}\mathbb{E}_Z\left[h(X)\left(\hat{G}\left(t, X\right)\right)^m\right]\right| \leq K_{T,m}N^{-1/8+o(1)}.$$

We note in passing the technicality we devise to prove Theorem 6. As noted, Theorems 5 and 2 separate treatments of $R$ and $\tilde{G}$, but recall we are interested in $R(\tilde{G}, t)$. This necessitates a way to describe them jointly. We do so via the following notion of convergence, extending Definition 1.

**Definition 7.** Suppose we are given a sequence (in $N$) of functions $\tilde{G}_{i,N} : \mathbb{T} \times \Omega_{i-1} \times \Omega_i \times \mathbb{X} \to \mathbb{W}_i$ for $i \in [L]$ such that $\tilde{G}_N = \{\tilde{G}_{i,N}\}_{i\in[L]}$ converges $G$-polynomially in moment to $G$. Given further $\tilde{R}_{i,N} : \mathbb{T} \times \Omega_{i-1} \times \Omega_i \times \mathbb{X} \to \mathbb{W}_i$ defined jointly with $\tilde{G}_{i,N}$ and $\tilde{R}_N = \{\tilde{R}_{i,N}\}_{i\in[L]}$, we say that $(\tilde{R}_N, \tilde{G}_N)$ *converges $G$-polynomially in moment and $R$-linearly in moment* (as $N \to \infty$) to $(R, G)$

if for any $f_\ell, h_\ell$ that are $\mathbb{T} \times \Omega_{i(\ell)-1} \times \Omega_{i(\ell)} \times \mathbb{X} \to \mathbb{W}_{i(\ell)}$ mappings and continuous in time and any integers $\alpha_\ell, \beta_\ell \geq 1$,

$$\sup_{t(\ell) \leq T \, \forall \ell} \left| \mathbb{E}\left[ \prod_\ell \langle f_\ell, \tilde{G}_{i(\ell),N}^{\alpha_\ell} \rangle_{t(\ell)}^{\beta_\ell} \prod_\ell \langle h_\ell, \tilde{R}_{i(\ell),n} \rangle_{t(\ell)}^{\beta_\ell} \right] - \mathbb{E}\left[ \prod_\ell \langle f_\ell, G_{i(\ell)}^{\alpha_\ell} \rangle_{t(\ell)}^{\beta_\ell} \prod_\ell \langle h_\ell, R_{i(\ell)} \rangle_{t(\ell)}^{\beta_\ell} \right] \right|$$

$$= O_D\left( \sup_{t \leq T} \max_\ell \|f_\ell\|_t^D, \ \sup_{t \leq T} \max_\ell \|h_\ell\|_t^D \right) N^{-1/8+o(1)},$$

for $D = \sum_\ell \beta_\ell \alpha_\ell + \sum_\ell \beta_\ell$.

**Proposition 8.** *Assume that $\tilde{G}$ converges $G$-polynomially in moment to $G = \{G^y, G_i^w, i \in [L]\}$. Then $(R(\tilde{G}, \cdot), \tilde{G})$ converges $G$-polynomially in moment and $R$-linearly in moment to $(R(G, \cdot), G)$.*

### 3.4 A heuristic derivation and discussion

**A heuristic argument.** We wish to give a heuristic to derive the relation between $\mathbf{R}_i$ and $R_i(\tilde{G}, t)$, i.e. Theorem 5. Suppose we look at the $i$-th layer:

$$\partial_t \mathbf{R}_i(t, j_{i-1}, j_i)$$
$$= \sqrt{N} \partial_t \mathbf{w}_i(t, j_{i-1}, j_i) - \sqrt{N} \partial_t w_i(t, C_{i-1}(j_{i-1}), C_i(j_i))$$
$$= \sqrt{N} \mathbb{E}_Z\left[ \partial_2 \mathcal{L}(Y, \hat{y}(t, X)) \frac{\partial \hat{y}(t, X)}{\partial w_i(C_{i-1}(j_{i-1}), C_i(j_i))} \right] - \sqrt{N} \mathbb{E}_Z\left[ \partial_2 \mathcal{L}(Y, \hat{\mathbf{y}}(t, X)) \frac{\partial \hat{\mathbf{y}}(t, X)}{\partial \mathbf{w}_i(j_{i-1}, j_i)} \right]$$
$$= \mathbb{E}_Z\left[ \underbrace{\sqrt{N}\left( \partial_2 \mathcal{L}(Y, \hat{y}(t, X)) - \partial_2 \mathcal{L}(Y, \hat{\mathbf{y}}(t, X)) \right)}_{A_1} \cdot \frac{\partial \hat{y}(t, X)}{\partial w_i(C_{i-1}(j_{i-1}), C_i(j_i))} \right]$$
$$+ \mathbb{E}_Z\left[ \partial_2 \mathcal{L}(Y, \hat{\mathbf{y}}(t, X)) \cdot \sqrt{N}\left( \frac{\partial \hat{y}(t, X)}{\partial w_i(C_{i-1}(j_{i-1}), C_i(j_i))} - \frac{\partial \hat{\mathbf{y}}(t, X)}{\partial \mathbf{w}_i(j_{i-1}, j_i)} \right) \right]. \tag{4}$$

Let us zoom into $A_1$, with some care not to eliminate the fluctuation of interest:

$$A_1 \approx \sqrt{N} \partial_2^2 \mathcal{L}(Y, \hat{y}(t, X)) (\hat{y}(t, X) - \hat{\mathbf{y}}(t, X))$$
$$= \partial_2^2 \mathcal{L}(Y, \hat{y}(t, X)) \left( -\tilde{G}^y(t, X) + \sqrt{N}(\tilde{y}(t, X) - \hat{\mathbf{y}}(t, X)) \right).$$

Now observe that $\tilde{y}(t, X) - \hat{\mathbf{y}}(t, X)$ is a difference between the sampled neurons and the neural network, and hence one should be able to express it in terms of $\mathbf{R}$:

$$\sqrt{N}(\tilde{y}(t, X) - \hat{\mathbf{y}}(t, X))$$
$$\approx \sum_{r=1}^L \mathbb{E}_J\left[ \frac{\partial \tilde{y}(t, X)}{\partial w_r(C_{r-1}(J_{r-1}), C_r(J_r))} \cdot \sqrt{N}(w_r(t, C_{r-1}(J_{r-1}), C_r(J_r)) - \mathbf{w}_r(t, J_{r-1}, J_r)) \right]$$
$$= -\sum_{r=1}^L \mathbb{E}_J\left[ \frac{\partial \tilde{y}(t, X)}{\partial w_r(C_{r-1}(J_{r-1}), C_r(J_r))} \cdot \mathbf{R}_r(t, J_{r-1}, J_r) \right].$$

Furthermore in the limit $N \to \infty$, we expect from the LLN, for a test function $f$,

$$\mathbb{E}_J[f(C_{r-1}(J_{r-1}), C_r(J_r))] \approx \mathbb{E}_C[f(C_{r-1}, C_r)]. \tag{5}$$

As such, one can identify the term that involves $A_1$ in Eq. (4) for $\partial_t \mathbf{R}_i(t, j_{i-1}, j_i)$ with the first term in Eq. (3) for $\partial_t R_i(G, t, c_{i-1}, c_i)$. One can derive similarly for the rest of the terms. With these, one arrives at $\partial_t \mathbf{R}_i(t, j_{i-1}, j_i) \approx \partial_t R_i(\tilde{G}, t, C_{i-1}(j_{i-1}), C_i(j_i))$. Recalling that $\mathbf{R}_i(0, \cdot, \cdot) = 0 = R_i(\cdot, 0, \cdot, \cdot)$, one arrives at the conclusion of Theorem 5.

We make two comments. Firstly, the sampling of neurons makes transparent the derivation of $R$ as the limit of $\mathbf{R}$: it allows to compare one-to-one a "neuron" of the former to a neuron of the latter. This demonstrates an advantage of the neuronal embedding framework, which easily accommodates the sampling of neurons in the multilayer setup. We also see that the only involvement of $\tilde{G}$ in this derivation is via its definition.

Secondly, $R$ arises (by replacing $\mathbf{R}$) at steps that invoke (5). This LLN-type nature signifies MF interactions among neurons at the fluctuation level, and hence the name second-order MF limit. In contrast, the component $\tilde{G}$ displays a Gaussian CLT-type nature, evident from its definition.

**Technical difficulties.** So far we have seen $R$ captures the difference between the neural network and the sampled neurons. Recall the other component in the fluctuation is the process $\tilde{G}$, which captures the $\sqrt{N}$-scaled difference between the sampled neurons and the MF limit. Taking into account the whole system of fluctuation, we note a major technical subtlety: the aforementioned two differences captured by $R$ and $\tilde{G}$ share the same source of randomness $\{C_i(j_i): j_i \in [N_i], i \in [L]\}$.

We mention two particular complications that arise from this subtlety. The first complication is how one should define the process $R$ in relation with $\tilde{G}$, given that $R$ is meant to be the infinite-width limit of $\mathbf{R}$ while $\mathbf{R}$ and $\tilde{G}$ are stochastically coupled. Our solution is to let $t \mapsto R(G, t)$ defined for any $G \in \mathcal{G}$, not restricted to only $\tilde{G}$. This streamlines the definition of the second-order MF limit and allows to separate our treatments of $R$ and $\tilde{G}$, as evident from Theorems 5 and 2.

The second complication lies with Eq. (5). In fact, to arrive at the desired conclusion, we require the function $f$ to depend on $\tilde{G}$. This is because the main object of interest is $R(\tilde{G}, t)$, even though we have treated $R$ and $\tilde{G}$ separately. In a nutshell, $f$ shares randomness with $C_r(j_r)$ and $C_{r-1}(j_{r-1})$, $j_r \in [N_r], j_{r-1} \in [N_{r-1}]$. The random variable on the left-hand side of Eq. (5) is therefore complex, and it is highly questionable whether this equation should hold. The analysis becomes delicate; without taking into account this shared randomness, the derivation would be a mere heuristic. As we present in the proof, we verify this equation for a relevant set of functions $f$.

## 4 Asymptotic variance of the output function

We study the following width-scaled asymptotic variance quantity:
$$V^*(t) = \lim_{N \to \infty} \mathbf{E}\mathbb{E}_Z\left[\left|\sqrt{N}(\hat{\mathbf{y}}(t, X) - \hat{y}(t, X))\right|^2\right] = \lim_{N \to \infty} \mathbf{E}\mathbb{E}_Z\left[\left|\sqrt{N}(\hat{\mathbf{y}}(t, X) - \mathbf{E}[\hat{\mathbf{y}}(t, X)])\right|^2\right].$$
The second equality holds and the limits in $N$ exist by Theorems 5 and 6. We would like to understand $V^*(t)$ in the long-time horizon, and specifically in a situation of considerable interest where the MF limit converges to a global optimum as $t \to \infty$. To that end, we assume the following.

**Assumption 2.** *We assume* $\mathbb{E}_Z\left[\partial_2^2 \mathcal{L}(Y, f(X)) \big| X\right] = K$ *a positive constant a.s. for any $f$ in which* $\mathbb{E}_Z\left[\partial_2 \mathcal{L}(Y, f(X)) | X\right] = 0$ *a.s.*

Note that convexity of the loss is not required. For example, the assumption holds for $\mathcal{L}(y, y') = \ell(y - y')$ for any quasi-convex smooth function $\ell$ when $Y = y(X)$ is a deterministic function of $X$. It is driven by our interest in what happens at global convergence: [17, 20] show convergence to a global optimum is attainable (with suitable initialization strategies) if either (case 1) $\mathcal{L}$ is convex in the second variable, or (case 2) $\partial_2 \mathcal{L}(y, y') = 0$ implies $\mathcal{L}(y, y') = 0$ (recalling $\mathcal{L} \geq 0$) and $Y = y(X)$. In both cases, one can find reasonable loss functions that satisfy Assumption 2. We also note in both cases, any $f$ with $\mathbb{E}_Z\left[\partial_2 \mathcal{L}(Y, f(X)) | X\right] = 0$ a.s. is a global optimizer.

Our first result indicates that even if one initializes the network at a global optimum and hence there is no evolution with time at the MF limit level, GD training still helps by reducing the output variance $V^*(t)$ with time at the fluctuation level, in the large-width regime.

**Theorem 9.** *Suppose at initialization* $\mathbb{E}_Z\left[\partial_2 \mathcal{L}(Y, \hat{y}(0, X)) | X\right] = 0$ *a.s. Under Assumptions 1 and 2, $V^*(t)$ is non-increasing and $V^*(t) \to 0$ as $t \to \infty$.*

This variance reduction effect continues to hold in the long-time horizon if, instead of initializing at a global optimum, we assume to have global convergence at a sufficiently fast rate.

**Theorem 10.** *Assume* $\int_0^\infty t^{2+\delta} \mathbb{E}_X\left[\left|\mathbb{E}_Z[\partial_2 \mathcal{L}(Y, \hat{y}(t, X)) | X]\right|^2\right]^{1/2} dt < \infty$ *for some $\delta > 0$. Under Assumptions 1 and 2, $V^*(t) \to 0$ as $t \to \infty$.*

These results suggest that the variance reduction effect of GD in MF training is a phenomenon more general than the case of shallow networks with a squared loss and finitely many training points in [3]. Specifically it is shown for multilayer networks with a loss that is not necessarily convex with arbitrary training data distribution $\mathcal{P}$. Though we discuss in the context of a global optimum as our main interest, the theorems apply to any stationary point where Assumption 2 holds.

Let us briefly discuss the proof. For simplicity, consider the context of Theorem 9. Recall from Theorem 6 the Gaussian process $\hat{G}$ that the output fluctuation converges to as $N \to \infty$; we thus

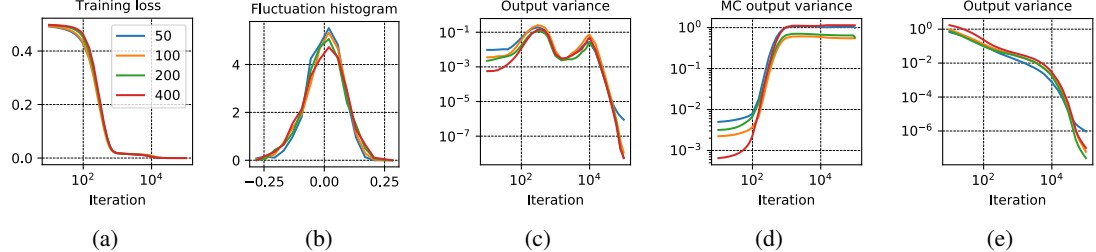

Figure 1: MNIST classification of digits 0 and 4 versus 5 and 9, with full-batch GD training on 100 images. The network has 3 layers, $\tanh$ activations and a Huber loss. We vary the width $N \in \{50, 100, 200, 400\}$. See Appendix G for details. (a): The evolution of the training loss, at a random initialization. (b): Histogram of $\sqrt{N}(\hat{\mathbf{y}}(t, x) - \hat{y}(t, x))$ at iteration $10^3$ for a test image $x$. (c): Output variance $\mathbf{E}\mathbb{E}_Z\left[\left|\sqrt{N}(\hat{\mathbf{y}}(t, X) - \hat{y}(t, X))\right|^2\right]$. (d): Monte-Carlo (MC) output variance, a.k.a. $\mathbf{E}\mathbb{E}_Z[|\tilde{G}^y(t, X)|^2]$. (e): Similar to (c), but the network is initialized at a global optimum.

would like to show $\mathbf{E}\mathbb{E}_Z\left[\left|\hat{G}(t, X)\right|^2\right] \to 0$ as $t \to \infty$. In this case, one can show that for $\mathfrak{A}$ : $L^2(\mathcal{P}) \to L^2(\mathcal{P})$ a linear operator defined by

$$(\mathfrak{A}f)(x) = -c \sum_i \mathbb{E}_{Z,C}\left[\frac{\partial \hat{y}(0, x)}{\partial w_i(C_{i-1}, C_i)} \frac{\partial \hat{y}(0, X)}{\partial w_i(C_{i-1}, C_i)} f(X)\right],$$

with a constant $c > 0$, we have $\partial_t \hat{G}(t, \cdot) = \mathfrak{A}\hat{G}(t, \cdot)$. In particular, $\mathfrak{A}$ is self-adjoint and has non-positive eigenvalues and as such, if at initialization $\hat{G}(0, \cdot)$ lies in the range of $\mathfrak{A}$, we reach the desired conclusion. This is simple in the shallow case. Indeed note that $\hat{G}(0, \cdot) = G^y(0, \cdot)$ by the assumption on initialization of Theorem 9. Assuming $\varphi_L(u) = u$ for simplicity, in the shallow case where $\hat{y}(0, x) = \mathbb{E}_{C_1}\left[w_2(0, C_1, 1)\varphi_1(\langle w_1(0, 1, C_1), x\rangle)\right]$, we have $G^y(0, \cdot)$ is zero-mean Gaussian with the covariance $\mathbf{E}[G^y(0, x)G^y(0, x')]$ equal to

$$\mathbb{C}_{C_1}\left[w_2(0, C_1, 1)\varphi_1(\langle w_1(0, 1, C_1), x\rangle); \ w_2(0, C_1, 1)\varphi_1(\langle w_1(0, 1, C_1), x'\rangle)\right].$$

Immediately from this simple covariance structure, we see that $x \mapsto G^y(0, x)$ lies in the span of $\{x \mapsto \varphi_1(\langle w_1(0, 1, c_1), x\rangle) : c_1 \in \text{supp}(P_1)\}$, i.e. the desired conclusion since $\frac{\partial \hat{y}(0, x)}{\partial w_2(C_1, 1)} = \varphi_1(\langle w_1(0, 1, C_1), x\rangle)$. In the multilayer case, this is no longer obvious due to the complex covariance structure as pointed out in Section 3, and yet interestingly we show that it still indeed holds.

The proof of Theorem 10 leverages on Theorem 9. In particular, implicit in this proof is the fact that the variance reduction effect takes place mostly after a global optimum is reached. Here the technical bulk lies in uniform control on weight movements, which is delicate in the multilayer case.

## 5 Numerical illustration

We give several illustrations in Fig. 1 via a simple experimental setup with MNIST [10] (see the caption and also Appendix G for more details). Fig. 1(a) shows that the network converges to a global optimum around iteration $10^4$ and displays a nonlinear dynamics (which takes the shape of a superposition of two sigmoids). The agreement of the different histogram plots of the output fluctuation for varying widths $N$ in Fig 1(b) verifies the existence of a limiting Gaussian-like behavior, predicted by Theorems 5 and 6. Fig 1(c) shows that the output variance $\mathbf{E}\mathbb{E}_Z\left[\left|\sqrt{N}(\hat{\mathbf{y}}(t, X) - \hat{y}(t, X))\right|^2\right]$ decreases with time quickly after iteration $10^4$, which is predicted by Theorem 10. Fig 1(d) plots the Gaussian component $\tilde{G}^y$ in the output variance. Note that after iteration $10^4$, this component no longer moves since a global optimum is reached. Contrasting this plot with Fig 1(c), we see the central role of GD training (in particular, the second-order MF limit component $R$) in reducing the variance. As shown in Fig 1(e), congruent with Theorem 9, when the network is instead initialized at a global optimum, the output variance is decreasing on the entire period. These plots also highlight an interesting fact (previously mentioned in Section 4): the variance reduction effect takes place mostly after a global optimum is reached.

## Acknowledgement

The work of H. T. Pham is partially supported by a Two Sigma Fellowship.

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
