Supplementary information for **"Limiting fluctuation and trajectorial stability of multilayer neural networks with mean field training"**:

- Appendix C presents the proof of Theorem 3 for well-posedness of $R$.

- Appendix D proves Theorem 5 that connects the neural network with the second-order MF limit at the fluctuation level.

## A    Preliminaries

We introduce notations that are not in the main paper. For convenience, for each index $i \in [L]$, we use $\mathbf{i}$ to refer to either $i$ or the pair $(i-1, i)$, which depends on the context, and we let $|\mathbf{i}| = 1$ and $|\mathbf{i}| = 2$ respectively in each case. For instance, when we write $C_{\mathbf{i}}$, we refer to either $C_i$ or $(C_{i-1}, C_i)$. A statement that is stated for $\mathbf{i}$ should hold in both cases.

For the MF limit:

$$\frac{\partial H_i(t,x,c_i)}{\partial H_j(c'_j)} = \begin{cases} 0, & i \leq j, \\ w_i(t,c'_j,c_i)\varphi'_j(H_j(t,x,c'_j)), & i = j+1, \\ \mathbb{E}_{C'_{j+1}}\left[w_{j+1}(t,c'_j,C'_{j+1})\varphi'_j(H_j(t,x,c'_j))\dfrac{\partial H_i(t,x,c_i)}{\partial H_{j+1}(C'_{j+1})}\right], & i > j+1, \end{cases}$$

$$\frac{\partial H_j(t,x,c_j)}{\partial_* H_j(c_j)} = 1,$$

$$\frac{\partial H_i(t,x,c_i)}{\partial w_j(c'_{j-1},c'_j)} = \begin{cases} 0, & i < j, \\ \dfrac{\partial H_i(t,x,c_i)}{\partial H_j(c'_j)}x, & 1 = j < i, \\ \dfrac{\partial H_i(t,x,c_i)}{\partial H_j(c'_j)}\varphi_{j-1}(H_{j-1}(t,x,c'_{j-1})), & 1 < j < i, \end{cases}$$

$$\frac{\partial H_j(t,x,c_j)}{\partial_* w_j(c'_{j-1},c_j)} = \begin{cases} x, & j = 1, \\ \dfrac{\partial H_j(t,x,c_j)}{\partial_* H_j(c_j)}\varphi_{j-1}(H_{j-1}(t,x,c'_{j-1})), & j > 1, \end{cases}$$

$$\frac{\partial^2 \hat{y}(t,x)}{\partial w_j(c'_{j-1},c'_j)\partial H_L(1)} = \varphi''_L(H_L(t,x,1))\frac{\partial H_L(t,x,1)}{\partial w_j(c'_{j-1},c'_j)},$$

$$\begin{aligned}\frac{\partial^2 \hat{y}(t,x)}{\partial w_j(c'_{j-1},c'_j)\partial H_{i-1}(c_{i-1})} = {}& \mathbb{E}_{C_i}\left[w_i(t,c_{i-1},C_i)\varphi''_{i-1}(H_{i-1}(t,x,c_{i-1}))\frac{\partial H_{i-1}(t,x,c_{i-1})}{\partial w_j(c'_{j-1},c'_j)}\frac{\partial \hat{y}(t,x)}{\partial H_i(C_i)}\right] \\ & + \mathbb{E}_{C_i}\left[w_i(t,c_{i-1},C_i)\varphi'_{i-1}(H_{i-1}(t,x,c_{i-1}))\frac{\partial^2 \hat{y}(t,x)}{\partial w_j(c'_{j-1},c'_j)\partial H_i(C_i)}\right], \quad i \leq L,\end{aligned}$$

$$\frac{\partial^2 \hat{y}(t,x)}{\partial_* w_i(c_{i-1},c'_i)\partial H_{i-1}(c_{i-1})} = \varphi'_{i-1}(H_{i-1}(t,x,c_{i-1}))\frac{\partial \hat{y}(t,x)}{\partial H_i(c'_i)},$$

$$\frac{\partial^2 \hat{y}(t,x)}{\partial_* w_{i-1}(c'_{i-2},c_{i-1})\partial H_{i-1}(c_{i-1})} = \begin{cases} \mathbb{E}_{C_i}\left[w_i(t,c_{i-1},C_i)\varphi''_{i-1}(H_{i-1}(t,x,c_{i-1}))\varphi_{i-2}(H_{i-2}(t,x,c'_{i-2}))\dfrac{\partial \hat{y}(t,x)}{\partial H_i(C_i)}\right], & 2 < i \leq L, \\ \mathbb{E}_{C_2}\left[w_2(t,c_1,C_2)\varphi''_1(H_1(t,x,c_1))x\dfrac{\partial \hat{y}(t,x)}{\partial H_2(C_2)}\right], & i = 2, \\ \dfrac{\partial^2 \hat{y}(t,x)}{\partial w_L(c'_{L-1},1)\partial H_L(1)}, & i = L+1, \end{cases}$$

$$\frac{\partial^2 \hat{y}(t,x)}{\partial w_j(c'_{j-1},c'_j)\partial w_i(c_{i-1},c_i)} = \begin{cases} \dfrac{\partial^2 \hat{y}(t,x)}{\partial w_j(c'_{j-1},c'_j)\partial H_i(c_i)}\varphi_{i-1}(H_{i-1}(t,x,c_{i-1})) \\ \quad + \dfrac{\partial \hat{y}(t,x)}{\partial H_i(c_i)}\varphi'_{i-1}(H_{i-1}(t,x,c_{i-1}))\dfrac{\partial H_{i-1}(t,x,c_{i-1})}{\partial w_j(c'_{j-1},c'_j)}, & i>1, \\ \dfrac{\partial^2 \hat{y}(t,x)}{\partial w_j(c'_{j-1},c'_j)\partial H_1(c_1)}x, & i=1, \end{cases}$$

$$\frac{\partial^2 \hat{y}(t,x)}{\partial_* w_{i-1}(c'_{i-2},c_{i-1})\partial w_i(c_{i-1},c_i)} = \frac{\partial \hat{y}(t,x)}{\partial H_i(c_i)}\varphi'_{i-1}(H_{i-1}(t,x,c_{i-1}))\frac{\partial H_{i-1}(t,x,c_{i-1})}{\partial_* w_{i-1}(c'_{i-2},c_{i-1})},$$

$$\frac{\partial^2 \hat{y}(t,x)}{\partial_* w_i(c'_{i-1},c_i)\partial w_i(c_{i-1},c_i)} = \begin{cases} \varphi_{i-1}(H_{i-1}(t,x,c_{i-1}))\dfrac{\partial^2 \hat{y}(t,x)}{\partial_* w_i(c'_{i-1},c_i)\partial H_i(c_i)}, & i>1, \\ \dfrac{\partial^2 \hat{y}(t,x)}{\partial_* w_1(1,c_1)\partial H_1(c_1)}x, & i=1 \end{cases}$$

$$\frac{\partial^2 \hat{y}(t,x)}{\partial_* w_{i+1}(c_i,c'_{i+1})\partial w_i(c_{i-1},c_i)} = \begin{cases} \varphi_{i-1}(H_{i-1}(t,x,c_{i-1}))\dfrac{\partial^2 \hat{y}(t,x)}{\partial_* w_{i+1}(c_i,c'_{i+1})\partial H_i(c_i)}, & i>1, \\ \dfrac{\partial^2 \hat{y}(t,x)}{\partial_* w_2(c_1,c'_2)\partial H_1(c_1)}x, & i=1. \end{cases}$$

With this, we write the dynamics (3) for the second-order MF limit $R$ in its complete form as follows:

$$\partial_t R_i(G,t,c_{i-1},c_i) =$$
$$- \mathbb{E}_Z\left[\frac{\partial \hat{y}(t,X)}{\partial w_i(c_{i-1},c_i)}\partial_2^2 \mathcal{L}(Y,\hat{y}(t,X))\left(G^y(t,X) + \sum_{r=1}^L \mathbb{E}_C\left[R_r(G,t,C_{r-1},C_r)\frac{\partial \hat{y}(t,X)}{\partial w_r(C_{r-1},C_r)}\right]\right)\right]$$
$$- \mathbb{E}_Z\left[\partial_2 \mathcal{L}(Y,\hat{y}(t,X))\mathbb{E}_C\left[\sum_{r=1}^L R_r(G,t,C_{r-1},C_r)\frac{\partial^2 \hat{y}(t,X)}{\partial w_r(C_{r-1},C_r)\partial w_i(c_{i-1},c_i)}\right]\right]$$
$$- \mathbb{E}_Z\left[\partial_2 \mathcal{L}(Y,\hat{y}(t,X))\mathbb{E}_C\left[R_{i-1}(G,t,C_{i-2},c_{i-1})\frac{\partial^2 \hat{y}(t,X)}{\partial_* w_{i-1}(C_{i-2},c_{i-1})\partial w_i(c_{i-1},c_i)}\right]\right]$$
$$- \mathbb{E}_Z\left[\partial_2 \mathcal{L}(Y,\hat{y}(t,X))\mathbb{E}_C\left[R_i(G,t,C_{i-1},c_i)\frac{\partial^2 \hat{y}(t,X)}{\partial_* w_i(C_{i-1},c_i)\partial w_i(c_{i-1},c_i)}\right]\right]$$
$$- \mathbb{E}_Z\left[\partial_2 \mathcal{L}(Y,\hat{y}(t,X))\mathbb{E}_C\left[R_{i+1}(G,t,c_i,C_{i+1})\frac{\partial^2 \hat{y}(t,X)}{\partial_* w_{i+1}(c_i,C_{i+1})\partial w_i(c_{i-1},c_i)}\right]\right]$$
$$- \mathbb{E}_Z\left[\partial_2 \mathcal{L}(Y,\hat{y}(t,X))G_i^w(t,c_{i-1},c_i,X)\right] \tag{6}$$

where we take by convention that $R_0 = R_{L+1} = 0$. We also define secondary quantities, similar to those in Section 3.1 e.g. $\frac{\partial^2 \tilde{y}(t,x)}{\partial w_j(c'_{j-1},c'_j)\partial w_i(c_{i-1},c_i)}$, in a similar fashion, by taking their MF counterparts and replacing $\mathbb{E}_{C_i}$ being replaced by an empirical average over $\{C_i(j_i)\}_{j_i \in [N_i]}$.

We also recall the following bounds from [17].

**Lemma 11.** *Under Assumption 1, for any $T \geq 0$,*

$$\max_{i \in [L]} m^{-1/2}\mathbb{E}_C\left[\sup_{t \leq T}|w_i(t,C_{i-1},C_i)|^m\right]^{1/m} \leq K_T.$$

Boundedness of moments of several other MF quantities at any time $t \leq T$ are consequences of this lemma and Assumption 1. We omit the details.

# B  CLT for the Gaussian component $\tilde{G}$: Proof of Theorem 2

We recall the process $\tilde{G}$ defined in Section 3.1. Recall that for each $j_i \in [N_i]$, we sample $C_i(j_i) \in \Omega_i$ independently at random from $P_i$. Let $S_i = \{C_i(1),\ldots,C_i(N_i)\}$. We denote by $\mathbf{E}_i$ for the expectation over the random choice of $S_i$. We also recall that $\mathbf{E}$ is the expectation over the random choice of $S_i$, $i=1,\ldots,L$.

We will use the following notation throughout. Let $\delta$ be a function of $t \in \mathbb{T}$, $z \in \mathbb{Z}$, $c_1 \in \Omega_1, c_2 \in \Omega_2, \ldots, c_L \in \Omega_L$, and $j_1 \in [N_1], j_2 \in [N_2], \ldots, j_L \in [N_L]$ (where $\delta$ may not necessarily depend on all those variables). For $o_N$ decreasing in $N$, we write $\delta = \mathbf{O}_T(o_N)$ if for any $t \leq T$,

$$\mathbf{E}\mathbb{E}_J\mathbb{E}_{Z,C}[|\delta(t, Z, C_1, \ldots, C_L, J_1, \ldots, J_L|^2] \leq K_T o_N^2$$

for sufficiently large $N$. Thus, if we write

$$f(t, z, c_1, c_2, \ldots, c_L) = g(t, z, c_1, c_2, \ldots, c_L) + \mathbf{O}_T(o_N)$$

we mean

$$\mathbf{E}\mathbb{E}_J\mathbb{E}_{Z,C}[|f(t, Z, C_1, \ldots, C_L, J_1, \ldots, J_L) - g(t, Z, C_1, \ldots, C_L, J_1, \ldots, J_L)|^2] \leq K_T o_N^2$$

for all $t \leq T$, for sufficiently large $N$.

We define additional processes as follows

$$\tilde{G}_i^{\partial\tilde{H}}(t, c_i, x) = \sqrt{N}\left(\frac{\partial\tilde{y}(t, x)}{\partial\tilde{H}_i(c_i)} - \frac{\partial\hat{y}(t, x)}{\partial H_i(c_i)}\right),$$

$$\tilde{G}_{i-1}^{\partial H}(t, c_{i-1}, x) = \sqrt{N}\left(\mathbb{E}_{J_i}\left[w_i(t, c_{i-1}, C_i(J_i))\frac{\partial\hat{y}(t, x)}{\partial H_i(C_i(J_i))}\right] - \mathbb{E}_{C_i}\left[w_i(t, c_{i-1}, C_i)\frac{\partial\hat{y}(t, x)}{\partial H_i(C_i)}\right]\right)\varphi'_{i-1}\left(H_{i-1}(t, x, c_{i-1})\right),$$

$$\tilde{G}_L^{\partial H}(t, 1, x) = \tilde{G}_{L-1}^{\partial H}(t, c_{L-1}, x) = 0,$$

$$\tilde{G}_i^{\tilde{H}}(t, c_i, x) = \sqrt{N}\left(\tilde{H}_i(t, x, c_i) - H_i(t, x, c_i)\right),$$

$$\tilde{G}_i^H(t, c_i, x) = \sqrt{N}\left(\mathbb{E}_{J_{i-1}}[w_i(t, C_{i-1}(J_{i-1}), c_i)\varphi_{i-1}(H_{i-1}(t, x, C_{i-1}(J_{i-1})))] - \mathbb{E}_{C_{i-1}}[w_i(t, C_{i-1}, c_i)\varphi_{i-1}(H_{i-1}(t, x, C_{i-1}))]\right),$$

$$\tilde{G}_1^H(t, c_1, x) = 0.$$

The next lemma is a key tool in the argument.

**Lemma 12.** *For an index* $\mathbf{i}$*, let* $f(S, c_\mathbf{i}, c_{\mathbf{i}'})$ *be a function of* $S = \{C_\mathbf{i}(j_\mathbf{i}) : j_\mathbf{i} \in [N_\mathbf{i}]\} \cup \{S'\}$ *and* $c_\mathbf{i}, c_{\mathbf{i}'}$*, where* $S'$ *denotes some random variable that is independent of* $\{C_\mathbf{i}(j_\mathbf{i}) : j_\mathbf{i} \in [N_\mathbf{i}]\}$*. Assume that for some* $\alpha > 0$*,*

$$\mathbf{E}\mathbb{E}_{C_{\mathbf{i}'}}\left[\left|f(S^{j_\mathbf{i}}, C_\mathbf{i}(j_\mathbf{i}), C_{\mathbf{i}'})\right|^2\right] \leq K,$$

$$\mathbf{E}\mathbb{E}_{C_{\mathbf{i}'}}\left[\left|f(S, C_\mathbf{i}(j_\mathbf{i}), C_{\mathbf{i}'}) - f(S^{j_\mathbf{i}}, C_\mathbf{i}(j_\mathbf{i}), C_{\mathbf{i}'})\right|^2\right] \leq K/N^\alpha,$$

*for all* $j_\mathbf{i}$*, where we define* $S^{j_\mathbf{i}}$ *similar to* $S$ *except with* $C_\mathbf{i}(j_\mathbf{i})$ *in* $S_\mathbf{i}$ *replaced by an independent copy* $C'_\mathbf{i}(j_\mathbf{i})$ *and* $\mathbf{E}$ *denotes the expectation w.r.t* $S$*. Then we have*

$$\mathbb{E}_{J_\mathbf{i}}[f(S, C_\mathbf{i}(J_\mathbf{i}), c_{\mathbf{i}'})] = \mathbb{E}_{C_\mathbf{i}}[f(S, C_\mathbf{i}, c_{\mathbf{i}'})] + \mathbf{O}(N^{-\alpha/4} + N^{-|\mathbf{i}|/2}).$$

*Proof.* We have

$$\mathbf{E}\mathbb{E}_{C_{\mathbf{i}'}}\left[|\mathbb{E}_{J_\mathbf{i}}[f(S, C_\mathbf{i}(J_\mathbf{i}), C_{\mathbf{i}'})] - \mathbb{E}_{C_\mathbf{i}}[f(S, C_\mathbf{i}, C_{\mathbf{i}'})]|^2\right]$$

$$= \frac{1}{N^{2|\mathbf{i}|}}\sum_{j_\mathbf{i}, j'_\mathbf{i}}\mathbf{E}\mathbb{E}_{C_{\mathbf{i}'}}[(f(S, C_\mathbf{i}(j_\mathbf{i}), C_{\mathbf{i}'}) - \mathbb{E}_{C_\mathbf{i}}[f(S, C_\mathbf{i}, C_{\mathbf{i}'})]) \cdot (f(S, C_\mathbf{i}(j'_\mathbf{i}), C_{\mathbf{i}'}) - \mathbb{E}_{C_\mathbf{i}}[f(S, C_\mathbf{i}, C_{\mathbf{i}'})])].$$

Define $S^{j_\mathbf{i}}$ – with an abuse of notation – similar to $S$ but with the terms involving $C_\mathbf{i}(j_\mathbf{i})$ and $C_\mathbf{i}(j'_\mathbf{i})$ replaced by independent copies $C'_\mathbf{i}(j_\mathbf{i})$ and $C'_\mathbf{i}(j'_\mathbf{i})$. (In particular, $S^{j_\mathbf{i}}$ has the same distribution as $S$.) We have:

$$\left|\mathbf{E}\left[(f(S, C_\mathbf{i}(j_\mathbf{i}), c_{\mathbf{i}'}) - \mathbb{E}_{C_\mathbf{i}}[f(S, C_\mathbf{i}, c_{\mathbf{i}'})])(f(S, C_\mathbf{i}(j'_\mathbf{i}), c_{\mathbf{i}'}) - \mathbb{E}_{C_\mathbf{i}}[f(S, C_\mathbf{i}, c_{\mathbf{i}'})])\right]\right.$$

$$\left. - \mathbf{E}\left[(f(S^{j_\mathbf{i}}, C_\mathbf{i}(j_\mathbf{i}), c_{\mathbf{i}'}) - \mathbb{E}_{C_\mathbf{i}}[f(S^{j_\mathbf{i}}, C_\mathbf{i}, c_{\mathbf{i}'})])(f(S^{j_\mathbf{i}}, C_\mathbf{i}(j'_\mathbf{i}), c_{\mathbf{i}'}) - \mathbb{E}_{C_\mathbf{i}}[f(S^{j_\mathbf{i}}, C_\mathbf{i}, c_{\mathbf{i}'})])\right]\right|$$

$$= \left|\mathbf{E}\left[f(S, C_\mathbf{i}(j_\mathbf{i}), c_{\mathbf{i}'})f(S, C_\mathbf{i}(j'_\mathbf{i}), c_{\mathbf{i}'}) - f(S^{j_\mathbf{i}}, C_\mathbf{i}(j_\mathbf{i}), c_{\mathbf{i}'})f(S^{j_\mathbf{i}}, C_\mathbf{i}(j'_\mathbf{i}), c_{\mathbf{i}'})\right]\right|$$

$$\leq K\mathbf{E}\left[|f(S, C_\mathbf{i}(j_\mathbf{i}), c_{\mathbf{i}'})(f(S, C_\mathbf{i}(j'_\mathbf{i}), c_{\mathbf{i}'}) - f(S^{j_\mathbf{i}}, C_\mathbf{i}(j'_\mathbf{i}), c_{\mathbf{i}'}))|\right] + K\mathbf{E}\left[|f(S^{j_\mathbf{i}}, C_\mathbf{i}(j'_\mathbf{i}), c_{\mathbf{i}'})(f(S, C_\mathbf{i}(j_\mathbf{i}), c_{\mathbf{i}'}) - f(S^{j_\mathbf{i}}, C_\mathbf{i}(j_\mathbf{i}), c_{\mathbf{i}'}))|\right]$$

$$\leq K\mathbf{E}\left[(f(S, C_\mathbf{i}(j_\mathbf{i}), c_{\mathbf{i}'}))^2\right]^{1/2}\mathbf{E}\left[(f(S, C_\mathbf{i}(j'_\mathbf{i}), c_{\mathbf{i}'}) - f(S^{j_\mathbf{i}}, C_\mathbf{i}(j'_\mathbf{i}), c_{\mathbf{i}'}))^2\right]^{1/2}$$

$$+ K\mathbf{E}\left[(f(S^{j_\mathbf{i}}, C_\mathbf{i}(j'_\mathbf{i}), c_{\mathbf{i}'}))^2\right]^{1/2}\mathbf{E}\left[(f(S, C_\mathbf{i}(j_\mathbf{i}), c_{\mathbf{i}'}) - f(S^{j_\mathbf{i}}, C_\mathbf{i}(j_\mathbf{i}), c_{\mathbf{i}'}))^2\right]^{1/2}$$

$$\leq K\mathbf{E}\left[\left(f(S,C_{\mathbf{i}}(j'_{\mathbf{i}}),c_{\mathbf{i'}}) - f(S^{j_{\mathbf{i}}},C_{\mathbf{i}}(j'_{\mathbf{i}}),c_{\mathbf{i'}})\right)^2\right]^{1/2}\mathbf{E}\left[\left(f(S,C_{\mathbf{i}}(j_{\mathbf{i}}),c_{\mathbf{i'}}) - f(S^{j_{\mathbf{i}}},C_{\mathbf{i}}(j_{\mathbf{i}}),c_{\mathbf{i'}})\right)^2\right]^{1/2}$$

$$+ K\mathbf{E}\left[\left(f(S,C_{\mathbf{i}}(j'_{\mathbf{i}}),c_{\mathbf{i'}}) - f(S^{j_{\mathbf{i}}},C_{\mathbf{i}}(j'_{\mathbf{i}}),c_{\mathbf{i'}})\right)^2\right]^{1/2}\mathbf{E}\left[\left(f(S^{j_{\mathbf{i}}},C_{\mathbf{i}}(j_{\mathbf{i}}),c_{\mathbf{i'}})\right)^2\right]^{1/2}$$

$$+ K\mathbf{E}\left[\left(f(S^{j_{\mathbf{i}}},C_{\mathbf{i}}(j'_{\mathbf{i}}),c_{\mathbf{i'}})\right)^2\right]^{1/2}\mathbf{E}\left[\left(f(S,C_{\mathbf{i}}(j_{\mathbf{i}}),c_{\mathbf{i'}}) - f(S^{j_{\mathbf{i}}},C_{\mathbf{i}}(j_{\mathbf{i}}),c_{\mathbf{i'}})\right)^2\right]^{1/2}.$$

Furthermore,

$$\mathbb{E}_{C_{\mathbf{i'}}}\left[\mathbf{E}\left[\left(f(S,C_{\mathbf{i}}(j'_{\mathbf{i}}),C_{\mathbf{i'}}) - f(S^{j_{\mathbf{i}}},C_{\mathbf{i}}(j'_{\mathbf{i}}),C_{\mathbf{i'}})\right)^2\right]^{1/2}\mathbf{E}\left[\left(f(S^{j_{\mathbf{i}}},C_{\mathbf{i}}(j_{\mathbf{i}}),C_{\mathbf{i'}})\right)^2\right]^{1/2}\right]$$

$$\leq \mathbb{E}_{C_{\mathbf{i'}}}\mathbf{E}\left[\left(f(S,C_{\mathbf{i}}(j'_{\mathbf{i}}),C_{\mathbf{i'}}) - f(S^{j_{\mathbf{i}}},C_{\mathbf{i}}(j'_{\mathbf{i}}),C_{\mathbf{i'}})\right)^2\right]^{1/2}\mathbb{E}_{C_{\mathbf{i'}}}\mathbf{E}\left[\left(f(S^{j_{\mathbf{i}}},C_{\mathbf{i}}(j_{\mathbf{i}}),C_{\mathbf{i'}})\right)^2\right]^{1/2}$$

$$\leq K/N^{\alpha/2},$$

and the other two terms can be bounded similarly. Now recall that $S^{j_{\mathbf{i}}}$ is independent of $C_{\mathbf{i}}(j_{\mathbf{i}})$ and $C_{\mathbf{i}}(j'_{\mathbf{i}})$. Let $\mathcal{S}^{j_{\mathbf{i}}}$ be the $\sigma$-algebra generated by $S'$ and $C_{\mathbf{k}}(j_{\mathbf{k}})$ for all $j_{\mathbf{k}}$ except $j_{\mathbf{i}}$ and $j'_{\mathbf{i}}$. Then if $j_{\mathbf{i}} \neq j'_{\mathbf{i}}$,

$$\mathbf{E}\left[\left(f(S^{j_{\mathbf{i}}},C_{\mathbf{i}}(j_{\mathbf{i}}),c_{\mathbf{i'}}) - \mathbb{E}_{C_{\mathbf{i}}}[f(S^{j_{\mathbf{i}}},C_{\mathbf{i}},c_{\mathbf{i'}})]\right)\left(f(S^{j_{\mathbf{i}}},C_{\mathbf{i}}(j'_{\mathbf{i}}),c_{\mathbf{i'}}) - \mathbb{E}_{C_{\mathbf{i}}}[f(S^{j_{\mathbf{i}}},C_{\mathbf{i}},c_{\mathbf{i'}})]\right)\right]$$

$$= \mathbf{E}_{\mathcal{S}}\left[\mathbf{E}_{C_{\mathbf{i}}(j_{\mathbf{i}}),C_{\mathbf{i}}(j'_{\mathbf{i}})}\left[\left(f(S^{j_{\mathbf{i}}},C_{\mathbf{i}}(j_{\mathbf{i}}),c_{\mathbf{i'}}) - \mathbb{E}_{C_{\mathbf{i}}}[f(S^{j_{\mathbf{i}}},C_{\mathbf{i}},c_{\mathbf{i'}})]\right)\left(f(S^{j_{\mathbf{i}}},C_{\mathbf{i}}(j'_{\mathbf{i}}),c_{\mathbf{i'}}) - \mathbb{E}_{C_{\mathbf{i}}}[f(S^{j_{\mathbf{i}}},C_{\mathbf{i}},c_{\mathbf{i'}})]\right)\right]\right]$$

$$= 0.$$

Thus under the assumptions in the lemma,

$$\mathbf{E}\mathbb{E}_{C_{\mathbf{i'}}}\left[\left(\mathbb{E}_{J_{\mathbf{i}}}[f(S,C_{\mathbf{i}}(J_{\mathbf{i}}),C_{\mathbf{i'}})] - \mathbb{E}_{C_{\mathbf{i}}}[f(S,C_{\mathbf{i}},C_{\mathbf{i'}})]\right)^2\right] \leq \frac{K}{N^{\alpha/2}} + \frac{K}{N^{|\mathbf{i}|}}.$$

$\square$

We proceed in several sections.

### B.1   A priori moment estimates

**Lemma 13.** *Under Assumption 1, we have for $t \leq T$,*

$$\mathbf{E}\mathbb{E}_{C_i}\left[\left(\tilde{G}_i^H(t,x,C_i)\right)^{2p}\right], \ \mathbf{E}\mathbb{E}_{J_i}\left[\left(\tilde{G}_i^H(t,x,C_i(J_i))\right)^{2p}\right], \ \mathbf{E}\mathbb{E}_{C_i}\left[\left(\tilde{G}_i^{\tilde{H}}(t,x,C_i)\right)^{2p}\right], \ \mathbf{E}\mathbb{E}_{C_i}\left[\left(\tilde{G}_i^{\tilde{H}}(t,x,C_i(J_i))\right)^{2p}\right] \leq K_{T,p}.$$

*Proof.* Recall Lemma 11. The base case $i = 1$ easily follows. Notice that

$$\mathbf{E}\mathbb{E}_{J_{i-1}}\left[\left(\tilde{G}_{i-1}^H(t,x,C_{i-1}(J_{i-1}))\right)^{2p}\right] = \mathbf{E}\mathbb{E}_{C_{i-1}}\left[\left(\tilde{G}_{i-1}^H(t,x,C_{i-1})\right)^{2p}\right],$$

$$\mathbf{E}\mathbb{E}_{J_{i-1}}\left[\left(\tilde{G}_{i-1}^{\tilde{H}}(t,x,C_{i-1}(J_{i-1}))\right)^{2p}\right] = \mathbf{E}\mathbb{E}_{C_{i-1}}\left[\left(\tilde{G}_{i-1}^{\tilde{H}}(t,x,C_{i-1})\right)^{2p}\right],$$

where the second claim is because the randomness of $\tilde{G}_{i-1}^{\tilde{H}}(t,x,c_{i-1})$ comes from $\{C_k(j_k): \ j_k \in [N_k], \ k < i-1\}$.

Note that

$$\tilde{G}_i^{\tilde{H}}(t,c_i,x)$$

$$= \sqrt{N}\left(\tilde{H}_i(t,x,c_i) - H_i(t,x,c_i)\right)$$

$$= \sqrt{N}\left(\mathbb{E}_{J_{i-1}}[w_i(t,C_{i-1}(J_{i-1}),c_i)\varphi_{i-1}(\tilde{H}_{i-1}(t,x,C_{i-1}(J_{i-1})))] - \mathbb{E}_{C_{i-1}}[w_i(t,C_{i-1},c_i)\varphi_{i-1}(H_{i-1}(t,x,C_{i-1}))]\right)$$

$$= \tilde{G}_i^H(t,x,c_i) + \sqrt{N}\mathbb{E}_{J_{i-1}}[w_i(t,C_{i-1}(J_{i-1}),c_i)\left(\varphi_{i-1}(\tilde{H}_{i-1}(t,x,C_{i-1}(J_{i-1}))) - \varphi_{i-1}(H_{i-1}(t,x,C_{i-1}(J_{i-1})))\right)].$$

We have

$$\mathbf{E}\mathbb{E}_{C_i}\left[\left(\tilde{G}_i^H(t,x,C_i)\right)^{2p}\right]$$

$$\leq \mathbf{E}\mathbb{E}_{C_i}\left[\left(\sqrt{N}\left(\mathbb{E}_{J_{i-1}}[w_i(t,C_{i-1}(J_{i-1}),C_i)\varphi_{i-1}(H_{i-1}(t,x,C_{i-1}(J_{i-1})))] - \mathbb{E}_{C_{i-1}}[w_i(t,C_{i-1},C_i)\varphi_{i-1}(H_{i-1}(t,x,C_{i-1}))]\right)\right)^{2p}\right]$$

$$\leq \mathbb{E}_{C_i}\left[\mathbb{V}_{C_{i-1}}[w_i(t,C_{i-1},C_i)\varphi_{i-1}(H_{i-1}(t,x,C_{i-1}))]^p\right] + \frac{K_{T,p}}{N}$$

$$\leq K_{T,p}.$$

We also have:

$$\mathbf{E}\mathbb{E}_{C_i}\left[\left(\sqrt{N}\mathbb{E}_{J_{i-1}}\left[w_i(t,C_{i-1}(J_{i-1}),C_i)\left(\varphi_{i-1}(\tilde{H}_{i-1}(t,x,C_{i-1}(J_{i-1}))) - \varphi_{i-1}(H_{i-1}(t,x,C_{i-1}(J_{i-1})))\right)\right]\right)^{2p}\right]$$

$$\leq \mathbf{E}\mathbb{E}_{C_i}\left[\left(\mathbb{E}_{J_{i-1}}\left[w_i(t,C_{i-1}(J_{i-1}),C_i)\tilde{G}_{i-1}^{\tilde{H}}(t,x,C_{i-1}(J_{i-1}))\right]\right)^{2p}\right]$$

$$\leq \mathbf{E}\mathbb{E}_{C_i,J_{i-1}}\left[|w_i(t,C_{i-1}(J_{i-1}),C_i)|^{4p}\right]^{1/2}\mathbf{E}\mathbb{E}_{J_{i-1}}\left[\left(\tilde{G}_{i-1}^{\tilde{H}}(t,x,C_{i-1}(J_{i-1}))\right)^{4p}\right]^{1/2}$$

$$\leq K_{T,p}\mathbf{E}\mathbb{E}_{J_{i-1}}\left[\left(\tilde{G}_{i-1}^{\tilde{H}}(t,x,C_{i-1}(J_{i-1}))\right)^{4p}\right]^{1/2}.$$

Hence, by the induction hypothesis, we obtain

$$\mathbf{E}\mathbb{E}_{C_i}\left[\left(\tilde{G}_i^{\tilde{H}}(t,x,C_i)\right)^{2p}\right] \leq K_{T,p}.$$

$\square$

**Lemma 14.** *Under Assumption 1, we have for any $t \leq T$,*

$$\mathbf{E}\mathbb{E}_{C_i}\left[\left(\tilde{G}_i^{\partial H}(t,C_i,x)\right)^{2p}\right],\ \mathbf{E}\mathbb{E}_{J_{i-1}}\left[\left(\tilde{G}_{i-1}^{\partial H}(t,C_{i-1}(J_{i-1}),x)\right)^{2p}\right] \leq K_{T,p},$$

$$\mathbf{E}\mathbb{E}_{C_i}\left[\left(\tilde{G}_i^{\partial \tilde{H}}(t,C_i,x)\right)^{2p}\right],\ \mathbf{E}\mathbb{E}_{J_i}\left[\left(\tilde{G}_i^{\partial \tilde{H}}(t,C_i(J_i),x)\right)^{2p}\right] \leq K_{T,p}.$$

*Proof.* Recall that $\tilde{G}_L^{\partial H} = 0$. For $\tilde{G}_L^{\partial \tilde{H}}$:

$$\mathbf{E}\left[\left(\tilde{G}_L^{\partial \tilde{H}}(t,1,x)\right)^{2p}\right] = \mathbf{E}\left[\left(\sqrt{N}\left(\frac{\partial \tilde{y}(t,x)}{\partial \tilde{H}_L(1)} - \frac{\partial \hat{y}(t,x)}{\partial H_L(1)}\right)\right)^{2p}\right]$$

$$= \mathbf{E}\left[\left(\sqrt{N}\left(\varphi_L'\left(\tilde{H}_L(t,x,1)\right) - \varphi_L'\left(H_L(t,x,1)\right)\right)\right)^{2p}\right]$$

$$\leq K\mathbf{E}\left[\left(\tilde{G}_L^{\tilde{H}}(t,1,x)\right)^{2p}\right]$$

$$\leq K_{T,p}$$

by Lemma 13.

We note that

$$\tilde{G}_{i-1}^{\partial \tilde{H}}(t,c_{i-1},x)$$
$$= \sqrt{N}\left(\frac{\partial \tilde{y}(t,x)}{\partial \tilde{H}_{i-1}(c_{i-1})} - \frac{\partial \hat{y}(t,x)}{\partial H_{i-1}(c_{i-1})}\right)$$
$$= \sqrt{N}\left(\mathbb{E}_{J_i}\left[w_i(t,c_{i-1},C_i(J_i))\varphi_{i-1}'(\tilde{H}_{i-1}(t,x,c_{i-1}))\frac{\partial \tilde{y}(t,x)}{\partial \tilde{H}_i(C_i(J_i))}\right] - \mathbb{E}_{C_i}\left[w_i(t,c_{i-1},C_i)\varphi_{i-1}'(H_{i-1}(t,x,c_{i-1}))\frac{\partial \hat{y}(t,x)}{\partial H_i(C_i)}\right]\right)$$
$$= \varphi_{i-1}'(H_{i-1}(t,x,c_{i-1}))\tilde{G}_{i-1}^{\partial H}(t,c_{i-1},x)$$
$$\quad + \sqrt{N}\left(\varphi_{i-1}'(\tilde{H}_{i-1}(t,x,c_{i-1})) - \varphi_{i-1}'(H_{i-1}(t,x,c_{i-1}))\right)\mathbb{E}_{J_i}\left[w_i(t,c_{i-1},C_i(J_i))\frac{\partial \hat{y}(t,x)}{\partial H_i(C_i(J_i))}\right]$$
$$\quad + \varphi_{i-1}'(\tilde{H}_{i-1}(t,x,c_{i-1}))\mathbb{E}_{J_i}\left[w_i(t,c_{i-1},C_i(J_i))\tilde{G}_i^{\partial \tilde{H}}(t,C_i(J_i),x)\right].$$

We bound each term. We have for $i < L$,

$$\mathbf{E}\mathbb{E}_{C_{i-1}}\left[\left(\tilde{G}_{i-1}^{\partial H}(t, C_{i-1}, x)\right)^{2p}\right]$$

$$\leq K_p \mathbf{E}\mathbb{E}_{C_{i-1}}\left[\left(\sqrt{N}\left(\mathbb{E}_{J_i}\left[w_i(t, C_{i-1}, C_i(J_i))\frac{\partial \hat{y}(t,x)}{\partial H_i(C_i(J_i))}\right] - \mathbb{E}_{C_i}\left[w_i(t, C_{i-1}, C_i)\frac{\partial \hat{y}(t,x)}{\partial H_i(C_i)}\right]\right)\right)^{2p}\right]$$

$$\leq K_p \mathbb{E}_{C_{i-1}}\left[\mathbb{V}_{C_i}\left[w_i(t, C_{i-1}, C_i)\frac{\partial \hat{y}(t,x)}{\partial H_i(C_i)}\right]^p\right] + \frac{K_{T,p}}{N}$$

$$\leq K_{T,p},$$

and similarly,

$$\mathbf{E}\mathbb{E}_{J_{i-1}}\left[\left(\tilde{G}_{i-1}^{\partial H}(t, C_{i-1}(J_{i-1}), x)\right)^{2p}\right] \leq K_{T,p}.$$

The same of course holds in the case $i = L$ since $\tilde{G}_{L-1}^{\partial H}(t, c_{L-1}, x) = 0$. By Lemma 13,

$$\mathbf{E}\mathbb{E}_{C_{i-1}}\left[\left(\sqrt{N}\left(\varphi'_{i-1}(\tilde{H}_{i-1}(t, x, C_{i-1})) - \varphi'_{i-1}(H_{i-1}(t, x, C_{i-1}))\right)\mathbb{E}_{J_i}\left[w_i(t, C_{i-1}, C_i(J_i))\frac{\partial \hat{y}(t,x)}{\partial H_i(C_i(J_i))}\right]\right)^{2p}\right]$$

$$\leq K\mathbf{E}\mathbb{E}_{C_{i-1}}\left[\left(\tilde{G}_{i-1}^{\tilde{H}}(t, C_{i-1}, x)\mathbb{E}_{J_i}\left[w_i(t, C_{i-1}, C_i(J_i))\frac{\partial \hat{y}(t,x)}{\partial H_i(C_i(J_i))}\right]\right)^{2p}\right]$$

$$\leq K\mathbf{E}\mathbb{E}_{C_{i-1}}\left[\left(\tilde{G}_{i-1}^{\tilde{H}}(t, C_{i-1}, x)\right)^{4p}\right]^{1/2}\mathbf{E}\mathbb{E}_{C_{i-1}}\left[\mathbb{E}_{J_i}\left[w_i(t, C_{i-1}, C_i(J_i))\frac{\partial \hat{y}(t,x)}{\partial H_i(C_i(J_i))}\right]^{4p}\right]^{1/2}$$

$$\leq K_{T,p},$$

and by noticing that $\mathbf{E}\mathbb{E}_{J_{i-1}}\left[\left(\tilde{G}_{i-1}^{\tilde{H}}(t, C_{i-1}(J_{i-1}), x)\right)^{4p}\right] = \mathbf{E}\mathbb{E}_{C_{i-1}}\left[\left(\tilde{G}_{i-1}^{\tilde{H}}(t, C_{i-1}, x)\right)^{4p}\right]$, we have similarly:

$$\mathbf{E}\mathbb{E}_{J_{i-1}}\left[\left(\sqrt{N}\left(\varphi'_{i-1}(\tilde{H}_{i-1}(t, x, C_{i-1}(J_{i-1}))) - \varphi'_{i-1}(H_{i-1}(t, x, C_{i-1}(J_{i-1})))\right)\mathbb{E}_{J_i}\left[w_i(t, C_{i-1}(J_{i-1}), C_i(J_i))\frac{\partial \hat{y}(t,x)}{\partial H_i(C_i(J_i))}\right]\right)^{2p}\right]$$

$$\leq K_{T,p}.$$

We also have:

$$\mathbf{E}\mathbb{E}_{C_{i-1}}\left[\mathbb{E}_{J_i}\left[w_i(t, C_{i-1}, C_i(J_i))\tilde{G}_i^{\partial \tilde{H}}(t, C_i(J_i), x)\right]^{2p}\right]$$

$$\leq \mathbf{E}\mathbb{E}_{C_{i-1}}\mathbb{E}_{J_i}\left[\left|w_i(t, C_{i-1}, C_i(J_i))\tilde{G}_i^{\partial \tilde{H}}(t, C_i(J_i), x)\right|^{2p}\right]$$

$$\leq \mathbf{E}\mathbb{E}_{C_{i-1}}\mathbb{E}_{J_i}\left[|w_i(t, C_{i-1}, C_i(J_i))|^{4p}\right]^{1/2}\mathbf{E}\mathbb{E}_{J_i}\left[\left|\tilde{G}_i^{\partial \tilde{H}}(t, C_i(J_i), x)\right|^{4p}\right]^{1/2}$$

$$\leq K_{T,p}$$

by the induction hypothesis, and similarly,

$$\mathbf{E}\mathbb{E}_{J_{i-1}}\left[\mathbb{E}_{J_i}\left[w_i(t, C_{i-1}(J_{i-1}), C_i(J_i))\tilde{G}_i^{\partial \tilde{H}}(t, C_i(J_i), x)\right]^{2p}\right] \leq K_{T,p}.$$

This completes the proof. □

## B.2 Relation among $\tilde{G}$ quantities

**Lemma 15.** *Under Assumption 1, we have for $t \leq T$,*

$$\tilde{G}_1^{\tilde{H}}(t, c_1, x) = 0,$$

$$\tilde{G}_i^{\tilde{H}}(t, c_i, x) = \tilde{G}_i^H(t, c_i, x) + \mathbb{E}_{C_{i-1}}\left[w_i(t, C_{i-1}, c_i)\varphi'_{i-1}(H_{i-1}(t, x, C_{i-1}))\tilde{G}_{i-1}^{\tilde{H}}(t, x, C_{i-1})\right] + \mathbf{O}_T(N^{-1/2}).$$

*Proof.* We have
$$\tilde{G}_1^{\tilde{H}}(t, c_1, x) = \sqrt{N}(\tilde{H}_1(t, x, c_1) - H_1(t, x, c_1)) = 0,$$
and
$$\tilde{G}_i^{\tilde{H}}(t, c_i, x)$$
$$= \sqrt{N}\left(\tilde{H}_i(t, x, c_i) - H_i(t, x, c_i)\right)$$
$$= \sqrt{N}\left(\mathbb{E}_{J_{i-1}}[w_i(t, C_{i-1}(J_{i-1}), c_i)\varphi_{i-1}(\tilde{H}_{i-1}(t, x, C_{i-1}(J_{i-1})))] - \mathbb{E}_{C_{i-1}}[w_i(t, C_{i-1}, c_i)\varphi_{i-1}(H_{i-1}(t, x, C_{i-1}))]\right)$$
$$= \sqrt{N}\tilde{G}_i^H(t, c_i, x)$$
$$\quad + \sqrt{N}\mathbb{E}_{J_{i-1}}\left[w_i(t, C_{i-1}(J_{i-1}), c_i)\left(\varphi_{i-1}(\tilde{H}_{i-1}(t, x, C_{i-1}(J_{i-1}))) - \varphi_{i-1}(H_{i-1}(t, x, C_{i-1}(J_{i-1})))\right)\right].$$

We note that by the mean value theorem, for some $h_{i-1}(t, x, C_{i-1}(J_{i-1}))$ between $H_{i-1}(t, x, C_{i-1}(J_{i-1}))$ and $\tilde{H}_{i-1}(t, x, C_{i-1}(J_{i-1}))$:

$$\mathbb{E}_{C_i}\mathbf{E}\left[\mathbb{E}_{J_{i-1}}\left[\sqrt{N}w_i(t, C_{i-1}(J_{i-1}), C_i)\left(\varphi_{i-1}(\tilde{H}_{i-1}(t, x, C_{i-1}(J_{i-1}))) - \varphi_{i-1}(H_{i-1}(t, x, C_{i-1}(J_{i-1})))\right)\right.\right.$$
$$\left.\left. - w_i(t, C_{i-1}(J_{i-1}), C_i)\varphi_{i-1}'(H_{i-1}(t, x, C_{i-1}(J_{i-1})))\tilde{G}_{i-1}^{\tilde{H}}(t, x, C_{i-1}(J_{i-1}))\right]^2\right]$$
$$\leq \mathbb{E}_{C_i}\mathbf{E}\left[\mathbb{E}_{J_{i-1}}\left[w_i(t, C_{i-1}(J_{i-1}), C_i)\left(\varphi_{i-1}'(h_{i-1}(t, x, C_{i-1}(J_{i-1}))) - \varphi_{i-1}'(H_{i-1}(t, x, C_{i-1}(J_{i-1})))\right)\tilde{G}_{i-1}^{\tilde{H}}(t, x, C_{i-1}(J_{i-1}))\right]^2\right]$$
$$\leq K\mathbb{E}_{C_i}\mathbf{E}\left[\mathbb{E}_{J_{i-1}}\left[|w_i(t, C_{i-1}(J_{i-1}), C_i)|\,|h_{i-1}(t, x, C_{i-1}(J_{i-1})) - H_{i-1}(t, x, C_{i-1}(J_{i-1}))|\left|\tilde{G}_{i-1}^{\tilde{H}}(t, x, C_{i-1}(J_{i-1}))\right|\right]^2\right]$$
$$\leq K\mathbb{E}_{C_i}\mathbf{E}\left[\mathbb{E}_{J_{i-1}}\left[|w_i(t, C_{i-1}(J_{i-1}), C_i)|\left(\tilde{G}_{i-1}^{\tilde{H}}(t, x, C_{i-1}(J_{i-1}))\right)^2\right]^2\right]/N$$
$$\leq K_T/N,$$

by Lemmas 11 and 13. Thus, it remains to show that

$$\mathbb{E}_{C_i}\mathbf{E}\left[\left(\mathbb{E}_{J_{i-1}}\left[w_i(t, C_{i-1}(J_{i-1}), C_i)\varphi_{i-1}'(H_{i-1}(t, x, C_{i-1}(J_{i-1})))\tilde{G}_{i-1}^{\tilde{H}}(t, x, C_{i-1}(J_{i-1}))\right]\right.\right.$$
$$\left.\left. - \mathbb{E}_{C_{i-1}}\left[w_i(t, C_{i-1}, C_i)\varphi_{i-1}'(H_{i-1}(t, x, C_{i-1}))\tilde{G}_{i-1}^{\tilde{H}}(t, x, C_{i-1})\right]\right)^2\right] = O_T(1/N).$$

This is shown by applying Lemma 12, where we take $\mathbf{i} = i - 1$, $\mathbf{i}' = i$, $S' = \{C_k(j_k) : j_k \in [N_k], k < i - 1\}$ and
$$f(S, c_{i-1}, c_i) = w_i(t, c_{i-1}, c_i)\varphi_{i-1}'(H_{i-1}(t, x, c_{i-1}))\tilde{G}_{i-1}^{\tilde{H}}(t, x, c_{i-1}).$$
Observe that $f(S, c_{i-1}, c_i)$ is independent of $C_{i-1}(j_{i-1})$. In addition:

$$\mathbf{E}\mathbb{E}_{C_i}\left[\left(f\left(S^{j_{i-1}}, C_{i-1}(j_{i-1}), C_i\right)\right)^2\right] \leq K\mathbf{E}\mathbb{E}_{C_i}\left[\left(w_i(t, C_{i-1}(j_{i-1}), C_i)\right)^4\right]^{1/2}\mathbf{E}\left[\left(\tilde{G}_{i-1}^{\tilde{H}}(t, x, C_{i-1}(j_{i-1}))\right)^4\right]^{1/2} \leq K_T$$

from Lemma 13. This shows the claim. $\qquad\square$

**Lemma 16.** *Under Assumption 1, we have for any $t \leq T$,*

$$\tilde{G}_L^{\partial\tilde{H}}(t, c_L, x) = \tilde{G}_L^{\tilde{H}}(t, x, c_L)\,\varphi_L''(H_L(t, x, c_L)) + \mathbf{O}_T(N^{-1/4}),$$
$$\tilde{G}_{i-1}^{\partial\tilde{H}}(t, c_{i-1}, x) = \tilde{G}_{i-1}^{\partial H}(t, c_{i-1}, x) + \tilde{G}_{i-1}^{\tilde{H}}(t, c_{i-1}, x)\varphi_{i-1}''(H_{i-1}(t, x, c_{i-1}))\mathbb{E}_{C_i}\left[w_i(t, c_{i-1}, C_i)\frac{\partial\hat{y}(t, x)}{\partial H_i(C_i)}\right]$$
$$\quad + \varphi_{i-1}'(H_{i-1}(t, x, c_{i-1}))\mathbb{E}_{C_i}\left[w_i(t, c_{i-1}, C_i)\tilde{G}_i^{\partial\tilde{H}}(t, C_i, x)\right] + \mathbf{O}_T(N^{-1/4}).$$

*Proof.* We have
$$\tilde{G}_{i-1}^{\partial\tilde{H}}(t, c_{i-1}, x)$$

$$= \sqrt{N}\left(\frac{\partial \tilde{y}(t,x)}{\partial \tilde{H}_{i-1}(c_{i-1})} - \frac{\partial \hat{y}(t,x)}{\partial H_{i-1}(c_{i-1})}\right)$$

$$= \sqrt{N}\left(\mathbb{E}_{J_i}\left[w_i(t,c_{i-1},C_i(J_i))\varphi'_{i-1}(\tilde{H}_{i-1}(t,x,c_{i-1}))\frac{\partial \tilde{y}(t,x)}{\partial \tilde{H}_i(C_i(J_i))}\right] - \mathbb{E}_{C_i}\left[w_i(t,c_{i-1},C_i)\varphi'_{i-1}(H_{i-1}(t,x,c_{i-1}))\frac{\partial \hat{y}(t,x)}{\partial H_i(C_i)}\right]\right)$$

$$= \tilde{G}^{\partial H}_{i-1}(t,c_{i-1},x)$$

$$+ \sqrt{N}\left(\varphi'_{i-1}(\tilde{H}_{i-1}(t,x,c_{i-1})) - \varphi'_{i-1}(H_{i-1}(t,x,c_{i-1}))\right)\mathbb{E}_{J_i}\left[w_i(t,c_{i-1},C_i(J_i))\frac{\partial \hat{y}(t,x)}{\partial H_i(C_i(J_i))}\right]$$

$$+ \varphi'_{i-1}(H_{i-1}(t,x,c_{i-1}))\mathbb{E}_{J_i}\left[w_i(t,c_{i-1},C_i(J_i))\tilde{G}^{\partial \tilde{H}}_i(t,C_i(J_i),x)\right]$$

$$+ \left(\varphi'_{i-1}(\tilde{H}_{i-1}(t,x,c_{i-1})) - \varphi'_{i-1}(H_{i-1}(t,x,c_{i-1}))\right)\mathbb{E}_{J_i}\left[w_i(t,c_{i-1},C_i(J_i))\tilde{G}^{\partial \tilde{H}}_i(t,C_i(J_i),x)\right].$$

We proceed with analyzing each term.

**First estimate.** Firstly we claim that

$$\mathbf{E}\mathbb{E}_{C_{i-1}}\left[\left(\sqrt{N}\left(\varphi'_{i-1}(\tilde{H}_{i-1}(t,x,C_{i-1})) - \varphi'_{i-1}(H_{i-1}(t,x,C_{i-1}))\right)\mathbb{E}_{J_i}\left[w_i(t,C_{i-1},C_i(J_i))\frac{\partial \hat{y}(t,x)}{\partial H_i(C_i(J_i))}\right]\right.\right.$$

$$\left.\left. - \sqrt{N}\left(\varphi'_{i-1}(\tilde{H}_{i-1}(t,x,C_{i-1})) - \varphi'_{i-1}(H_{i-1}(t,x,C_{i-1}))\right)\mathbb{E}_{C_i}\left[w_i(t,C_{i-1},C_i)\frac{\partial \hat{y}(t,x)}{\partial H_i(C_i)}\right]\right)^2\right]$$

$$= O_T(1/N).$$

Indeed, this is obvious for $i = L$, and for $i < L$, we verify the condition of Lemma 12 for $\mathbf{i} = i$, $\mathbf{i}' = i-1$, $S' = \{C_k(j_k): j_k \in [N_k],\ k < i\}$ and

$$f(S,c_i,c_{i-1}) = \sqrt{N}\left(\varphi'_{i-1}(\tilde{H}_{i-1}(t,x,c_{i-1})) - \varphi'_{i-1}(H_{i-1}(t,x,c_{i-1}))\right)w_i(t,c_{i-1},c_i)\frac{\partial \hat{y}(t,x)}{\partial H_i(c_i)}.$$

Observe that $f$ is independent of $C_i(j_i)$. In this case, we only need to bound the following:

$$\mathbf{E}\mathbb{E}_{C_{i-1},C_i}\left[\left(\sqrt{N}\left(\varphi'_{i-1}(\tilde{H}_{i-1}(t,x,C_{i-1})) - \varphi'_{i-1}(H_{i-1}(t,x,C_{i-1}))\right)w_i(t,C_{i-1},C_i)\frac{\partial \hat{y}(t,x)}{\partial H_i(C_i)}\right)^2\right]$$

$$\leq K\mathbf{E}\mathbb{E}_{C_{i-1}}\left[\left(\sqrt{N}(\tilde{H}_{i-1}(t,x,C_{i-1}) - H_{i-1}(t,x,C_{i-1}))\right)^4\right] + K\mathbf{E}\mathbb{E}_{C_{i-1},C_i}\left[\left(w_i(t,C_{i-1},C_i)\frac{\partial \hat{y}(t,x)}{\partial H_i(C_i)}\right)^4\right]$$

$$\leq K_T,$$

by Lemmas 11 and 13. This proves the claim.

Next, we extend this claim. In particular,

$$\mathbf{E}\mathbb{E}_{C_{i-1},C_i}\left[\left(\sqrt{N}\left(\varphi'_{i-1}(\tilde{H}_{i-1}(t,x,C_{i-1})) - \varphi'_{i-1}(H_{i-1}(t,x,C_{i-1}))\right)\mathbb{E}_{C_i}\left[w_i(t,C_{i-1},C_i)\frac{\partial \hat{y}(t,x)}{\partial H_i(C_i)}\right]\right.\right.$$

$$\left.\left. - \varphi''_{i-1}(H_{i-1}(t,x,C_{i-1}))\tilde{G}^{\tilde{H}}_{i-1}(t,x,C_{i-1})\mathbb{E}_{C_i}\left[w_i(t,C_{i-1},C_i)\frac{\partial \hat{y}(t,x)}{\partial H_i(C_i)}\right]\right)^2\right]$$

$$\leq \frac{K}{N}\mathbf{E}\mathbb{E}_{C_{i-1},C_i}\left[\left(\left(\tilde{G}^{\tilde{H}}_{i-1}(t,x,C_{i-1})\right)^2\mathbb{E}_{C_i}\left[w_i(t,C_{i-1},C_i)\frac{\partial \hat{y}(t,x)}{\partial H_i(C_i)}\right]\right)^2\right]$$

$$\leq \frac{K_T}{N},$$

by Lemmas 11 and 13. Therefore,

$$\mathbf{E}\mathbb{E}_{C_{i-1}}\left[\left(\sqrt{N}\left(\varphi'_{i-1}(\tilde{H}_{i-1}(t,x,C_{i-1})) - \varphi'_{i-1}(H_{i-1}(t,x,C_{i-1}))\right)\mathbb{E}_{J_i}\left[w_i(t,C_{i-1},C_i(J_i))\frac{\partial \hat{y}(t,x)}{\partial H_i(C_i(J_i))}\right]\right.\right.$$

$$\left.\left. - \varphi''_{i-1}(H_{i-1}(t,x,C_{i-1}))\tilde{G}^{\tilde{H}}_{i-1}(t,x,C_{i-1})\mathbb{E}_{C_i}\left[w_i(t,C_{i-1},C_i)\frac{\partial \hat{y}(t,x)}{\partial H_i(C_i)}\right]\right)^2\right]$$

$$= O_T(1/N).$$

**Second estimate.** We have:

$$\mathbf{E}\mathbb{E}_{C_{i-1}}\left[\left(\varphi'_{i-1}(H_{i-1}(t,x,C_{i-1}))\mathbb{E}_{J_i}\left[w_i(t,C_{i-1},C_i(J_i))\tilde{G}_i^{\partial\tilde{H}}(t,C_i(J_i),x)\right]\right.\right.$$

$$\left.\left.-\varphi'_{i-1}(H_{i-1}(t,x,C_{i-1}))\mathbb{E}_{C_i}\left[w_i(t,C_{i-1},C_i)\tilde{G}_i^{\partial\tilde{H}}(t,C_i,x)\right]\right)^2\right]$$

$$=O(N^{-1/2}).$$

This is again obvious for $i=L$. For $i<L$, by applying Lemma 12 for $\mathbf{i}=i$, $\mathbf{i}'=i-1$ and

$$f(S,c_i,c_{i-1})=\varphi'_{i-1}(H_{i-1}(t,x,c_{i-1}))w_i(t,c_{i-1},c_i)\tilde{G}_i^{\partial\tilde{H}}(t,c_i,x),$$

we have the claim since firstly it is easy to see that by Lemma 11,

$$\mathbf{E}\mathbb{E}_{C_{i-1}}\left[\left(f\left(S,C_i\left(j_i\right),C_{i-1}\right)-f\left(S^{j_i},C_i\left(j_i\right),C_{i-1}\right)\right)^2\right]=O_T\left(1/N\right),$$

and secondly

$$\mathbf{E}\mathbb{E}_{C_{i-1},C_i}\left[\left(\varphi'_{i-1}(\tilde{H}_{i-1}(t,x,C_{i-1}))w_i(t,C_{i-1},C_i)\tilde{G}_i^{\partial\tilde{H}}(t,C_i,x)\right)^2\right]$$

$$\leq K\mathbf{E}\mathbb{E}_{C_{i-1},C_i}\left[(w_i(t,C_{i-1},C_i))^4\right]+K\mathbf{E}\mathbb{E}_{C_i}\left[\left(\tilde{G}_i^{\partial\tilde{H}}(t,C_i,x)\right)^4\right]$$

$$\leq K_T.$$

by Lemmas 11 and 14.

**Last estimate.** Finally we have:

$$\mathbf{E}\mathbb{E}_{C_{i-1}}\left[\left(\varphi'_{i-1}(\tilde{H}_{i-1}(t,x,C_{i-1}))-\varphi'_{i-1}(H_{i-1}(t,x,C_{i-1}))\right)^2\mathbb{E}_{J_i}\left[w_i(t,C_{i-1},C_i(J_i))\tilde{G}_i^{\partial\tilde{H}}(t,C_i(J_i),x)\right]^2\right]$$

$$\leq K\mathbf{E}\mathbb{E}_{C_{i-1}}\left[\left(\tilde{G}_{i-1}^{\tilde{H}}(t,x,C_{i-1})\right)^2\mathbb{E}_{J_i}\left[w_i(t,C_{i-1},C_i(J_i))\tilde{G}_i^{\partial\tilde{H}}(t,C_i(J_i),x)\right]^2\right]/N$$

$$\leq K_T/N.$$

by Lemmas 11, 13 and 14.

**Combining estimates.** Combining all above estimates, we obtain

$$\tilde{G}_{i-1}^{\partial\tilde{H}}(t,c_{i-1},x)=\tilde{G}_{i-1}^{\partial H}(t,c_{i-1},x)+\tilde{G}_{i-1}^{\tilde{H}}(t,c_{i-1},x)\varphi''_{i-1}(H_{i-1}(t,x,c_{i-1}))\mathbb{E}_{C_i}\left[w_i(t,c_{i-1},C_i)\frac{\partial\hat{y}(t,x)}{\partial H_i(C_i)}\right]$$

$$+\varphi'_{i-1}(H_{i-1}(t,x,c_{i-1}))\mathbb{E}_{C_i}\left[w_i(t,c_{i-1},C_i)\tilde{G}_i^{\partial\tilde{H}}(t,C_i,x)\right]+\mathbf{O}_T(N^{-1/4}).$$

Note that the claim for $\tilde{G}_L^{\partial\tilde{H}}$ follows by the same argument. $\qquad\square$

**Lemma 17.** *Under Assumption 1, we have for any $t\leq T$,*

$$\tilde{G}_1^w(t,1,c_1,x)=\tilde{G}_1^{\partial\tilde{H}}(t,c_1,x)x+\mathbf{O}_T(N^{-1/2}),$$

$$\tilde{G}_i^w(t,c_{i-1},c_i,x)=\tilde{G}_i^{\partial\tilde{H}}(t,c_i,x)\varphi_{i-1}(H_{i-1}(t,x,c_{i-1}))+\varphi'_{i-1}(H_{i-1}(t,x,c_{i-1}))\frac{\partial\hat{y}(t,x)}{\partial H_i(c_i)}\tilde{G}_{i-1}^{\tilde{H}}(t,c_{i-1},x)+\mathbf{O}_T(N^{-1/2}),$$

$$\tilde{G}^y(t,x)=\varphi'_L(H_L(t,1,x))\tilde{G}_L^{\tilde{H}}(t,1,x)+\mathbf{O}_T(N^{-1/2}).$$

*Proof.* We prove the second statement; the claims for $\tilde{G}_1^w$ and $\tilde{G}^y$ can be proven similarly. We have:

$$\tilde{G}_i^w(t,c_{i-1},c_i,x)$$

$$=\sqrt{N}\left(\frac{\partial\tilde{y}(t,x)}{\partial w_i(c_{i-1},c_i)}-\frac{\partial\hat{y}(t,x)}{\partial w_i(c_{i-1},c_i)}\right)$$

$$= \sqrt{N} \left( \frac{\partial \tilde{y}(t,x)}{\partial \tilde{H}_i(c_i)} - \frac{\partial \hat{y}(t,x)}{\partial H_i(c_i)} \right) \varphi_{i-1}(H_{i-1}(t,x,c_{i-1})) + \sqrt{N} \left( \varphi_{i-1}(\tilde{H}_{i-1}(t,x,c_{i-1})) - \varphi_{i-1}(H_{i-1}(t,x,c_{i-1})) \right) \frac{\partial \hat{y}(t,x)}{\partial H_i(c_i)}$$

$$+ \sqrt{N} \left( \frac{\partial \tilde{y}(t,x)}{\partial \tilde{H}_i(c_i)} - \frac{\partial \hat{y}(t,x)}{\partial H_i(c_i)} \right) \left( \varphi_{i-1}(\tilde{H}_{i-1}(t,x,c_{i-1})) - \varphi_{i-1}(H_{i-1}(t,x,c_{i-1})) \right)$$

$$= \tilde{G}_i^{\partial \tilde{H}}(t,c_i,x) \varphi_{i-1}(H_{i-1}(t,x,c_{i-1})) + \varphi_{i-1}'(H_{i-1}(t,x,c_{i-1})) \frac{\partial \hat{y}(t,x)}{\partial H_i(c_i)} \tilde{G}_{i-1}^{\tilde{H}}(t,c_{i-1},x) + \mathbf{O}_T(N^{-1/2}).$$

Here, we have used that by Lemmas 11 and 13,

$$\mathbf{E}\mathbb{E}_{C_{i-1},C_i} \left[ \left( \sqrt{N} \left( \varphi_{i-1}(\tilde{H}_{i-1}(t,x,C_{i-1})) - \varphi_{i-1}(H_{i-1}(t,x,C_{i-1})) \right) \frac{\partial \hat{y}(t,x)}{\partial H_i(C_i)} \right. \right.$$

$$\left. \left. - \varphi_{i-1}'(H_{i-1}(t,x,C_{i-1})) \frac{\partial \hat{y}(t,x)}{\partial H_i(C_i)} \tilde{G}_{i-1}^{\tilde{H}}(t,C_{i-1},x) \right)^2 \right]$$

$$\leq \frac{K}{N} \mathbf{E}\mathbb{E}_{C_{i-1},C_i} \left[ \left( \tilde{G}_{i-1}^{\tilde{H}}(t,C_{i-1},x) \frac{\partial \hat{y}(t,x)}{\partial H_i(C_i)} \right)^2 \right]$$

$$\leq \frac{K_T}{N},$$

and that by Lemmas 13 and 14,

$$\mathbf{E}\mathbb{E}_{C_{i-1},C_i} \left[ \left( \sqrt{N} \left( \frac{\partial \tilde{y}(t,x)}{\partial \tilde{H}_i(C_i)} - \frac{\partial \hat{y}(t,x)}{\partial H_i(C_i)} \right) \left( \varphi_{i-1}(\tilde{H}_{i-1}(t,x,C_{i-1})) - \varphi_{i-1}(H_{i-1}(t,x,C_{i-1})) \right) \right)^2 \right]$$

$$\leq \frac{K}{N} \mathbf{E}\mathbb{E}_{C_{i-1},C_i} \left[ \left( \tilde{G}_{i-1}^{\tilde{H}}(t,C_{i-1},x) \tilde{G}_i^{\partial \tilde{H}}(t,x,C_i) \right)^2 \right]$$

$$\leq \frac{K_T}{N}.$$

$\square$

## B.3 Structure of the limiting Gaussian process and the proof of Theorem 2

Each process $\tilde{G}_i^{\partial H}$, $\tilde{G}_i^H$, $\tilde{G}_i^{\partial \tilde{H}}$, $\tilde{G}_i^{\tilde{H}}$, $\tilde{G}_i^w$, $\tilde{G}^y$ on $t \leq T$ can be viewed as an element of $\mathcal{G}_T = C([0,T], L^2(\mathbb{X} \times \tilde{P}))$ where $\tilde{P}$ is $P_i$ or $P_{i-1} \times P_i$. Equip $\mathcal{G}_T$ with the norm

$$\|g\|_T = \sup_{t \in [0,T]} \mathbb{E} \left[ \|g(t)\|^2 \right]^{1/2}.$$

We define the following Gaussian processes $\underline{G}^{\partial H}$, $\underline{G}^H$ with zero mean and covariance given by

$$\mathbf{E} \left[ \underline{G}_i^H(t,c_i,x) \underline{G}_j^H(t',c_j',x') \right]$$
$$= \begin{cases} \delta_{ij} \mathbb{C}_{C_{i-1}} \left[ w_i(t,C_{i-1},c_i) \varphi_{i-1}(H_{i-1}(t,C_{i-1},x)); \; w_j(t',C_{i-1},c_j') \varphi_{i-1}(H_{i-1}(t',C_{i-1},x')) \right], & i > 1, \\ 0, & \text{otherwise,} \end{cases}$$

$$\mathbf{E} \left[ \underline{G}_i^{\partial H}(t,c_i,x) \underline{G}_j^{\partial H}(t',c_j',x') \right]$$
$$= \begin{cases} \delta_{ij} \mathbb{C}_{C_{i+1}} \left[ \varphi_i'(H_i(t,x,c_i)) w_i(t,c_i,C_{i+1}) \frac{\partial \hat{y}(t,x)}{\partial H_{i+1}(C_{i+1})}; \; \varphi_i'(H_i(t',x',c_j')) w_i(t',c_j',C_{i+1}) \frac{\partial \hat{y}(t',x')}{\partial H_{i+1}(C_{i+1})} \right], & i < L, \\ 0, & \text{otherwise,} \end{cases}$$

$$\mathbf{E} \left[ \underline{G}_i^H(t,c_i,x) \underline{G}_j^{\partial H}(t',c_j',x') \right]$$
$$= \begin{cases} \delta_{i,j+2} \mathbb{C}_{C_{i-1}} \left[ \varphi_{i-2}'(H_{i-2}(t,x,c_j')) w_{i-1}(t,c_j',C_{i-1}) \frac{\partial \hat{y}(t,x)}{\partial H_{i-1}(C_{i-1})}; \; w_i(t,C_{i-1},c_i) \varphi_{i-1}(H_{i-1}(t,x,C_{i-1})) \right], & i > 2, \\ 0, & \text{otherwise.} \end{cases}$$

As a side remark, note that the covariance of $\underline{G}^{\partial H}$ and $\underline{G}^H$ can be recognized through the following simple rule: for functions $f(C_i)$ and $g(C_j)$, we have

$$\mathbf{E} \left[ \sqrt{N}(\mathbb{E}_{J_i}[f(C_i(J_i))] - \mathbb{E}_{C_i}[f(C_i)]) \cdot \sqrt{N}(\mathbb{E}_{J_j}[g(C_j(J_j))] - \mathbb{E}_{C_j}[g(C_j)]) \right] = \delta_{ij}(\mathbb{E}_{C_i}[f(C_i)g(C_i)] - \mathbb{E}_{C_i}[f(C_i)]\mathbb{E}_{C_i}[g(C_i)]). \quad (7)$$

Define $\underline{G}^{\tilde{H}}, \underline{G}^{\partial\tilde{H}}, \underline{G}^w, \underline{G}^y$ by the following linear combination of $\underline{G}^H$ and $\underline{G}^{\partial H}$:

$$\underline{G}_1^{\tilde{H}}(t, c_1, x) = 0,$$

$$\underline{G}_i^{\tilde{H}}(t, c_i, x) = \underline{G}_i^H(t, c_i, x) + \mathbb{E}_{C_{i-1}}\left[w_i(t, C_{i-1}, c_i)\varphi_{i-1}'(H_{i-1}(t, x, C_{i-1}))\underline{G}_{i-1}^{\tilde{H}}(t, x, C_{i-1})\right],$$

$$\underline{G}_L^{\partial\tilde{H}}(t, c_L, x) = \varphi_L''(H_L(t, x, c_L))\underline{G}_L^{\tilde{H}}(t, x, c_L),$$

$$\underline{G}_{i-1}^{\partial\tilde{H}}(t, c_{i-1}, x) = \underline{G}_{i-1}^{\partial H}(t, c_{i-1}, x) + \underline{G}_{i-1}^{\tilde{H}}(t, c_{i-1}, x)\varphi_{i-1}''(H_{i-1}(t, x, c_{i-1}))\mathbb{E}_{C_i}\left[w_i(t, c_{i-1}, C_i)\frac{\partial\hat{y}(t, x)}{\partial H_i(C_i)}\right]$$

$$+ \varphi_{i-1}'(H_{i-1}(t, x, c_{i-1}))\mathbb{E}_{C_i}\left[w_i(t, c_{i-1}, C_i(J_i))\underline{G}_i^{\partial\tilde{H}}(t, C_i(J_i), x)\right],$$

$$\underline{G}_1^w(t, 1, c_1, x) = \underline{G}_1^{\partial\tilde{H}}(t, c_1, x)\, x,$$

$$\underline{G}_i^w(t, c_{i-1}, c_i, x) = \underline{G}_i^{\partial\tilde{H}}(t, c_i, x)\varphi_{i-1}(H_{i-1}(t, x, c_{i-1})) + \varphi_{i-1}'(H_{i-1}(t, x, c_{i-1}))\frac{\partial\hat{y}(t, x)}{\partial H_i(c_i)}\underline{G}_{i-1}^{\tilde{H}}(t, c_{i-1}, x),$$

$$\underline{G}^y(t, x) = \varphi_L'(H_L(t, x, 1))\underline{G}_L^{\tilde{H}}(t, x, 1).$$

Recall that we say $\tilde{G}$ converges $G$-polynomially in moment to a Gaussian process $\underline{G}$ if for any square-integrable $f_j : \mathbb{T} \times \Omega_{i(j)-1} \times \Omega_{i(j)} \times \mathbb{X} \to \mathbb{R}$ which is continuous in time,

$$\sup_{t_j \le T}\left|\mathbf{E}\left[\prod_j\langle f_j, \tilde{G}_{i(j)}^{\alpha_j}\rangle_{t_j}^{\beta_j}\right] - \mathbf{E}\left[\prod_j\langle f_j, \underline{G}_{i(j)}^{\alpha_j}\rangle_{t_j}^{\beta_j}\right]\right| = O_{D,T}\left(\sup_{t\le T}\max_j\|f_j\|_t^D\right)\cdot N^{-1/8},$$

where $D = \sum_j \alpha_j\beta_j$. (For $\tilde{G}^H, \tilde{G}^{\tilde{H}}, \tilde{G}^{\partial H}$ and $\tilde{G}^{\partial\tilde{H}}$, we take $f_j : \mathbb{T} \times \Omega_{i(j)} \times \mathbb{X} \to \mathbb{R}$.) We restate and prove Theorem 2.

**Theorem 18** (Theorem 2 restated). *We have $\tilde{G}$ converges $G$-polynomially in moment to $\underline{G}$, where $\underline{G}$ is the Gaussian process with mean and covariance structure as given above.*

*Proof.* First, we show the following:

$$\left|\mathbf{E}\left[\prod_\ell\left\langle f_\ell, \left(\tilde{G}_{i(\ell)}^{\tau(\ell)}\right)^{\alpha_\ell}\right\rangle_{t_\ell}^{\beta_\ell}\right] - \mathbf{E}\left[\prod_\ell\left\langle f_\ell, \left(\underline{G}_{i(\ell)}^{\tau(\ell)}\right)^{\alpha_\ell}\right\rangle_{t_\ell}^{\beta_\ell}\right]\right| = O_D(\max_\ell\|f_\ell\|_{t_\ell}^D)\cdot N^{-1/2},$$

where for each $\ell$, $\tau(\ell)$ is either $H$ or $\partial H$. By independence of $C_i(j_i)$ for distinct $i$'s, it suffices to consider the case where $i(\ell) = i$ for all $\ell$.

Consider the case $\tau(\ell) = H$ and $i(\ell) = i$, we have

$$\left\langle f_\ell, \left(\tilde{G}_i^H\right)^{\alpha_\ell}\right\rangle_{t_\ell} = \mathbb{E}_{Z,C_i}\left[f_\ell(t_\ell, C_i, X)\left(\sqrt{N}\mathbb{E}_{J_{i-1}}[Z_i(t_\ell, J_{i-1}, C_i, X)]\right)^{\alpha_\ell}\right],$$

where we denote

$$Z_i(t_\ell, j_{i-1}, c_i, x) = w_i(t_\ell, C_{i-1}(j_{i-1}), c_i)\varphi_{i-1}(H_{i-1}(t_\ell, x, C_{i-1}(j_{i-1}))) - \mathbb{E}_{C_{i-1}}[w_i(t_\ell, C_{i-1}, c_i)\varphi_{i-1}(H_{i-1}(t_\ell, x, C_{i-1}))].$$

One can write $\prod_\ell\left\langle f_\ell, \left(\tilde{G}_i^H\right)^{\alpha_\ell}\right\rangle_{t_\ell}^{\beta_\ell}$ as a sum of terms of the form:

$$N^{-D/2}\prod_\ell\prod_{r\le\beta_\ell}\mathbb{E}_{Z,C_i}\left[f_\ell(t_\ell, C_i, X)\prod_{h\le\alpha_\ell}Z_i(t_\ell, j_{\ell,r,h}, C_i, X)\right].$$

Note that $\mathbf{E}\mathbb{E}_{J_{i-1}}[Z_i(\cdot, J_{i-1}, C_i, \cdot)] = 0$. Thus, if there is any index $j_{i-1}$ for which $j_{i-1}$ appears exactly once among $j_{\ell,r,h}$, then the above term after taking $\mathbf{E}$ vanishes. The number of terms with each $j_{i-1}$ appearing either zero or exactly twice among $j_{\ell,r,h}$ is $\Theta(N^{D/2})$. Notice that these terms make up the quantity $\mathbf{E}\left[\prod_\ell\left\langle f_\ell, \left(\underline{G}_{i(\ell)}^{\tau(\ell)}\right)^{\alpha_\ell}\right\rangle_{t_\ell}^{\beta_\ell}\right]$, which is to be subtracted away. The number of remaining terms where no $j_{i-1}$ appears exactly once is at most $O(N^{(D-1)/2})$. Thus we are done upon bounding

$$\mathbf{E}\left[\left|\prod_\ell\prod_{r\le\beta_\ell}\mathbb{E}_{Z,C_i}\left[f_\ell(t_\ell, C_i, X)\prod_{h\le\alpha_\ell}Z_i(t_\ell, j_{\ell,r,h}, C_i, X)\right]\right|\right].$$

By Lemma 11, it is easy to see that

$$\mathbf{E}\mathbb{E}_{X,C_i}\left[|Z_i(t_\ell, j_{\ell,r,h}, C_i, X)|^D\right] \leq K_{T,D}.$$

Then:

$$\mathbf{E}\left[\left|\prod_\ell \prod_{r\leq\beta_\ell} \mathbb{E}_{Z,C_i}\left[f_\ell(t_\ell, C_i, X)\prod_{h\leq\alpha_\ell} Z_i(t_\ell, j_{\ell,r,h}, C_i, X)\right]\right|\right]$$

$$\leq \prod_\ell \prod_{r\leq\beta_\ell} \mathbb{E}_{Z,C_i}\left[\left|f_\ell(t_\ell, C_i, X)\right|^2\right]^{1/2} \mathbf{E}\left[\mathbb{E}_{Z,C_i}\left[\prod_{h\leq\alpha_\ell}\left|Z_i(t_\ell, j_{\ell,r,h}, C_i, X)\right|^2\right]^{1/2}\right]$$

$$\leq \max_\ell \|f_\ell\|_{t_\ell}^D \prod_\ell \prod_{r\leq\beta_\ell} \mathbf{E}\left[\mathbb{E}_{Z,C_i}\left[\prod_{h\leq\alpha_\ell}\left|Z_i(t_\ell, j_{\ell,r,h}, C_i, X)\right|^2\right]^{1/2}\right]$$

$$\leq \max_\ell \|f_\ell\|_{t_\ell}^D \prod_\ell \prod_{r\leq\beta_\ell} \mathbf{E}\left[\prod_{h\leq\alpha_\ell} \mathbb{E}_{Z,C_i}\left[\left|Z_i(t_\ell, j_{\ell,r,h}, C_i, X)\right|^{2\alpha_\ell}\right]^{1/(2\alpha_\ell)}\right]$$

$$\leq \max_\ell \|f_\ell\|_{t_\ell}^D \prod_\ell \prod_{r\leq\beta_\ell} \prod_{h\leq\alpha_\ell} \mathbf{E}\mathbb{E}_{Z,C_i}\left[\left|Z_i(t_\ell, j_{\ell,r,h}, C_i, X)\right|^{2\alpha_\ell}\right]^{1/(2\alpha_\ell)}$$

$$\leq \max_\ell \|f_\ell\|_{t_\ell}^D \cdot K_{T,D}.$$

This completes the proof for the case $\tau(\ell) = H$ and $i(\ell) = i$.

The involvement of terms with $\tau(\ell) = \partial H$ can be dealt with similarly. To deal with the case $\tau(\ell) \in \{H, \partial H\}$, we note that the same argument holds with an appropriate modification of the definition of $\tilde{Z}_i$; in particular, this function now depends on $\ell$ via $\tau(\ell)$, but this detail does not affect the argument since all we need is that $\mathbf{E}\mathbb{E}_{J_{i-1}}[Z_i(\cdot, J_{i-1}, C_i, \cdot)] = 0$ and $\mathbf{E}\mathbb{E}_{X,C_i}\left[|Z_i(t, \cdot, C_i, X)|^D\right] \leq K_{T,D}$. This completes the argument that $(\tilde{G}^{\partial H}, \tilde{G}^H) \to (\underline{G}^{\partial H}, \underline{G}^H)$ in $G$-polynomial moment.

Finally, observe that $\tilde{G}_i^{\tilde{H}}$, $\tilde{G}_i^{\partial\tilde{H}}$, $\tilde{G}_i^w$, $\tilde{G}^y$ can be written as a linear combination of $\tilde{G}_i^H$ and $\tilde{G}_i^{\partial H}$ by Lemmas 15, 16 and 17, from which it is easy to verify that $\tilde{G} \to \underline{G}$ in $G$-polynomial moment. □

## B.4 Dynamical form of the limiting Gaussian process

We also have the following property of the limiting process $\underline{G}$ that will be useful later. Define further auxiliary processes:

$$\tilde{G}_{i-1}^{\partial_t(\partial H)}(t, c_{i-1}, x)$$
$$= \sqrt{N}\left(\mathbb{E}_{J_i}\left[w_i(t, c_{i-1}, C_i(J_i))\frac{\partial\hat{y}(t,x)}{\partial H_i(C_i(J_i))}\right] - \mathbb{E}_{C_i}\left[w_i(t, c_{i-1}, C_i)\frac{\partial\hat{y}(t,x)}{\partial H_i(C_i)}\right]\right)\varphi_{i-1}''(H_{i-1}(t,x,c_{i-1}))\partial_t H_{i-1}(t,x,c_{i-1}),$$
$$+ \sqrt{N}\left(\mathbb{E}_{J_i}\left[\partial_t w_i(t, c_{i-1}, C_i(J_i))\frac{\partial\hat{y}(t,x)}{\partial H_i(C_i(J_i))}\right] - \mathbb{E}_{C_i}\left[\partial_t w_i(t, c_{i-1}, C_i)\frac{\partial\hat{y}(t,x)}{\partial H_i(C_i)}\right]\right)\varphi_{i-1}'(H_{i-1}(t,x,c_{i-1}))$$
$$+ \sqrt{N}\left(\mathbb{E}_{J_i}\left[\partial_t w_i(t, c_{i-1}, C_i(J_i))\partial_t\left(\frac{\partial\hat{y}(t,x)}{\partial H_i(C_i(J_i))}\right)\right] - \mathbb{E}_{C_i}\left[\partial_t w_i(t, c_{i-1}, C_i)\partial_t\left(\frac{\partial\hat{y}(t,x)}{\partial H_i(C_i)}\right)\right]\right)\varphi_{i-1}'(H_{i-1}(t,x,c_{i-1})),$$

and

$$\tilde{G}_i^{\partial_t H}(t, c_i, x)$$
$$= \sqrt{N}\left(\mathbb{E}_{J_{i-1}}[\partial_t w_i(t, C_{i-1}(J_{i-1}), c_i)\varphi_{i-1}(H_{i-1}(t,x,C_{i-1}(J_{i-1})))] - \mathbb{E}_{C_{i-1}}[\partial_t w_i(t, C_{i-1}, c_i)\varphi_{i-1}(H_{i-1}(t,x,C_{i-1}))]\right)$$
$$+ \sqrt{N}\Big(\mathbb{E}_{J_{i-1}}[w_i(t, C_{i-1}(J_{i-1}), c_i)\varphi_{i-1}'(H_{i-1}(t,x,C_{i-1}(J_{i-1})))\partial_t H_{i-1}(t,x,C_{i-1}(J_{i-1}))]$$
$$- \mathbb{E}_{C_{i-1}}[w_i(t, C_{i-1}, c_i)\varphi_{i-1}'(H_{i-1}(t,x,C_{i-1}))\partial_t H_{i-1}(t,x,C_{i-1})]\Big),$$

with $\tilde{G}_{L-1}^{\partial_t(\partial H)} = \tilde{G}_L^{\partial_t(\partial H)} = \tilde{G}_1^{\partial_t H} = 0$. As above, we can show that jointly with previously defined processes $\tilde{G}^H, \tilde{G}^{\partial H}, \tilde{G}^{\partial_t(\partial H)}, \tilde{G}^{\partial_t H}$ converges to Gaussian processes $\underline{G}^{\partial_t(\partial H)}, \underline{G}^{\partial_t H}$ whose covariance structure (jointly with the previously defined processes $\underline{G}$) can be deduced by following the rule (7).

**Theorem 19.** *The limiting Gaussian process $(\underline{G}^H, \underline{G}^{\partial H})$ is the solution to the ODE*

$$\partial_t\underline{G}_i^H(t, c_i, x) = \underline{G}_i^{\partial_t H}(t, c_i, x),$$

$$\partial_t \underline{G}_i^{\partial H}(t, c_i, x) = \underline{G}_i^{\partial_t(\partial H)}(t, c_i, x).$$

*Similarly we can express $\partial_t \underline{G}_i^{\tilde{H}}$ and $\partial_t \underline{G}_i^{\partial \tilde{H}}$ in terms of $\partial_t G_i^H, \partial_t G_i^{\partial H}$ using the expression of $G^{\tilde{H}}$ and $G^{\partial \tilde{H}}$ in terms of $G^H$ and $G^{\partial H}$. Finally, we have*

$$\partial_t \underline{G}^y(t, x) = \varphi'_L(H_L(t, 1, x))\partial_t \underline{G}_L^{\tilde{H}}(t, x) + \partial_t H_L(t, 1, x)\varphi''_L(H_L(t, 1, x))\underline{G}_L^{\tilde{H}}(t, x).$$

*Proof.* Defining $\overline{G}_i^H(t, c_i, x), \overline{G}_i^{\partial H}(t, c_i, x)$ as solutions to

$$\partial_t \overline{G}_i^H(t, c_i, x) = G_i^{\partial_t H}(t, c_i, x),$$
$$\partial_t \overline{G}_i^{\partial H}(t, c_i, x) = G_i^{\partial_t(\partial H)}(t, c_i, x),$$

it is straightforward to verify that $\overline{G}$ has the same covariance structure as $G$. Since $G^{\tilde{H}}, G^{\partial \tilde{H}}, G^w, G^y$ are linear combinations of $G^H, G^{\partial H}$, the conclusions for $\partial_t G$ easily follow. $\qquad\square$

*Remark* 20. We can in fact show that jointly with $\tilde{G}^H$ and $\tilde{G}^{\partial H}$, we have $(\partial_t \tilde{G}^H, \partial_t \tilde{G}^{\partial H})$ converges in $G$-polynomial moment to $(\partial_t G^H, \partial_t G^{\partial H})$, and this extends by linearity to $\partial_t \tilde{G}^{\tilde{H}}, \partial_t \tilde{G}^{\partial \tilde{H}}, \partial_t \tilde{G}^w, \partial_t \tilde{G}^y$, by following the procedure leading to Theorem 18. Since we will not need to use this fact, we omit the details of the proof.

## C  Existence and uniqueness of $R$: Proof of Theorem 3

For $G \in \mathcal{G}$ and $p \geq 1$, define

$$\|G\|_{T,2p}^{2p} = \sup_{t \leq T} \mathbb{E}_{Z,C}\left[|G^y(t, X)|^{2p} + \sum_{i=1}^{L} |G_i^w(t, C_{i-1}, C_i, X)|^{2p}\right].$$

**Theorem 21.** *Under Assumption 1, for any $\epsilon > 0$ and all $G \in \mathcal{G}$ with $\|G\|_{T,2+\epsilon} < \infty$, there exists a unique solution $R_i(G, \cdot, \cdot, \cdot) \in L^2(P_{i-1} \times P_i)$ which is continuous in time. Furthermore, for each $t \leq T$, $R(G, t)$ is a continuous linear functional in $G$ and $\|R(G)\|_{T,2} \leq K_{T,\epsilon}\|G\|_{T,2+\epsilon}$. In fact, for each $B$ sufficiently large in $T$, there exists a sequence in $B$ of processes $R^B$ which is a continuous linear functional in $G$ with*

$$\|R^B(G)\|_{T,2}^2 \leq \exp(K_T B)\|G\|_{T,2}^2$$

*for all $G$ with $\|G\|_{T,2} < \infty$, and for $G$ with $\|G\|_{T,2+\epsilon} < \infty$,*

$$\|R(G) - R^B(G)\|_{T,2}^2 \leq \|G\|_{T,2+\epsilon}^2 \exp(-K_T \epsilon B^2/(2+\epsilon)).$$

*Here we define*

$$\|R(G)\|_{T,2}^2 = \sup_{t \leq T} \mathbb{E}_C\left[\sum_{i=1}^{L} |R_i(G, t, C_{i-1}, C_i)|^2\right],$$

*and similar for $\|R(G) - R^B(G)\|_{T,2}^2$.*

The main technical difficulty in the following proof lies in the fact that the weights $w_i(t, c_{i-1}, c_i)$ could be unbounded, in which case the linear operator $R \mapsto \partial_t R$ is unbounded. Our proof of both existence and uniqueness follows from a delicate truncation scheme. This scheme requires careful treatment: there is no a priori bound on $R$ again due to the unboundedness problem, and as such, usual truncation argument does not work. We remark that this problem is unique to the multilayer structure; we do not encounter the same problem for shallow networks.

*Proof of Theorem 21.* Let us define:

$$\Delta F_i^{(1)}(R)(G, t, c_{i-1}, c_i) = \mathbb{E}_Z\left[\frac{\partial \hat{y}(t, X)}{\partial w_i(c_{i-1}, c_i)}\partial_2^2 \mathcal{L}(Y, \hat{y}(t, X))\sum_{r=1}^{L} \mathbb{E}_C\left[R_r(G, t, C_{r-1}, C_r)\frac{\partial \hat{y}(t, X)}{\partial w_r(C_{r-1}, C_r)}\right]\right]$$

$$+ \mathbb{E}_Z\left[\partial_2 \mathcal{L}(Y, \hat{y}(t, X))\mathbb{E}_C\left[\sum_{r=1}^{L} R_r(G, t, C_{r-1}, C_r)\frac{\partial^2 \hat{y}(t, X)}{\partial w_r(C_{r-1}, C_r)\partial w_i(c_{i-1}, c_i)}\right]\right]$$

$$+ \mathbb{E}_Z\left[\partial_2 \mathcal{L}(Y, \hat{y}(t, X))\mathbb{E}_C\left[R_{i-1}(G, t, C_{i-2}, c_{i-1})\frac{\partial^2 \hat{y}(t, X)}{\partial_* w_{i-1}(C_{i-2}, c_{i-1})\partial w_i(c_{i-1}, c_i)}\right]\right]$$

$$+ \mathbb{E}_Z \left[ \partial_2 \mathcal{L}\left(Y, \hat{y}\left(t, X\right)\right) \mathbb{E}_C \left[ R_i\left(G, t, C_{i-1}, c_i\right) \frac{\partial^2 \hat{y}\left(t, X\right)}{\partial_* w_i\left(C_{i-1}, c_i\right) \partial w_i\left(c_{i-1}, c_i\right)} \right] \right]$$

$$+ \mathbb{E}_Z \left[ \partial_2 \mathcal{L}\left(Y, \hat{y}\left(t, X\right)\right) \mathbb{E}_C \left[ R_{i+1}\left(G, t, c_i, C_{i+1}\right) \frac{\partial^2 \hat{y}\left(t, X\right)}{\partial_* w_{i+1}\left(c_i, C_{i+1}\right) \partial w_i\left(c_{i-1}, c_i\right)} \right] \right],$$

$$\Delta F_i^{(2)}(R)(G, t, c_{i-1}, c_i) = \mathbb{E}_Z \left[ \frac{\partial \hat{y}(t, X)}{\partial w_i(c_{i-1}, c_i)} \partial_2^2 \mathcal{L}\left(Y, \hat{y}\left(t, X\right)\right) G^y\left(t, X\right) \right]$$

$$+ \mathbb{E}_Z \left[ \partial_2 \mathcal{L}\left(Y, \hat{y}\left(t, X\right)\right) G_i^w\left(t, c_{i-1}, c_i, X\right) \right],$$

$$\Delta F_i(R)(G, t, c_{i-1}, c_i) = \Delta F_i^{(1)}(R)(G, t, c_{i-1}, c_i) + \Delta F_i^{(2)}(R)(G, t, c_{i-1}, c_i).$$

In this notation, $\partial_t R_i(G, t, c_{i-1}, c_i) = -\Delta F_i(R)(G, t, c_{i-1}, c_i)$. Let

$$|R|_{t,2p}^{2p} = \sum_{j=1}^{L} \mathbb{E}_C[|R_j(G, t, C_{j-1}, C_j)|^{2p}],$$

$$|R_{i+1}(c_i)|_{t,2p}^{2p} = \mathbb{E}_{C_{i+1}}[|R_{i+1}(G, t, c_i, C_{i+1})|^{2p}],$$

$$|R_i(c_i)|_{t,2p}^{2p} = \mathbb{E}_{C_{i-1}}[|R_i(G, t, C_{i-1}, c_i)|^{2p}].$$

We define $|w_{i+1}(c_i)|_{t,2p}$, $|w_i(c_i)|_{t,2p}$, $|G_i^w(c_i)|_{t,2p}$, $|G_i^w(c_{i-1})|_{t,2p}$, $|G_i^y|_{t,2p}$ similarly (where the last three should include $\mathbb{E}_Z$ in addition). We define

$$|G_i^w|_{t,2p}^{2p} = \mathbb{E}_C \left[ |G_i^w(C_i)|_{t,2p}^{2p} + |G_i^w(C_{i-1})|_{t,2p}^{2p} \right], \quad |G|_{t,2p}^{2p} = |G^y|_{t,2p}^{2p} + \sum_{i=1}^{L} |G_i^w|_{t,2p}^{2p}.$$

When we drop the subscript $2p$, we implicitly assume that $2p = 2$.

**Step 1: truncated process $R^B$.** For each threshold $B > 1$, let

$$\mathbb{B}^B(t, c_{i-1}, c_i) = \mathbb{I}(\max(|w_{i+1}(c_i)|_t, |w_i(c_i)||_t, |w_i(c_{i-1})|_t, |w_{i-1}(c_{i-1})|_t) \le B)$$

$$\mathbb{B}^B(t, c_i) = \mathbb{I}(\max(|w_{i+1}(c_i)|_t, |w_i(c_i)|_t) \le B).$$

We define $R^B$ as the solution to

$$\partial_t R_i^B(G, t, c_{i-1}, c_i)] = -\Delta F_i(R^B)(G, t, c_{i-1}, c_i) \cdot \mathbb{B}^B(t, c_{i-1}, c_i).$$

We can rewrite – with an abuse of notations – in the following form:

$$\partial_t R^B(G, t, \cdot, \cdot) = \mathfrak{A}_t^B(R^B(G, t)) + H(G, t),$$

for $\mathfrak{A}_t^B$ a linear operator. (One may easily recognize that the first term corresponds to $\Delta F_i^{(1)}$ and the second one corresponds to $\Delta F_i^{(2)}$.)

By Lemma 11, we have the bound:

$$\left| \mathfrak{A}_t^B(R^B(G, t))(c_{i-1}, c_i) \right| \le K_T |R^B|_t |w_{i+1}(c_i)|_t + K_T |R^B|_t |w_i(c_i)|_t |w_{i+1}(c_i)|_t$$

$$+ K_T |R_i^B(c_i)|_t |w_{i+1}(c_i)|_t + K_T |R_{i+1}^B(c_i)|_t + K_T |R_{i-1}^B(c_{i-1})|_t |w_i(c_i)|_t.$$

Note that by Lemma 11,

$$\mathbb{E}_{C_i} \left[ |w_i(C_i)|_t^2 |w_{i+1}(C_i)|_t^2 \right] \le K_T.$$

Thus,

$$\mathbb{E}_C \left[ \left| \mathfrak{A}_t^B(R^B(G, t))(C_{i-1}, C_i) \right|^2 \right] \le K_T B^2 |R^B|_t^2.$$

Existence and uniqueness of $R^B$ follows immediately from boundedness of $\mathfrak{A}^B$. Also $R^B$ is a bounded linear functional in $G$.

**Step 2: bounds for $R^B$.** We obtain a bound on $|R^B|_t$. Observe that

$$|H(G, t)(c_{i-1}, c_i)| \le K_T |w_{i+1}(c_i)|_t |G^y|_t + K \mathbb{E}_Z \left[ |G_i^w(t, c_{i-1}, c_i, X)| \right]$$

and as such,

$$\partial_t(|R_i^B|_t^2) = \partial_t \mathbb{E}_C \left[ \left| R_i^B(G, t, C_{i-1}, C_i) \right|^2 \right]$$

$$\leq 2|R_i^B|_t \mathbb{E}_C \left[ \left| \partial R_i^B(G, t, C_{i-1}, C_i) \right|^2 \right]^{1/2}$$
$$\leq K_T |R^B|_t^2 + K_T B |R_i^B|_t^2 + K_T |G|_t^2.$$

This implies
$$\partial_t(|R^B|_t^2) \leq K_T B |R^B|_t^2 + K_T |G|_t^2,$$

so by Gronwall's lemma, for any $t \leq T$,
$$|R^B|_t \leq \exp(K_T B) \sup_{s \leq T} |G|_s = \exp(K_T B) \|G\|_{T,2}.$$

Now let us zoom into $R_{i+1}^B(c_i)$ and $R_i^B(c_i)$. From the bounds on $\left| \mathfrak{A}_t^B(R^B(G, t))(c_i, c_{i+1}) \right|$ and $|H(G, t)(c_i, c_{i+1})|$, we deduce:
$$\partial_t(|R_{i+1}^B(c_i)|_t^2) = 2\mathbb{E}_{C_{i+1}} \left[ R_{i+1}^B(G, t, c_i, C_{i+1}) \partial_t R_{i+1}^B(G, t, c_i, C_{i+1}) \right]$$
$$\leq K_T |R_{i+1}^B(c_i)|_t \left( |R^B|_t + |R_i^B(c_i)|_t + |G^y|_t + |G_{i+1}^w(c_i)|_t \right),$$

which yields
$$\partial_t |R_{i+1}^B(c_i)|_t \leq K_T \left( \exp(K_T B) \|G\|_{T,2} + |R_i^B(c_i)|_t + |G^y|_t + |G_{i+1}^w(c_i)|_t \right).$$

Similarly:
$$\partial_t(|R_i^B(c_i)|_t^2) = 2\mathbb{E}_{C_{i-1}} \left[ R_i^B(G, t, C_{i-1}, c_i) \partial_t R_i^B(G, t, C_{i-1}, c_i) \right]$$
$$\leq K_T |R_i^B(c_i)|_t \Big( (1 + |w_i(c_i)|_t)(1 + |w_{i+1}(c_i)|_t)|R^B|_t + |w_{i+1}(c_i)|_t |R_i^B(c_i)|_t$$
$$+ |R_{i+1}^B(c_i)|_t + |G_i^w(c_i)|_t + |w_{i+1}(c_i)|_t |G^y|_t \Big)$$
$$\leq K_T |R_i^B(c_i)|_t \left( B^2 |R^B|_t + B|R_i^B(c_i)|_t + |R_{i+1}^B(c_i)|_t + |G_i^w(c_i)|_t + |w_{i+1}(c_i)|_t |G^y|_t \right),$$

which yields
$$\partial_t |R_i^B(c_i)|_t \leq K_T \left( \exp(K_T B) \|G\|_{T,2} + B|R_i^B(c_i)|_t + |R_{i+1}^B(c_i)|_t + |G_i^w(c_i)|_t + B|G^y|_t \right).$$

Therefore,
$$\partial_t \left( |R_i^B(c_i)|_t + |R_{i+1}^B(c_i)|_t \right) \leq K_T \left( \exp(K_T B) \|G\|_{T,2} + B \left( |R_i^B(c_i)|_t + |R_{i+1}^B(c_i)|_t \right) + |G_i^w(c_i)|_t + |G_{i+1}^w(c_i)|_t + B|G^y|_t \right).$$

By Gronwall's lemma,
$$|R_i^B(c_i)|_t + |R_{i+1}^B(c_i)|_t \leq \left( \|G\|_{T,2} + \int_0^t \left( |G_i^w(c_i)|_s + |G_{i+1}^w(c_i)|_s + |G^y|_s \right) ds \right) \exp(K_T B).$$

**Step 3: taking $B \to \infty$.** Next we compare $R^B$ and $R^{B'}$ for $B' > B$. Let us define
$$\Delta R_i(G, t, c_{i-1}, c_i) = R_i^{B'}(G, t, c_{i-1}, c_i) - R_i^B(G, t, c_{i-1}, c_i).$$

We have that if $\mathbb{B}^B(t, c_{i-1}, c_i) = 1$, then
$$\partial_t \Delta R_i(G, t, c_{i-1}, c_i) = -\Delta F_i^{(1)}(\Delta R)(G, t, c_{i-1}, c_i).$$

If $\mathbb{B}^{B'}(t, c_{i-1}, c_i) = 1$ but $\mathbb{B}^B(t, c_{i-1}, c_i) = 0$, we have
$$\partial_t \Delta R_i(G, t, c_{i-1}, c_i) = -\Delta F_i(R^{B'})(G, t, c_{i-1}, c_i).$$

Finally, if $\mathbb{B}^{B'}(t, c_{i-1}, c_i) = 0$, then
$$\partial_t \Delta R_i(G, t, c_{i-1}, c_i) = 0.$$

With $\mathbb{D}_{B,B'}(t, c_{i-1}, c_i) = \mathbb{I}(\mathbb{B}^{B'}(t, c_{i-1}, c_i) = 1, \mathbb{B}^B(t, c_{i-1}, c_i) = 0)$, we obtain:
$$\partial_t \mathbb{E}_C \left[ |\Delta R_i(G, t, C_{i-1}, C_i)|^2 \mathbb{D}_{B,B'}(t, C_{i-1}, C_i) \right]$$
$$\leq K \mathbb{E}_C \left[ \Delta R_i(G, t, C_{i-1}, C_i) \mathbb{D}_{B,B'}(t, C_{i-1}, C_i) \cdot \partial_t \Delta R_i(G, t, C_{i-1}, C_i) \mathbb{D}_{B,B'}(t, C_{i-1}, C_i) \right]$$

$$\leq K\mathbb{E}_C\left[|\Delta R_i(G,t,C_{i-1},C_i)|^2\,\mathbb{D}_{B,B'}(t,C_{i-1},C_i)\right]^{1/2}\mathbb{E}_C\left[|\partial_t\Delta R_i(G,t,C_{i-1},C_i)|^2\,\mathbb{D}_{B,B'}(t,C_{i-1},C_i)\right]^{1/2}$$

$$\leq K_T\mathbb{E}_C\left[|\Delta R_i(G,t,C_{i-1},C_i)|^2\,\mathbb{D}_{B,B'}(t,C_{i-1},C_i)\right]^{1/2}\cdot\mathbb{E}_C\left[\mathbb{D}_{B,B'}(t,C_{i-1},C_i)\Big(|R^{B'}|_t^2|w_{i+1}(C_i)|_t^2+|R^{B'}|_t^2|w_i(C_i)|_t^2|w_{i+1}(C_i)|_t^2\right.$$

$$\left.+|R_i^{B'}(C_i)|_t^2|w_{i+1}(C_i)|_t^2+|R_{i+1}^{B'}(C_i)|_t^2+|R_i^{B'}(C_{i-1})|_t^2|w_i(C_i)|_t^2+|w_{i+1}(C_i)|_t^2|G^y|_t^2+|G_i^w(C_{i-1},C_i|_t^2\Big)\right]^{1/2}$$

$$\leq K_T\mathbb{E}_C\left[|\Delta R_i(G,t,C_{i-1},C_i)|^2\,\mathbb{D}_{B,B'}(t,C_{i-1},C_i)\right]^{1/2}\cdot\mathbb{E}_C\left[\mathbb{D}_{B,B'}(t,C_{i-1},C_i)\Big((B'^2+B'^4)|R^{B'}|_t^2\right.$$

$$\left.+B'^2|R_i^{B'}(C_i)|_t^2+|R_{i+1}^{B'}(C_i)|_t^2+B'^2|R_i^{B'}(C_{i-1})|_t^2+B'^2|G^y|_t^2+|G_i^w(C_{i-1},C_i|_t^2\Big)\right]^{1/2}.$$

Using Lemma 11 and the bounds in Step 2, we get:

$$\mathbb{E}_C\left[\mathbb{D}_{B,B'}(t,C_{i-1},C_i)\left(|R^{B'}|_t^2(B'^2+B'^4)\right)\right]\leq\|G\|_{T,2}^2\exp\left(K_TB'-K_TB^2\right).$$

Similarly,

$$\mathbb{E}_C\left[\mathbb{D}_{B,B'}(t,C_{i-1},C_i)\left(B'^2|R_i^{B'}(C_i)|_t^2+\|R_{i+1}^{B'}(C_i)\|^2+B'^2|R_i^{B'}(C_{i-1})|_t^2\right)\right]$$

$$\leq\exp\left(K_TB'\right)\mathbb{E}_C\left[\mathbb{D}_{B,B'}(t,C_{i-1},C_i)\left(\|G\|_{T,2}^2+\int_0^t\left(|G_i^w(C_i)|_s^2+|G_{i+1}^w(C_i)|_s^2+|G_{i-1}^w(C_{i-1})|_s^2+|G_i^w(C_{i-1})|_s^2+|G^y|_s^2\right)ds\right)\right]$$

$$\leq\exp\left(K_TB'-K_T\frac{\epsilon B^2}{2+\epsilon}\right)\|G\|_{T,2+\epsilon}^2,$$

$$\mathbb{E}_C\left[\mathbb{D}_{B,B'}(t,C_{i-1},C_i)\left(B'^2|G^y|_t^2+|G_i^w(C_{i-1},C_i|_t^2)\right]$$

$$\leq\exp\left(K_TB'-K_T\frac{\epsilon B^2}{2+\epsilon}\right)\|G\|_{T,2+\epsilon}^2.$$

We thus get:

$$\partial_t\mathbb{E}_C\left[|\Delta R_i(G,t,C_{i-1},C_i)|^2\,\mathbb{D}_{B,B'}(t,C_{i-1},C_i)\right]\leq\|G\|_{T,2+\epsilon}^2\exp\left(K_TB'-K_T\frac{\epsilon B^2}{2+\epsilon}\right).$$

The term where $\mathbb{B}^B(t,c_{i-1},c_i)=1$ can be bounded as in the bound for $R^B$:

$$\partial_t\mathbb{E}_C\left[|\Delta R_i(G,t,C_{i-1},C_i)|^2\,\mathbb{B}^B(t,C_{i-1},C_i)\right]\leq K_TB|\Delta R|_t^2.$$

The last two displays give:

$$\partial_t|\Delta R|_t^2\leq K_TB|\Delta R|_t^2+\|G\|_{T,2+\epsilon}^2\exp\left(K_TB'-K_T\frac{\epsilon B^2}{2+\epsilon}\right).$$

Hence,

$$|\Delta R|_t^2\leq\|G\|_{T,2+\epsilon}^2\exp\left(K_TBB'-K_T\frac{\epsilon B^3}{2+\epsilon}\right).$$

In particular, for all $B$ sufficiently large, we have $|\Delta R|_t^2\leq\|G\|_{T,2+\epsilon}^2\exp(-K_T\epsilon B^2/(2+\epsilon))$ for all $B'\leq 2B$. Thus, we can easily deduce that $R^B$ converges in $L^2$ to a limit, which is the process $R$, as $B\to\infty$. Since $|\Delta R|_t^2$ decays exponentially with $B^2$ while $|R^B|_t\leq\exp(K_TB)\|G\|_{T,2}$, we deduce that:

$$\sup_{t\leq T}|R|_t^2\leq K_{T,\epsilon}\|G\|_{T,2+\epsilon}^2.$$

We can also deduce for fixed $c_{i-1},c_i$, $R^B(G,t,c_{i-1},c_i)$ converges, as $B\to\infty$, to $R(G,t,c_{i-1},c_i)$, and $R$ satisfies Eq. (6).

**Step 4: uniqueness.** Next, we show uniqueness of $R$. Assume that $R,R'$ are two solutions to the equation in $L^2$, we have $U=R-R'$ satisfies

$$\partial_t U_i(G,t,c_{i-1},c_i)=-\Delta F_i^{(1)}(U)(G,t,c_{i-1},c_i)$$

We then have

$$|\partial_t U_i(G,t,c_{i-1},c_i)|\leq K|U|_t|w_{i+1}(c_i)|_t+K|U|_t|w_i(c_i)|_t w_{i+1}(c_i)|_t$$

$$+ K|U_i(c_i)|_t|w_{i+1}(c_i)|_t + K|U_{i+1}(c_i)|_t + K|U_{i-1}(c_{i-1})|_t|w_i(c_i)|_t.$$

Let $\kappa = \max_{t\leq T}(|R|_t, |R'|_t) < \infty$. Then

$$\partial_t|U_i(c_{i-1})|_t^2 \leq K|U_i(c_{i-1})|_t \left( \kappa + |U_{i-1}(c_{i-1})|_t + \mathbb{E}_C[|U_i(C_i)|_t^2|w_{i+1}(C_i)|_t^2]^{1/2} \right),$$

$$\partial_t|U_i(c_i)|_t^2 \leq K|U_i(c_i)|_t \left( \kappa(1 + \max(|w_{i+1}(c_i)|_t, |w_i(c_i)|_t))^2 + |w_{i+1}(c_i)|_t|U_i(c_i)|_t + |U_{i+1}(c_i)|_t \right).$$

In particular

$$\partial_t|U_L(c_L)|_t^2 \leq K|U_L(c_L)|_t \left( \kappa(1 + |w_L(c_L)|_t) + |U_L(c_L)|_t \right),$$

from which we obtain $|U_L(c_L)|_t \leq K_{\kappa,T}(1+\sup_{s\leq T}|w_L(c_L)|_s)$ for all $t \leq T$ and $c_L$. Note that $\mathbb{E}_{C_L}[K_{\kappa,T}(1+\sup_{s\leq T}|w_L(C_L)|_s)^2] < \infty$. Assume the bound on $|U_{i+1}(c_{i+1})|_t$ such that $\mathbb{E}_{C_{i+1}}\left[|U_{i+1}(C_{i+1})|_t^2|w_{i+2}(C_{i+1})|_t^2\right] < \infty$, we have

$$\partial_t|U_i(c_i)|_t^2 \leq K|U_i(c_i)|_t \left( \kappa(1 + \max(|w_{i+1}(c_i)|_t, |w_i(c_i)|_t))^2 + |w_{i+1}(c_i)|_t|U_i(c_i)|_t + |U_{i+1}(c_i)|_t \right),$$

$$\partial_t|U_{i+1}(c_i)|_t^2 \leq K|U_{i+1}(c_i)|_t \left( \kappa + |U_i(c_i)|_t + \mathbb{E}[|U_{i+1}(C_{i+1})|_t^2|w_{i+2}(C_{i+1})|_t^2]^{1/2} \right)$$
$$\leq K|U_{i+1}(c_i)|_t \left( \kappa + |U_i(c_i)|_t + K_{\kappa,T} \right).$$

Thus, $|U_{i+1}(c_i)|_t \leq K_{\kappa,T}(\int_0^t |U_i(c_i)|_s ds + 1)$. Hence,

$$\partial_t|U_i(c_i)|_t \leq \kappa \left( 1 + \max(|w_{i+1}(c_i)|_t, |w_i(c_i)|_t))^2 + |w_{i+1}(c_i)|_t|U_i(c_i)|_t + \int_0^t |U_i(c_i)|_s ds \right).$$

From this we obtain

$$|U_i(c_i)|_t \leq K_{\kappa,T} \exp(K_{\kappa,T}|w_{i+1}(c_i)|_t)(1 + \max(|w_{i+1}(c_i)|_t, |w_i(c_i)|_t))^2).$$

Note that we still have

$$\mathbb{E}_{C_i}\left[|U_i(C_i)|_t^2|w_i(C_i)|_t^2\right] \leq K_{\kappa,T}\mathbb{E}_{C_i}\left[\exp(K_{\kappa,T}|w_{i+1}(C_i)|_t)(1 + \max(|w_{i+1}(c_i)|_t, |w_i(c_i)|_t))^2)^4\right]$$
$$< \infty,$$

by Lemma 11. By induction, we obtain bounds

$$\max_{t\leq T}\max(|U_{i+1}(c_i)|_t, |U_i(c_i)|_t) \leq \exp\left(K_{\kappa,T}(1 + \max(|w_{i+1}(c_i)|_t, |w_i(c_i)|_t))\right),$$

for all $i$ and $c_i$. From here we can easily obtain a similar bound on $U_i(t, c_{i-1}, c_i)$,

$$\max_{t\leq T}|U_i(t, c_{i-1}, c_i)| \leq \exp\left(K_{\kappa,T}(1 + \max(|w_i(c_{i-1})|_t, |w_{i-1}(c_{i-1})|_t, |w_{i+1}(c_i)|_t, |w_i(c_i)|_t))\right).$$

This also implies all moments of $U$ are finite again by Lemma 11.

Using those bounds, we have, for any $B > 0$, that

$$\partial_t|U|_t^2 \leq K|U|_t^2 + K|U|_t \sum_i \left(\mathbb{E}_{C_i}\left[|U_i(C_i)|_t^2|w_{i+1}(C_i)|_t^2\right]\right)^{1/2}$$

$$\leq K(1 + B)|U|_t^2 + K|U|_t \sum_i \left(\mathbb{E}_{C_i}\left[|U_i(C_i)|_t^2|w_{i+1}(C_i)|_t^2\mathbb{I}(|w_{i+1}(C_i)|_t \geq B)\right]\right)^{1/2}.$$

$$\leq K(1 + B)|U|_t^2 + K \sum_i \mathbb{E}_{C_i}\left[|U_i(C_i)|_t^2|w_{i+1}(C_i)|_t^2\mathbb{I}(|w_{i+1}(C_i)|_t \geq B)\right].$$

Our bound on $|U_i(C_i)|_t$ together with Lemma 11 gives

$$\mathbb{E}_{C_i}\left[|U_i(C_i)|_t^2|w_{i+1}(C_i)|_t^2\mathbb{I}(|w_{i+1}(C_i)|_t \geq B)\right] \leq \exp(K_{\kappa,T}B - cB^2).$$

This implies

$$\partial_t|U|_t^2 \leq K(1 + B)|U|_t^2 + \exp(K_{\kappa,T}B - cB^2).$$

Thus,

$$\sup_{t\leq T}|U|_t^2 \leq \exp(K(1 + B)T)\exp(K_{\kappa,T}B - cB^2).$$

Sending $B$ to infinity, we immediately obtain that $|U|_t = 0$ for all $t \leq T$. Thus, the solution $R$ is unique. $\qquad\square$

Repeating the proof of Theorem 21 identically gives the following.

**Lemma 22.** *If* $\|G\|_{T,2p+\epsilon} < \infty$, *then* $R_i(G, t, \cdot, \cdot) \in L^{2p}(\Omega_{i-1} \times \Omega_i)$, *and*

$$\|R(G)\|_{T,2p}^{2p} \leq K_{T,\epsilon}\|G\|_{T,2p+\epsilon}^{2p}.$$

# D Connecting finite-width neural network fluctuations with the limit system: Proof of Theorem 5

Let us recall that
$$\mathbf{R}_i(t, j_{i-1}, j_i) = \sqrt{N}(\mathbf{w}_i(t, j_{i-1}, j_i) - w_i(t, C_{i-1}(j_{i-1}), C_i(j_i)).$$
Our goal is to prove Theorem 5. A major technical difficulty here lies in establishing suitable a priori moment bounds for $\mathbf{R}$ that are independent of $N$. This task suffers from similar unboundedness issues that are encountered in the proof of Theorem 21. Again this is a problem unique to the multilayer structure. Yet it requires a delicate argument that is different from the one in the proof of Theorem 21. In particular, Lemma 23 below plays a crucial role and computes very high moments (of order increasing with $N$) of the neural network's weights under GD evolution and randomized initialization. It is also a result of independent interest, which we expect to have applications beyond the current pursuit.

In the following, we denote
$$\|\mathbf{R}_i(t) - R_i(t)\|_{2p}^{2p} = \sum_{i=1}^{L} \mathbb{E}_J \left[ \left| \mathbf{R}_i(t, J_{i-1}, J_i) - R_i(\tilde{G}, t, C_{i-1}(J_{i-1}), C_i(J_i)) \right|^{2p} \right].$$

For brevity, let us write
$$\tilde{w}_i(t, j_{i-1}, j_i) = w_i(t, C_{i-1}(j_{i-1}), C_i(j_i)).$$
Similar to the development in Appendix B, define
$$\|\tilde{w}_i(t, j_i)\|_{2p}^{2p} = \mathbb{E}_{J_{i-1}} \left[ |w_i(t, C_{i-1}(J_{i-1}), C_i(j_i))|^{2p} \right],$$
$$\|\mathbf{w}_i(t, j_i)\|_{2p}^{2p} = \mathbb{E}_{J_{i-1}} \left[ |\mathbf{w}_i(t, J_{i-1}, j_i)|^{2p} \right],$$
$$\|\tilde{w}_{i+1}(t, j_i)\|_{2p}^{2p} = \mathbb{E}_{J_{i+1}} \left[ |w_i(t, C_i(j_i), C_{i+1}(J_{i+1}))|^{2p} \right],$$
$$\|\mathbf{w}_{i+1}(t, j_i)\|_{2p}^{2p} = \mathbb{E}_{J_{i+1}} \left[ |\mathbf{w}_i(t, j_i, J_{i+1})|^{2p} \right],$$
$$\|\tilde{w}_i(t)\|_{2p}^{2p} = \mathbb{E}_{J_i} \left[ \|\tilde{w}_i(t, J_i)\|_{2p}^{2p} \right],$$
$$\|\mathbf{w}_i(t)\|_{2p}^{2p} = \mathbb{E}_{J_i} \left[ \|\mathbf{w}_i(t, J_i)\|_{2p}^{2p} \right].$$
We define similarly $\|\mathbf{R}_i(t, j_i)\|_{2p}, \|R_i(t, j_i)\|_{2p}, \|\mathbf{R}_{i+1}(t, j_{i+1})\|_{2p}, \|R_{i+1}(t, j_{i+1})\|_{2p}, \|\mathbf{R}_i(t)\|_{2p}, \|R_i(t)\|_{2p}$ (where quantities involving $R$ should be computed w.r.t. $R(\tilde{G}, t)$). We also let
$$\|\mathbf{R}(t)\|_{2p}^{2p} = \sum_{i=1}^{L} \|\mathbf{R}_i(t)\|_{2p}^{2p}, \qquad \|R(t)\|_{2p}^{2p} = \sum_{i=1}^{L} \|R_i(t)\|_{2p}^{2p}.$$
In all of these, when we drop the subscript $2p$, we implicitly take $2p = 2$. For $B > 0$, define
$$\tilde{\mathbb{B}}_i^{B,2k}(t, j_i) = \mathbb{I}(\|\tilde{w}_i(t, j_i)\|_{2k}, \|\tilde{w}_{i+1}(t, j_i)\|_{2k} \le B),$$
$$\mathbf{B}_i^{B,2k}(t, j_i) = \mathbb{I}(\|\mathbf{w}_i(t, j_i)\|_{2k}, \|\mathbf{w}_{i+1}(t, j_i)\|_{2k} \le B),$$
$$\mathbb{B}^{B,2k}(t, j_i) = \tilde{\mathbb{B}}_i^{B,2k}(t, j_i)\mathbf{B}_i^{B,2k}(t, j_i),$$
$$\mathbb{B}^{B,2k}(t, j_{i-1}, j_i) = \mathbb{B}^{B,2k}(t, j_{i-1})\mathbb{B}^{B,2k}(t, j_i).$$
We drop the superscripts $B$ and $2k$ if they are clear from context. Note that these are random variables due to the randomness of sampling $\{C_i(j_i): j_i \in [N_i], i \in [L]\}$, whose expectation is denoted by $\mathbf{E}$.

## D.1 High moments of neural network's weights under GD

We obtain the following estimate, which is important for our development.

**Lemma 23.** *For $k \ge 1$ and $B > 0$, we have:*
$$\mathbf{E}\mathbb{E}_J \left[ |w_i(t, C_{i-1}(J_{i-1}), C_i(J_i))|^k \right] = \mathbb{E}_C \left[ |w_i(t, C_{i-1}, C_i)|^k \right],$$
*and for all $N \ge (kL)^6$,*
$$\mathbf{E}\mathbb{E}_J \left[ |\mathbf{w}_i(t, J_{i-1}, J_i)|^k \right] \le k^{k/2} K_T^{kL}.$$
*As an immediate corollary, for a fixed constant $C$ and $B \le N^{K_C}$,*
$$\mathbf{E} \left[ \mathbf{B}_i^{B,2C}(t, j_i) \right] \le \exp(-K_{T,C}B^2).$$

*Proof.* The first equality is trivial. For the second bound, we consider $i \geq 2$; the case $i = 1$ can be done identically. We have

$$\partial_t \mathbb{E}_J \left[ |\mathbf{w}_i(t, J_{i-1}, J_i)|^k \right] = k \mathbb{E}_J \left[ \mathbf{w}_i(t, J_{i-1}, J_i)^{k-1} \partial_t \mathbf{w}_i(t, J_{i-1}, J_i) \right]$$

$$\leq k \mathbb{E}_J \left[ |\mathbf{w}_i(t, J_{i-1}, J_i)|^k \right]^{(k-1)/k} \mathbb{E}_J \left[ |\partial_t \mathbf{w}_i(t, J_{i-1}, J_i)|^k \right]^{1/k}.$$

Note that

$$\mathbb{E}_J \left[ |\partial_t \mathbf{w}_i(t, J_{i-1}, J_i)|^k \right] = \mathbb{E}_J \left[ \left| \mathbb{E}_Z \left[ \partial_2 \mathcal{L}(Y, \mathbf{y}) \frac{\partial \hat{\mathbf{y}}(t, X)}{\partial \mathbf{w}_i(J_{i-1}, J_i)} \right] \right|^k \right]$$

$$\leq K \mathbb{E}_J \left[ \mathbb{E}_Z \left[ \left| \frac{\partial \hat{\mathbf{y}}(t, X)}{\partial \mathbf{w}_i(J_{i-1}, J_i)} \right|^2 \right]^{k/2} \right]$$

$$\leq K^k \mathbb{E}_J \left[ |\mathbf{w}_{i+1}(t, J_i, J_{i+1})|^k \right] \prod_{j \geq i+2} \mathbb{E}_J \left[ \mathbf{w}_j(t, J_{j-1}, J_j)^2 \right]^{k/2}.$$

Thus,

$$\partial_t \mathbb{E}_J \left[ |\mathbf{w}_i(t, J_{i-1}, J_i)|^k \right] \leq K k \mathbb{E}_J \left[ |\mathbf{w}_i(t, J_{i-1}, J_i)|^k \right]^{(k-1)/k} \mathbb{E}_J \left[ |\mathbf{w}_{i+1}(t, J_i, J_{i+1})|^k \right]^{1/k} \prod_{j \geq i+2} \mathbb{E}_J \left[ \mathbf{w}_j(t, J_{j-1}, J_j)^2 \right]^{1/2}.$$

In particular,

$$\partial_t \mathbb{E}_J [|\mathbf{w}_L(t, J_{L-1}, J_L)|^k]^{1/k} \leq K_T,$$

which implies

$$\mathbb{E}_J [|\mathbf{w}_L(t, J_{L-1}, J_L)|^k]^{1/k} \leq \mathbb{E}_J [|\mathbf{w}_L(0, J_{L-1}, J_L)|^k]^{1/k} + Kt.$$

Then inductively, we obtain that

$$\sup_{t \leq T} \mathbb{E}_J [|\mathbf{w}_i(t, J_{i-1}, J_i)|^k]^{1/k}$$

$$\leq \mathbb{E}_J [|\mathbf{w}_i(0, J_{i-1}, J_i)|^k]^{1/k} + p_i \left( \left( \mathbb{E}_J \left[ \mathbf{w}_j(0, J_{j-1}, J_j)^2 \right]^{1/2} \right)_{j \geq i+1}, \left( \mathbb{E}_J \left[ |\mathbf{w}_j(0, J_{j-1}, J_j)|^k \right]^{1/k} \right)_{j \geq i+1}, T \right),$$

where $p_i$ is a polynomial of degree at most $L - i$, and furthermore, in each monomial in $p_i$ there is at most one term of degree one from the variables $\left( \mathbb{E}_J \left[ |\mathbf{w}_j(0, J_{j-1}, J_j)|^k \right]^{1/k} \right)_{j \geq i+1}$. This allows us to bound

$$\mathbf{E}\mathbb{E}_J [|\mathbf{w}_i(t, J_{i-1}, J_i)|^k] \leq K^k \mathbf{E}\mathbb{E}_J [|\mathbf{w}_i(0, J_{i-1}, J_i)|^k]$$

$$+ K_T^k \sum_{j \geq i+1} \mathbf{E} \left[ \mathbb{E}_J [|\mathbf{w}_i(0, J_{i-1}, J_i)|^k] \sum_{j' \geq i+1} \mathbb{E}_J \left[ |\mathbf{w}_{j'}(0, J_{j'-1}, J_{j'})|^2 \right]^{kL/2} \right].$$

We have

$$\mathbf{E} \left[ \mathbb{E}_J [|\mathbf{w}_i(0, J_{i-1}, J_i)|^k] \mathbb{E}_J \left[ |\mathbf{w}_{j'}(0, J_{j'-1}, J_{j'})|^2 \right]^{kL/2} \right]$$

$$\leq \left( \mathbf{E} \left[ \mathbb{E}_J [|\mathbf{w}_i(0, J_{i-1}, J_i)|^k]^2 \right] \right)^{1/2} \left( \mathbf{E} \left[ \mathbb{E}_J \left[ \mathbf{w}_{j'}(0, J_{j'-1}, J_{j'})^2 \right]^{kL} \right] \right)^{1/2}$$

$$\leq \left( \mathbf{E} \left[ \mathbb{E}_J [\mathbf{w}_i(0, J_{i-1}, J_i)^{2k}] \right] \right)^{1/2} \left( \mathbf{E} \left[ \mathbb{E}_J \left[ \mathbf{w}_{j'}(0, J_{j'-1}, J_{j'})^2 \right]^{kL} \right] \right)^{1/2}$$

$$\leq K^k k^{k/2} \left( \mathbf{E} \left[ \mathbb{E}_J \left[ \mathbf{w}_{j'}(0, J_{j'-1}, J_{j'})^2 \right]^{kL} \right] \right)^{1/2}.$$

Let us analyze the term in the last display:

$$\mathbb{E}_{J'} \left[ \mathbf{w}_{j'}(0, J_{j'-1}, J_{j'})^2 \right]^{kL} = N^{-2kL} \sum_{\alpha_1, \dots, \alpha_{kL} \in [N_{j'-1}], \beta_1, \dots, \beta_{kL} \in [N_{j'}]} \prod_{t=1}^{kL} \mathbf{w}_{j'}(0, \alpha_t, \beta_t)^2.$$

Furthermore, if each index appears at most once among $\alpha, \beta$, then

$$\mathbf{E} \left[ \prod_{t=1}^{kL} \mathbf{w}_{j'}(0, \alpha_t, \beta_t)^2 \right] = \prod_{t=1}^{kL} \mathbb{E}_C \left[ w_{j'}(0, C_{j'-1}, C_{j'})^2 \right]^{kL}$$

$$\leq K^{kL}.$$

On the other hand, if some index appears at least twice, let $d(a)$ be the number of times $a$ appears among $\{\alpha_s\}$, and we similarly define $d(b)$. We have by Finner's inequality[3] that

$$\mathbf{E}\left[\prod_{t=1}^{kL}\mathbf{w}_{j'}(0,\alpha_t,\beta_t)^2\right] \leq \prod_{t=1}^{kL}\mathbf{E}\left[\mathbf{w}_{j'}(0,\alpha_t,\beta_t)^{2\max(d(\alpha_t),d(\beta_t))}\right]^{1/\max(d(\alpha_t),d(\beta_t))}$$

$$\leq K^{k(1+L)}\prod_{t=1}^{kL}\max(d(\alpha_t),d(\beta_t)).$$

Note that $\sum_a d(a) = \sum_b d(b) = kL$. Fixing the multisets $\{d(a)\} = \{d(a) : a \in [N_{j'-1}]\}$, $\{d(b)\} = \{d(b) : b \in [N_{j'}]\}$, let $M_a$ be the number of positive values in $\{d(a)\}$ and $M_b$ the number of positive values of $\{d(b)\}$. Let $H_a$ be the number of values at least 2 in $\{d(a)\}$ and similarly for $H_b$. The number of unlabeled bipartite graphs for which the degrees on one part are given by $\{d(a)\}$ and the degrees of the other part are given by $\{d(b)\}$ is at most $(H_b+1)^{H_a}(H_a+1)^{H_b}$ (each half-edge from a vertex of degree at least 2 on the $a$-side is chosen to match to a half-edge of a vertex of degree at least 2 on the $b$-side or a vertex of degree 1, for a total of $H_b+1$ choices). After choosing the unlabeled bipartite graph, there are at most $N^{M_a+M_b}$ ways to label the vertices of the graph with indices in $[N_{j'-1}]$ or $[N_{j'}]$. Let $Q$ be the number of edges between vertices both having degree 1. We have at most $(kL)!/Q!$ ways to assign index $t \in [kL]$ to edges of the labeled bipartite graph. We have that $Q \geq kL - \sum_a d(a)\mathbb{I}(d(a) \geq 2) - \sum_b d(b)\mathbb{I}(d(b) \geq 2)$, while

$$M_a = \sum_a \mathbb{I}(d(a) \geq 1) \leq \sum_a \mathbb{I}(d(a) = 1) + \frac{1}{2}\sum_a d(a)\mathbb{I}(d(a) \geq 2) = kL - \frac{1}{2}\sum_a d(a)\mathbb{I}(d(a) \geq 2),$$

and similarly $M_b \leq kL - \frac{1}{2}\sum_b d(b)\mathbb{I}(d(b) \geq 2)$, so $M_a + M_b \leq 2kL - \frac{1}{2}\left(\sum_a d(a)\mathbb{I}(d(a) \geq 2) + \sum_b d(b)\mathbb{I}(d(b) \geq 2)\right)$. Thus, the number of edges between vertices both having degree 1 is at least $2(M_a + M_b) - 3kL$. We also have $H_a \leq \frac{1}{2}\sum_a d(a)\mathbb{I}(d(a) \geq 2) \leq kL - M_a$ and $H_b \leq kL - M_b$. Finally, note that

$$\prod_{t=1}^{kL}\max(d(\alpha_t),d(\beta_t)) \leq \left(\frac{kL}{\min(1,H_a)}\right)^{H_a}\left(\frac{kL}{\min(1,H_b)}\right)^{H_b}.$$

Thus, using convention that $x! = 1$ if $x \leq 0$ and using that the number of choices of $\{d(a)\}$ given $M_a = m_a$ is at most $\binom{kL-1}{m_a-1}$, we get

$$\sum_{\alpha,\beta}\prod_{t=1}^{kL}\max(d(\alpha_t),d(\beta_t)) \leq \sum_{\Gamma}\frac{(kL)!}{(2m_a + 2m_b - 3kL)!}N^{m_a+m_b}(h_b+1)^{h_a}(h_a+1)^{h_b}\left(\frac{kL}{h_a}\right)^{h_a}\left(\frac{kL}{h_b}\right)^{h_b}\binom{kL-1}{m_a-1}\binom{kL-1}{m_b-1}$$

$$\leq \sum_{\Gamma}\frac{(kL)!}{(2m_a + 2m_b - 3kL)!}N^{m_a+m_b}\left(\frac{2(kL)^2}{\min(1,h_a)}\right)^{h_a}\left(\frac{2(kL)^2}{\min(1,h_b)}\right)^{h_b}\binom{kL-1}{m_a-1}\binom{kL-1}{m_b-1}$$

$$\leq K^{kL}\sum_{u=0}^{2kL}N^{2kL-u}\left(\frac{4(kL)^4}{\min(1,u)}\right)^u$$

$$\leq K^{kL}N^{2kL}\sum_{u=0}^{2kL}\left(\frac{4(kL)^4}{N\min(1,u)}\right)^u,$$

where we use the shorthands $\Gamma = \{m_a, m_b \in [kL], h_a, h_b \leq kL/2, h_a \leq kL - m_a, h_b \leq kL - m_b\}$. Hence, for $N > (kL)^6$, we have

$$\sum_{\alpha,\beta}\prod_{t=1}^{kL}\max(d(\alpha_t),d(\beta_t)) \leq K^{kL}N^{2kL}.$$

In particular,

$$\mathbf{E}\mathbb{E}_J\left[\mathbf{w}_{j'}(0,J_{j'-1},J_{j'})^2\right]^{kL} \leq K^{kL}.$$

Therefore,

$$\mathbf{E}\mathbb{E}_J[\mathbf{w}_i(t,J_{i-1},J_i)^k] \leq k^{k/2}K_T^{kL}.$$

$\square$

*Remark* 24. The same argument shows that for any $\ell \geq 1$ and $N \geq (K_\ell kL)^{16}$,

$$\mathbf{E}\mathbb{E}_J\left[\mathbf{w}_j(0,J_{j-1},J_j)^{2\ell}\right]^{kL} \leq K_\ell^{kL}.$$

---

[3]Helmut Finner, "A generalization of Holder's inequality and some probability inequalities", The Annals of probability (1992), pp. 1893--1901.

## D.2 A priori estimates at the fluctuation level

**Lemma 25.** *We have for any $t \leq T$:*

$$\mathbf{E}\mathbb{E}_J\left[\left(\sqrt{N}(\tilde{H}_i(t, C_i(J_i), x) - H_i(t, C_i(J_i), x)\right)^{2k}\right] \leq K_{T,k},$$

$$\mathbf{E}\mathbb{E}_J\left[\left(\sqrt{N}\left(\frac{\partial\tilde{y}(t,x)}{\partial\tilde{H}_i(C_i(J_i))} - \frac{\partial\hat{y}(t,x)}{\partial H_i(C_i(J_i))}\right)\right)^{2k}\right] \leq K_{T,k},$$

$$\mathbf{E}\mathbb{E}_J\left[\left(\sqrt{N}\left(\frac{\partial\tilde{y}(t,x)}{\partial w_i(C_{i-1}(J_{i-1}), C_i(J_i))} - \frac{\partial\hat{y}(t,x)}{\partial w_i(C_{i-1}(J_{i-1}), C_i(J_i))}\right)\right)^{2k}\right] \leq K_{T,k},$$

$$\mathbf{E}\left[\left(\sqrt{N}\left(\partial_2\mathcal{L}(y, \tilde{y}(t,x)) - \partial_2\mathcal{L}(y, \hat{y}(t,x))\right)\right)^{2k}\right] \leq K_{T,k},$$

*Proof.* Let us drop $t$ and $x$ from the notations for brevity. We also recall Lemma 11. We prove claim by claim:

- We have:
$$\mathbf{E}\mathbb{E}_J\left[\left(\sqrt{N}(\tilde{H}_1(C_1(J_1)) - H_1(C_1(J_1)))\right)^{2k}\right] = 0$$

  and using induction,

$$\mathbf{E}\mathbb{E}_J\left[\left(\sqrt{N}(\tilde{H}_i(C_i(J_i)) - H_i(C_i(J_i)))\right)^{2k}\right]$$

$$= \mathbf{E}\mathbb{E}_{J_i}\left[\left(\sqrt{N}\mathbb{E}_{J_{i-1}}\left[w_i\left(C_{i-1}\left(J_{i-1}\right), C_i\left(J_i\right)\right)\varphi_{i-1}\left(\tilde{H}_{i-1}\left(C_{i-1}\left(J_{i-1}\right)\right)\right)\right]\right.\right.$$

$$\left.\left. - \sqrt{N}\mathbb{E}_{C_{i-1}}\left[w_i\left(C_{i-1}, C_i\left(J_i\right)\right)\varphi_{i-1}\left(H_{i-1}\left(C_{i-1}\right)\right)\right]\right)^{2k}\right]$$

$$\leq K_k\mathbf{E}\mathbb{E}_{J_i}\left[\mathbb{E}_{C_{i-1}}\left[\left|w_i\left(C_{i-1}, C_i\left(J_i\right)\right)\varphi_{i-1}\left(H_{i-1}\left(t, C_{i-1}\right)\right)\right|^2\right]^k\right]$$

$$+ K_k\mathbf{E}\mathbb{E}_{J_i}\left[\left(\sqrt{N}\mathbb{E}_{J_{i-1}}\left[w_i\left(C_{i-1}\left(J_{i-1}\right), C_i\left(J_i\right)\right)\left(\varphi_{i-1}\left(\tilde{H}_{i-1}\left(C_{i-1}\left(J_{i-1}\right)\right)\right) - \varphi_{i-1}\left(H_{i-1}\left(C_{i-1}\left(J_{i-1}\right)\right)\right)\right)\right]\right)^{2k}\right]$$

$$\leq K_{T,k} + K_k\mathbf{E}\mathbb{E}_{J_i}\left[\mathbb{E}_{J_{i-1}}\left[\left|w_i\left(C_{i-1}\left(J_{i-1}\right), C_i\left(J_i\right)\right)\right|^{2k}\right]\mathbb{E}_{J_{i-1}}\left[N\left(\tilde{H}_{i-1}\left(C_{i-1}\left(J_{i-1}\right)\right) - H_{i-1}\left(C_{i-1}\left(J_{i-1}\right)\right)\right)^{2k}\right]\right]$$

$$\leq K_{T,k}.$$

- Next we have:
$$\left(\sqrt{N}\left(\frac{\partial\tilde{y}}{\partial\tilde{H}_L(1)} - \frac{\partial\hat{y}}{\partial H_i(1)}\right)\right)^{2k} \leq K_k\left(\sqrt{N}\left(\tilde{H}_L(1) - H_i(1)\right)\right)^{2k} \leq K_{T,k}.$$

  Using backward induction:

$$\mathbf{E}\mathbb{E}_J\left[\left(\sqrt{N}\left(\frac{\partial\tilde{y}}{\partial\tilde{H}_i(C_i(J_i))} - \frac{\partial\hat{y}}{\partial H_i(C_i(J_i))}\right)\right)^{2k}\right]$$

$$\leq K_k\mathbf{E}\mathbb{E}_{J_i}\left[\mathbb{E}_{C_{i+1}}\left[\left|\frac{\partial\hat{y}}{\partial H_{i+1}(C_{i+1})}w_{i+1}\left(C_i\left(J_i\right), C_{i+1}\right)\right|^2\right]^k\right]$$

$$+ K_k\mathbf{E}\mathbb{E}_{J_i}\left[\mathbb{E}_{J_{i+1}}\left[\sqrt{N}\left|\frac{\partial\tilde{y}}{\partial\tilde{H}_{i+1}(C_{i+1}(J_{i+1}))} - \frac{\partial\hat{y}}{\partial H_{i+1}(C_{i+1}(J_{i+1}))}\right||w_{i+1}\left(C_i\left(J_i\right), C_{i+1}\left(J_{i+1}\right)\right)|\right]^{2k}\right]$$

$$+ K_k\mathbf{E}\mathbb{E}_{J_i}\left[\mathbb{E}_{J_{i+1}}\left[\sqrt{N}\left|\frac{\partial\hat{y}}{\partial H_{i+1}(C_{i+1}(J_{i+1}))}\right||w_{i+1}\left(C_i\left(J_i\right), C_{i+1}\left(J_{i+1}\right)\right)|\left|\tilde{H}_i\left(C_i\left(J_i\right)\right) - H_i\left(C_i\left(J_i\right)\right)\right|\right]^{2k}\right]$$

$$\leq K_{T,k} + K_k \mathbf{E}\mathbb{E}_{J_{i+1}}\left[\left(\sqrt{N}\left|\frac{\partial\tilde{y}}{\partial\tilde{H}_{i+1}(C_{i+1}(J_{i+1}))} - \frac{\partial\hat{y}}{\partial H_{i+1}(C_{i+1}(J_{i+1}))}\right|\right)^{4k}\right]^{1/2}$$

$$+ K_k \mathbf{E}\mathbb{E}_{J_i}\left[\left(\sqrt{N}\left|\tilde{H}_i\left(C_i\left(J_i\right)\right) - H_i\left(C_i\left(J_i\right)\right)\right|\right)^{8k}\right]^{1/3}$$

$$\leq K_{T,k}.$$

- For the third claim:

$$\left(\sqrt{N}\left(\frac{\partial\tilde{y}}{\partial w_i(c_{i-1}, c_i)} - \frac{\partial\hat{y}}{\partial w_i(c_{i-1}, c_i)}\right)\right)^{2k}$$

$$\leq K_k\left(\sqrt{N}\left(\frac{\partial\tilde{y}}{\partial\tilde{H}_i(c_i)} - \frac{\partial\hat{y}}{\partial H_i(c_i)}\right)\right)^{2k} + K_k\left(\frac{\partial\hat{y}}{\partial H_i(c_i)}\right)^{2k}\left(\sqrt{N}\left(\tilde{H}_{i-1}\left(c_{i-1}\right) - H_{i-1}\left(c_{i-1}\right)\right)\right)^{2k}.$$

Combining the previous claims, it is then easy to see that

$$\mathbf{E}\mathbb{E}_J\left[\left(\sqrt{N}\left(\frac{\partial\tilde{y}}{\partial w_i(C_{i-1}(J_{i-1}), C_i(J_i))} - \frac{\partial\hat{y}}{\partial w_i(C_{i-1}(J_{i-1}), C_i(J_i))}\right)\right)^{2k}\right] \leq K_{T,k}.$$

- The fourth claim is also immediate from the first claim by observing that

$$\sqrt{N}\left|\partial_2\mathcal{L}(\cdot, \tilde{y}) - \partial_2\mathcal{L}(\cdot, \hat{y})\right| \leq K\left|\sqrt{N}\left(\tilde{y} - \hat{y}\right)\right| \leq K\left|\sqrt{N}\left(\tilde{H}_L\left(1\right) - H_L\left(1\right)\right)\right|.$$

$\square$

**Lemma 26** (Lipschitz bounds). *We have for any $t \leq T$:*

$$\left(\sqrt{N}(\mathbf{H}_i(t, j_i, x) - \tilde{H}_i(t, C_i(j_i), x)\right)^{2k} \leq \|\mathbf{R}_i(j_i)\|_{2k}^{2k} + K_k\|\mathbf{R}\|_{2k}^{2k}(1 + \|\tilde{w}_i(j_i)\|_{2k}^{2k})\prod_{j=1}^{i-1}(1 + \|\tilde{w}_i\|_{2k}^{2k}),$$

$$\left(\sqrt{N}\left(\frac{\partial\hat{\mathbf{y}}(t, x)}{\partial\mathbf{H}_i(j_i)} - \frac{\partial\tilde{y}(t, x)}{\partial\tilde{H}_i(C_i(j_i))}\right)\right)^{2k}$$

$$\leq K_k\|\tilde{w}_{i+1}(j_i)\|_{2k}^{2k}\Bigg(\|R\|_{2k}^{2k}(1 + \|\tilde{w}_i(j_i)\|_{2k}^{2k})(1 + \|\tilde{w}_{i+1}\|_{4k}^{2k})\prod_{j\leq i-1}(1 + \|\tilde{w}_j\|_{2k}^{2k})\prod_{j\geq i+2}(1 + \|\tilde{w}_j\|_{4k}^{4k})$$

$$+ \sum_{j\geq i+1}\mathbb{E}_{J_j}[\|R_j(J_j)\|_2^{2k}\|\tilde{w}_{j+1}(J_j)\|_{2k}^{2k}]\prod_{i+2\leq j', j'\neq j+1}(1 + \|\tilde{w}_{j'}\|_{2k}^{2k}) + \|R_i(j_i)\|_2^{2k}\prod_{i+2\leq j'}(1 + \|\tilde{w}_{j'}\|_{2k}^{2k})\Bigg)$$

$$+ \|\mathbf{R}_{i+1}(j_i)\|_2^{2k}\prod_{j\geq i+2}(1 + \|\tilde{w}_j\|_{2k}^{2k}),$$

$$\left(\sqrt{N}\left(\frac{\partial\hat{\mathbf{y}}(t, x)}{\partial\mathbf{w}_i(j_{i-1}, j_i)} - \frac{\partial\tilde{y}(t, x)}{\partial w_i(C_{i-1}(j_{i-1}), C_i(j_i))}\right)\right)^{2k}$$

$$\leq K_k\left(\sqrt{N}\left(\frac{\partial\hat{\mathbf{y}}(t, x)}{\partial\mathbf{H}_i(j_i)} - \frac{\partial\tilde{y}(t, x)}{\partial\tilde{H}_i(C_i(j_i))}\right)\right)^{2k}$$

$$+ K_k\|\tilde{w}_{i+1}(j_i)\|_2^{2k}\prod_{j\geq i+2}\|\tilde{w}_j\|_2^{2k}\cdot\left(\sqrt{N}(\mathbf{H}_{i-1}(t, j_{i-1}, x) - \tilde{H}_{i-1}(t, C_{i-1}(j_{i-1}), x)\right)^{2k},$$

$$\sqrt{N}\left(\partial_2\mathcal{L}(y, \hat{\mathbf{y}}(t, x)) - \partial_2\mathcal{L}(y, \tilde{y}(t, x))\right) \leq K_k\|\mathbf{R}\|_{2k}^{2k}\prod_{j=1}^{L}(1 + \|\tilde{w}_i\|_{2k}^{2k}).$$

*In the above, we have dropped the notational dependency on $t$ on the right-hand side for brevity.*

*Proof.* Let us consider the first claim:

$$\left(\sqrt{N}(\mathbf{H}_i(t,j_i,x) - \tilde{H}_i(t,C_i(j_i),x)\right)^{2k}$$

$$= \left(\sqrt{N}\mathbb{E}_{J_{i-1}}[\mathbf{w}_i(t,C_{i-1}(J_{i-1}),C_i(j_i))\varphi_{i-1}(\mathbf{H}_{i-1}(t,C_{i-1}(J_{i-1}),x)) - w_i(t,C_{i-1}(J_{i-1}),C_i(j_i))\varphi_{i-1}(\tilde{H}_{i-1}(t,C_{i-1}(J_{i-1}),x))]\right)^{2k}$$

$$\leq K^k \left(\mathbb{E}_{J_{i-1}}\left[\left(\sqrt{N}(\mathbf{w}_i(t,C_{i-1}(J_{i-1}),C_i(j_i)) - w_i(t,C_{i-1}(J_{i-1}),C_i(j_i)))\right)^2\right]\right)^k$$

$$+ K^k \left(\mathbb{E}_{J_{i-1}}\left[\left(\sqrt{N}(\mathbf{H}_{i-1}(t,C_{i-1}(J_{i-1}),x) - \tilde{H}_{i-1}(t,C_{i-1}(J_{i-1}),x))\right)^2\right]\mathbb{E}_{J_{i-1}}\left[w_i(t,C_{i-1}(J_{i-1}),C_i(j_i))^2\right]\right)^k$$

$$\leq K^k\mathbb{E}_{J_{i-1}}\left[\left(\sqrt{N}(\mathbf{w}_i(t,C_{i-1}(J_{i-1}),C_i(j_i)) - w_i(t,C_{i-1}(J_{i-1}),C_i(j_i)))\right)^2\right]^k$$

$$+ K^k\mathbb{E}_{J_{i-1}}\left[\left(\sqrt{N}(\mathbf{H}_{i-1}(t,C_{i-1}(J_{i-1}),x) - \tilde{H}_{i-1}(t,C_{i-1}(J_{i-1}),x))\right)^2\right]^k \mathbb{E}_{J_{i-1}}\left[w_i(t,C_{i-1}(J_{i-1}),C_i(j_i))^2\right]^k.$$

Furthermore,

$$\left(\sqrt{N}(\mathbf{H}_1(t,j_1,x) - \tilde{H}_i(t,C_1(j_1),x)\right)^{2k} \leq K^k\left(\sqrt{N}(\varphi_1(\mathbf{w}_1(t,C_1(j_1))\cdot x) - \varphi_1(w_1(t,C_1(j_1))\cdot x)\right)^{2k}$$

$$\leq K^k\left(\sqrt{N}\left|\mathbf{w}(t,C_1(j_1)) - w_1(t,C_1(j_1))\right|\cdot|x|\right)^{2k}$$

$$\leq K^k\left(\sqrt{N}\left|\mathbf{w}(t,C_1(j_1)) - w_1(t,C_1(j_1))\right|\right)^{2k}.$$

Thus, by induction, we obtain that

$$\left(\sqrt{N}(\mathbf{H}_i(t,j_i,x) - \tilde{H}_i(t,C_i(j_i),x)\right)^{2k} \leq \|\mathbf{R}_i(t,j_i)\|_{2k}^{2k} + K_k\|\mathbf{R}(t)\|_{2k}^{2k}(1 + \|\tilde{w}_i(j_i)\|_{2k}^{2k})\prod_{j=1}^{i-1}(1 + \|\tilde{w}_i\|_{2k}^{2k}).$$

The remaining claims can be obtained similarly. $\qquad\square$

**Lemma 27** (A-priori moment estimate of $\mathbf{R}$). *We have for any $t \leq T$,*

$$\mathbb{E}\|\mathbf{R}(t)\|_{2k}^{2k} \leq K_{k,T}.$$

*Proof.* In the following, let us drop the dependency on $t$ for brevity. From the bounds in Lemmas 26 and 25, we can obtain the estimate

$$|\partial_t\mathbf{R}_i(j_{i-1},j_i)|^{2k}$$

$$= \left|\sqrt{N}\mathbb{E}_Z\left[\partial_2\mathcal{L}(Y,\hat{\mathbf{y}}(X))\frac{\partial\hat{\mathbf{y}}(X)}{\partial\mathbf{w}_i(j_{i-1},j_i)} - \partial_2\mathcal{L}(Y,\hat{y}(X))\frac{\partial\hat{y}(X)}{\partial w_i(C_{i-1}(j_{i-1}),C_i(j_i))}\right]\right|^{2k}$$

$$\leq K_k\left|\sqrt{N}\mathbb{E}_Z\left[\partial_2\mathcal{L}(Y,\hat{\mathbf{y}}(X))\frac{\partial\hat{\mathbf{y}}(X)}{\partial\mathbf{w}_i(j_{i-1},j_i)} - \partial_2\mathcal{L}(Y,\tilde{y}(X))\frac{\partial\tilde{y}(X)}{\partial w_i(C_{i-1}(j_{i-1}),C_i(j_i))}\right]\right|^{2k}$$

$$+ K_k\left|\sqrt{N}\mathbb{E}_Z\left[\partial_2\mathcal{L}(Y,\tilde{y}(X))\frac{\partial\tilde{y}(X)}{\partial w_i(C_{i-1}(j_{i-1}),C_i(j_i))} - \partial_2\mathcal{L}(Y,\hat{y}(X))\frac{\partial\hat{y}(X)}{\partial w_i(C_{i-1}(j_{i-1}),C_i(j_i))}\right]\right|^{2k}$$

$$\leq K_k\left|\sqrt{N}\mathbb{E}_Z\left[\frac{\partial\hat{\mathbf{y}}(X)}{\partial\mathbf{w}_i(j_{i-1},j_i)} - \frac{\partial\tilde{y}(X)}{\partial w_i(C_{i-1}(j_{i-1}),C_i(j_i))}\right]\right|^{2k}$$

$$+ K_k\left|\sqrt{N}\mathbb{E}_Z\left[(\partial_2\mathcal{L}(Y,\hat{\mathbf{y}}(X)) - \partial_2\mathcal{L}(Y,\tilde{y}(X)))\frac{\partial\tilde{y}(X)}{\partial w_i(C_{i-1}(j_{i-1}),C_i(j_i))}\right]\right|^{2k}$$

$$+ K_k\left|\sqrt{N}\mathbb{E}_Z\left[\frac{\partial\tilde{y}(X)}{\partial w_i(C_{i-1}(j_{i-1}),C_i(j_i))} - \frac{\partial\hat{y}(X)}{\partial w_i(C_{i-1}(j_{i-1}),C_i(j_i))}\right]\right|^{2k}$$

$$+ K_k\left|\sqrt{N}\mathbb{E}_Z\left[(\partial_2\mathcal{L}(Y,\tilde{y}(X)) - \partial_2\mathcal{L}(Y,\hat{y}(X)))\frac{\partial\hat{y}(X)}{\partial w_i(C_{i-1}(j_{i-1}),C_i(j_i))}\right]\right|^{2k}$$

$$\leq K_k\|\tilde{w}_{i+1}(j_i)\|_{2k}^{2k}(1+\|\tilde{w}_i(j_i)\|_{2k}^{2k})\|\mathbf{R}\|_{2k}^{2k}\prod_{j\leq L}(1+\|\tilde{w}_j\|_{4k}^{4k})$$

$$+K_k\|\tilde{w}_{i+1}(j_i)\|_{2k}^{2k}\|\mathbf{R}_i(j_i)\|_{2k}^{2k}\prod_{j\leq L}(1+\|\tilde{w}_j\|_{2k}^{2k})+K_k\|\tilde{w}_{i+1}(j_i)\|_{2k}^{2k}\sum_{j\geq i+1}\mathbb{E}_{J_j}[\|\mathbf{R}_j(J_j)\|_2^{2k}\|\tilde{w}_{j+1}(J_j)\|_{2k}^{2k}]\prod_{j'\leq L}(1+\|\tilde{w}_{j'}\|_{2k}^{2k})$$

$$+K_k\|\tilde{w}_{i+1}(j_i)\|_{2k}^{2k}\|\mathbf{R}_{i+1}(j_i)\|_2^{2k}\prod_{j\geq i+2}(1+\|\tilde{w}_j\|_{2k}^{2k})+K_k\|\mathbf{R}_{i-1}(t,j_{i-1})\|_2^{2k}$$

$$+K_k\|\tilde{w}_{i+1}(j_i)\|_{2k}^{2k}\|\mathbf{R}\|_{2k}^{2k}(1+\|\tilde{w}_{i-1}(j_{i-1})\|_2^{2k})\prod_{j=1}^L(1+\|\tilde{w}_j\|_2^{2k})+A_k^{2k}(t,j_{i-1},j_i),$$

where $A_k(t,j_{i-1},j_i)$ does not depend on $\mathbf{R}$ and satisfies the moment bounds in Lemma 25, i.e. $\mathbb{E}\mathbb{E}_J[A_k^{2k}(t,J_{i-1},J_i)]\leq K_{T,k}$. For each $k$, consider the ODE, initialized at zero:

$$\partial_t X_i(t,j_{i-1},j_i)=K_k F[X](t,j_{i-1},j_i)+A_k(t,j_{i-1},j_i),$$

where

$$F[X](t,j_{i-1},j_i)=\left(\|\tilde{w}_{i+1}(j_i)\|_{2k}^{2k}(1+\|\tilde{w}_i(j_i)\|_{2k}^{2k})\|X\|_{2k}^{2k}\prod_{j\leq L}(1+\|\tilde{w}_j\|_{4k}^{4k})\right)^{1/2k}$$

$$+\left(\|\tilde{w}_{i+1}(j_i)\|_{2k}^{2k}\|X_i(j_i)\|_{2k}^{2k}\prod_{j\leq L}(1+\|\tilde{w}_j\|_{2k}^{2k})\right)^{1/2k}$$

$$+\left(\|\tilde{w}_{i+1}(j_i)\|_{2k}^{2k}\sum_{j\geq i+1}\mathbb{E}_{J_j}[\|X_j(J_j)\|_{2k}^{2k}\|\tilde{w}_{j+1}(J_j)\|_{2k}^{2k}]\prod_{j'\leq L}(1+\|\tilde{w}_{j'}\|_{2k}^{2k})\right)^{1/2k}$$

$$+\left(\|\tilde{w}_{i+1}(j_i)\|_{2k}^{2k}\|X_{i+1}(j_i)\|_{2k}^{2k}\prod_{j\geq i+2}(1+\|\tilde{w}_j\|_{2k}^{2k})\right)^{1/2k}+K_k\|X_{i-1}(t,j_{i-1})\|_{2k}$$

$$+\left(\|\tilde{w}_{i+1}(j_i)\|_{2k}^{2k}\|X\|_{2k}^{2k}(1+\|\tilde{w}_{i-1}(j_{i-1})\|_2^{2k})\prod_{j=1}^L(1+\|\tilde{w}_j\|_2^{2k})\right)^{1/2k}.$$

The key property of $X$ is that for all $t,j_{i-1},j_i$, we have:

$$|\mathbf{R}_i(t,j_{i-1},j_i)|\leq|X_i(t,j_{i-1},j_i)|.$$

For $B>0$, define

$$\partial_t X_i^B(t,j_{i-1},j_i)=K_k F[X^B](t,j_{i-1},j_i)\mathbb{B}^B(t,j_{i-1},j_i)+A_k(t,j_{i-1},j_i).$$

We have the estimate:

$$\partial_t\|X^B(t)\|_{2k}^{2k}\leq K_k\prod_{j'\leq L}(1+\|\tilde{w}_{j'}^B(t)\|_{4k}^{4k})(1+B^{2k})\|X^B(t)\|_{2k}^{2k}+\|A_k(t)\|_{2k}^{2k}.$$

Here, we denote $\|\tilde{w}^B\|_p=\min(B,\|\tilde{w}\|_p)$. Furthermore, for solutions $X^B,\ X'^B$ to the equation, by the triangle inequality, we have

$$\left|\partial_t\left(X^B-X'^B\right)(t,j_{i-1},j_i)\right|\leq K_k F[(X^B-X'^B)](t,j_{i-1},j_i)\mathbb{B}^B(t,j_{i-1},j_i).$$

Existence and uniqueness of $X^B$ follows easily from the above Lipschitz estimate, and we furthermore have the bound

$$\|X^B(t)\|_{2k}\leq K_k\exp(K_k B\prod_{j'\leq L}(1+\sup_{s\leq t}\|\tilde{w}_{j'}^B(t)\|_{4k}^{4k})^{1/2k}t)\cdot\|A_k(t)\|_{2k},$$

and similar to the proof of Theorem 21, the bounds (for sufficiently large $B$)

$$\|X_i^B(t,j_{i-1})\|_{2k}\leq K(1+\|\tilde{w}_i(t,j_{i-1})\|_{2k}^{2k})(1+\|\tilde{w}_{i-1}(t,j_{i-1})\|_{2k}^{2k})\exp\left(K_k B\prod_{j'\leq L}(1+\sup_{s\leq t}\|\tilde{w}_{j'}^B(t)\|_{4k}^{4k})^{1/2k}t\right)\|A_k(t)\|_{2k},$$

$$\|X_i^B(t,j_i)\|_{2k}\leq K(1+\|\tilde{w}_i(t,j_i)\|_{2k}^{2k})(1+\|\tilde{w}_{i+1}(t,j_i)\|_{2k}^{2k})\exp\left(K_k B\prod_{j'\leq L}(1+\sup_{s\leq t}\|\tilde{w}_{j'}^B(t)\|_{4k}^{4k})^{1/2k}t\right)\|A_k(t)\|_{2k}.$$

Notice that $X^B$ is positive and monotonically increasing in $B$. For $B' > B$ and $\mathbb{D}_{B,B'}(t, j_{i-1}, j_i) = \mathbb{I}(\mathbb{B}^{B'}(t, j_{i-1}, j_i) = 1, \mathbb{B}^B(t, j_{i-1}, j_i) = 0)$, we have for $\Delta X = X^{B'} - X^B$:

$$\partial_t \mathbb{E}_J[|\Delta X(t, J_{i-1}, J_i)|^{2k} \mathbb{D}_{B,B'}(t, J_{i-1}, J_i)]$$

$$\leq K_k \left( \mathbb{E}_J[|\Delta X(t, J_{i-1}, J_i)|^{2k} \mathbb{D}_{B,B'}(t, J_{i-1}, J_i)] \right)^{(2k-1)/(2k)}$$

$$\times \left[ \left( \prod_{j' \leq L} (1 + \|\tilde{w}_{j'}(t)\|_{4k}^{4k})(1 + B'^{2k}) \|X^{B'}(t)\|_{2k}^{2k} + \|A_k(t)\|_{2k}^{2k} \right) \mathbb{E}_J[\mathbb{D}_{B,B'}(t, J_{i-1}, J_i)] \right.$$

$$+ K_k \mathbb{E}_J \left[ \prod_{j' \leq L} (1 + \|\tilde{w}_{j'}(t)\|_{4k}^{4k})(1 + B'^{2k}) \|X^{B'}(t)\|_{2k}^{2k} \right.$$

$$\left. \left. \times \left( \|X_i^{B'}(t, J_i)\|_{2k}^{2k} + \|X_{i+1}^{B'}(t, J_i)\|_{2k}^{2k} + \|X_{i-1}^{B'}(t, J_{i-1})\|_{2k}^{2k} \right) \mathbb{D}_{B,B'}(t, J_{i-1}, J_i) \right] \right]^{1/(2k)},$$

while

$$\partial_t \mathbb{E}[|\Delta X(t, J_{i-1}, J_i)|^{2k} \mathbb{B}^B(t, J_{i-1}, J_i)] \leq K_k \left[ \prod_{j' \leq L} (1 + \|\tilde{w}_{j'}(t)\|_{4k}^{4k})(1 + B'^{2k}) \right] \mathbb{E}[|\Delta X(t, J_{i-1}, J_i)|^{2k} \mathbb{B}^B(t, J_{i-1}, J_i)],$$

and if $\mathbb{B}^{B'}(t, \cdot) = 0$ then $\partial_t \Delta X(t, \cdot) = 0$. Thus,

$$\partial_t \|\Delta X(t)\|_{2k}^{2k} \leq K_k \left[ \prod_{j' \leq L} (1 + \|\tilde{w}_{j'}(t)\|_{4k}^{4k})(1 + B'^{2k}) \right]^{1/2k} \|\Delta X(t)\|_{2k}^{2k}$$

$$+ K_k \mathbb{E}[\mathbb{D}_{B,B'}(t, J_{i-1}, J_i)] \exp \left( K_T B' \prod_{j' \leq L} (1 + \sup_{s \leq t} \|\tilde{w}_{j'}^{B'}(s)\|_{4k}^{4k})^{1/2k} t \right).$$

This gives:

$$\|\Delta X(t)\|_{2k}^{2k} \leq K_k \int_0^t \mathbb{E}[\mathbb{D}_{B,B'}(s, J_{i-1}, J_i)] \exp(K_T B' \prod_{j' \leq L} (1 + \sup_{s \leq t} \|\tilde{w}_{j'}^{B'}(s)\|_{4k}^{4k})^{1/2k}) ds.$$

Therefore,

$$\mathbf{E} \|\Delta X(t)\|_{2k}^{2k} \leq K_{k,T} \left( \int_0^t \mathbf{E} \mathbb{E}[\mathbb{D}_{B,B'}(s, J_{i-1}, J_i)] ds \right)^{1/2} \mathbf{E} \left[ \exp(K_T B' \prod_{j' \leq L} (1 + \sup_{s \leq t} \|\tilde{w}_{j'}^{B'}(s)\|_{4k}^{4k})^{1/2k}) \right]^{1/2}.$$

Notice that

$$\exp \left( K_T B' \prod_{j' \leq L} (1 + \sup_{s \leq t} \|\tilde{w}_{j'}^{B'}(s)\|_{4k}^{4k})^{1/2k} \right) \leq \exp \left( K_T B' \prod_{j' \leq L} (1 + \|\tilde{w}_{j'}^{B'}(0)\|_{Kk}^{Kk})^{1/2k} \right).$$

Furthermore Lemma 23 implies that

$$\left( \int_0^t \mathbf{E} \mathbb{E}[\mathbb{D}_{B,B'}(s, J_{i-1}, J_i)] ds \right)^{1/2} \leq \exp(-K_T B^2),$$

for all $B \leq N^{1/16}$, while

$$\mathbf{E} \left[ \exp \left( K_T B' \prod_{j' \leq L} (1 + \|\tilde{w}_{j'}^{B'}(0)\|_{Kk}^{Kk})^{1/2k} \right) \right] \leq \mathbf{E} \left[ \exp \left( K_T B'(1 + \sum_{j' \leq L} \|\tilde{w}_{j'}^{B'}(0)\|_{Kk}^{K}) \right) \right].$$

Using Remark 24, and $\|\tilde{w}_{j'}^{B'}(0)\|_{Kk}^{K} \leq B'^K$, we have that if $N \geq (K_T B'^C)^{16}$ for some sufficiently large constant $C > 0$, then

$$\mathbf{E} \left[ \exp \left( K_T B' \prod_{j' \leq L} (1 + \|\tilde{w}_{j'}^{B'}(0)\|_{Kk}^{Kk})^{1/2k} \right) \right] \leq \exp(K_T B').$$

Combining the estimates, we obtain that for all $B \leq K_T N^c$ for a sufficiently small constant $c > 0$,

$$\mathbf{E} \|X^B\|_{2k}^{2k} \leq \mathbf{E} \|X^{K_T}\|_{2k}^{2k} + K_k \int_{K_T}^B \exp(K_T x) \exp(-Kx^2) dx \leq K_T.$$

Finally, for $\gamma > 0$ consider the event $\mathcal{E}$ that $\max_i \max_{j_i} \max(\|\tilde{w}_i(j_i)\|_{Kk}, \|\tilde{w}_{i+1}(j_i)\|_{Kk}, \|\mathbf{w}_i(j_i)\|_{Kk}, \|\mathbf{w}_{i+1}(j_i)\|_{Kk}) \leq \gamma$. This event has probability at least $1 - KN\exp(-K_k\gamma)$ for $\gamma \leq N^{c_k}$ with sufficiently small $c_k$. We have for $B = K_T N^c$ that

$$\mathbf{E}\left[\left|\|X\|_{2k}^{2k} - \|X^B\|_{2k}^{2k}\right|\mathbb{I}(\mathcal{E})\right] \leq \exp(-KB^2)\exp(K_T\gamma^K).$$

For $\gamma = N^\epsilon$ with $\epsilon$ sufficiently small compared to $c$ and $c_k$, we then have

$$\mathbf{E}\left[\left|\|X\|_{2k}^{2k} - \|X^B\|_{2k}^{2k}\right|\mathbb{I}(\mathcal{E})\right] \leq 1.$$

This immediately implies

$$\mathbf{E}\left[\|\mathbf{R}\|_{2k}^{2k}\mathbb{I}(\mathcal{E})\right] \leq \mathbf{E}\left[\|X\|_{2k}^{2k}\mathbb{I}(\mathcal{E})\right] \leq K_T.$$

On the other hand,

$$\mathbf{E}\left[\|\mathbf{R}\|_{2k}^{2k}(1 - \mathbb{I}(\mathcal{E}))\right] \leq \mathbf{E}[\|\mathbf{R}\|_{2k}^{4k}]^{1/2}(1 - \mathbf{P}(\mathcal{E}))^{1/2}.$$

Recall the definition of $\mathbf{R}$, and noting that the moments of $\mathbf{w}$ and $w$ are finite by Lemma 23, we have the trivial bound

$$\mathbf{E}[\|\mathbf{R}\|_{2k}^{4k}]^{1/2} \leq N^{K_k}K_T.$$

Thus

$$\mathbf{E}\left[\|\mathbf{R}\|_{2k}^{2k}(1 - \mathbb{I}(\mathcal{E}))\right] \leq N^{K_k}K_T\exp(-c_kN^{2\epsilon}) < 1.$$

We can thus conclude that

$$\mathbf{E}[\|\mathbf{R}\|_{2k}^{2k}] < K_{k,T}.$$

$\square$

## D.3   Comparison bounds and concentration estimates

Let us recall the notation $\mathbf{O}_T(\cdot)$ from Appendix B.

**Lemma 28.** *We have for any $t \leq T$,*

$$\mathbf{E}\mathbb{E}_J\left[\left(\sqrt{N}(\mathbf{H}_i(t, J_i, x) - H_i(t, C_i(J_i), x))\right)^{2k}\right] \leq K_{k,T},$$

$$\mathbf{E}\mathbb{E}_J\left[\left(\sqrt{N}\left(\frac{\partial\hat{\mathbf{y}}(t,x)}{\partial\mathbf{H}_i(J_i)} - \frac{\partial\hat{y}(t,x)}{\partial H_i(C_i(J_i))}\right)\right)^{2k}\right] \leq K_{k,T},$$

$$\mathbf{E}\mathbb{E}_J\left[\left(\sqrt{N}\left(\frac{\partial\hat{\mathbf{y}}(t,x)}{\partial\mathbf{w}_i(J_{i-1},J_i)} - \frac{\partial\hat{y}(t,x)}{\partial w_i(C_{i-1}(J_{i-1}),C_i(J_i))}\right)\right)^{2k}\right] \leq K_{k,T},$$

$$\mathbf{E}\left[\left(\sqrt{N}(\partial_2\mathcal{L}(y,\hat{\mathbf{y}}(t,x)) - \partial_2\mathcal{L}(y,\hat{y}(t,x)))\right)^{2k}\right] \leq K_{k,T}.$$

*Proof.* This is a consequence of Lemmas 25, 26, 27 and 23. $\square$

**Lemma 29.** *We have for $t \leq T$,*

$$\sqrt{N}\left(\partial_2\mathcal{L}(y,\hat{\mathbf{y}}(t,x)) - \partial_2\mathcal{L}(y,\hat{y}(t,x))\right) = \sqrt{N}(\hat{\mathbf{y}}(t,x) - \hat{y}(t,x))\partial_2^2\mathcal{L}(y,\hat{y}(t,x)) + \mathbf{O}_T(1/\sqrt{N}).$$

*Proof.* We have

$$\sqrt{N}\left(\partial_2\mathcal{L}(y,\hat{\mathbf{y}}(t,x)) - \partial_2\mathcal{L}(y,\hat{y}(t,x))\right) = \sqrt{N}(\hat{\mathbf{y}}(t,x) - \hat{y}(t,x))\partial_2^2\mathcal{L}(y,y'(t,x)),$$

for some $y'(t,x)$ between $\hat{y}(t,x)$ and $\hat{\mathbf{y}}(t,x)$. Hence,

$$\mathbf{E}\mathbb{E}_Z\left[\left(\sqrt{N}(\partial_2\mathcal{L}(Y,\hat{\mathbf{y}}(t,X)) - \partial_2\mathcal{L}(Y,\hat{y}(t,X))) - \sqrt{N}(\hat{\mathbf{y}}(t,X) - \hat{y}(t,X))\partial_2^2\mathcal{L}(Y,\hat{y}(t,X))\right)^2\right]$$

$$\leq K\mathbf{E}\mathbb{E}_X\left[N(\hat{\mathbf{y}}(t,X) - \hat{y}(t,X))^2 \cdot (\hat{\mathbf{y}}(t,X) - \hat{y}(t,X))^2\right]$$

$$\leq \frac{K}{N}\mathbf{E}\mathbb{E}_X\left[\left(\sqrt{N}(\hat{\mathbf{y}}(t,X) - \hat{y}(t,X))\right)^4\right]$$

$$\leq \frac{K_T}{N},$$

where the last step follows easily from Lemma 28. $\square$

**Lemma 30.** *We have for $t \leq T$,*

$$\sqrt{N}\left(\hat{\mathbf{y}}(t,x) - \tilde{y}(t,x)\right) = \sum_{i=1}^{L} \mathbb{E}_J\left[\mathbf{R}_i(t, J_{i-1}, J_i)\frac{\partial \hat{y}(t,x)}{\partial w_i(C_{i-1}(J_{i-1}), C_i(J_i))}\right] + \mathbf{O}_T(1/\sqrt{N}).$$

*Proof.* Following the same argument as Lemma 29, we get:

$$\sqrt{N}\left(\hat{\mathbf{y}}(t,x) - \tilde{y}(t,x)\right) = \sum_{i=1}^{L} \mathbb{E}_J\left[\mathbf{R}_i(t, J_{i-1}, J_i)\frac{\partial \hat{\mathbf{y}}(t,x)}{\partial \mathbf{w}_i(C_{i-1}(J_{i-1}), C_i(J_i))}\right] + \mathbf{O}_T(1/\sqrt{N}).$$

Again the same argument applied to each of the summands leads to:

$$\mathbf{E}\left[\left(\mathbb{E}_J\left[\mathbf{R}_i(t, J_{i-1}, J_i)\frac{\partial \hat{\mathbf{y}}(t,x)}{\partial \mathbf{w}_i(C_{i-1}(J_{i-1}), C_i(J_i))}\right] - \mathbb{E}_J\left[\mathbf{R}_i(t, J_{i-1}, J_i)\frac{\partial \hat{y}(t,x)}{\partial w_i(C_{i-1}(J_{i-1}), C_i(J_i))}\right]\right)^2\right]$$

$$\leq \frac{1}{N}\left(\mathbf{E}\mathbb{E}_J\left[\mathbf{R}_i(t, J_{i-1}, J_i)^4\right]\right)^{1/2}\left(\mathbf{E}\mathbb{E}_J\left[\left(\sqrt{N}\left(\frac{\partial \hat{\mathbf{y}}(t,x)}{\partial \mathbf{w}_i(C_{i-1}(J_{i-1}), C_i(J_i))} - \frac{\partial \hat{y}(t,x)}{\partial w_i(C_{i-1}(J_{i-1}), C_i(J_i))}\right)\right)^4\right]\right)^{1/2}$$

$$\leq K_T/N,$$

by Lemmas 27 and 28. $\square$

**Lemma 31.** *We have for $t \leq T$,*

$$\mathbb{E}_J\left[R_i(\tilde{G}, t, C_{i-1}(J_{i-1}), C_i(J_i))\frac{\partial \hat{y}(t,x)}{\partial w_i(C_{i-1}(J_{i-1}), C_i(J_i))}\right] = \mathbb{E}_C\left[R_i(\tilde{G}, t, C_{i-1}, C_i)\frac{\partial \hat{y}(t,x)}{\partial w_i(C_{i-1}, C_i)}\right] + \mathbf{O}_T(N^{-1/2}).$$

*Proof.* We verify the condition of Lemma 12 for $\mathbf{i} = (i-1, i)$, $\mathbf{i}' = \emptyset$, $S = \{C_k(j_k) : j_k \in [N_k], \ k \in [L]\}$ and

$$f(S, c_{i-1}, c_i) = R_i(\tilde{G}, t, c_{i-1}, c_i)\frac{\partial \hat{y}(t,x)}{\partial w_i(c_{i-1}, c_i)}.$$

We have from Lemma 22 and Theorem 18:

$$\mathbf{E}\left[\mathbb{E}_C\left[\left|R_i(\tilde{G}, t, C_{i-1}, C_i)\right|^4\right]\right] \leq K_T,$$

and therefore, with Lemma 11,

$$\mathbf{E}\left[\left|f(S^{j_\mathbf{i}}, C_\mathbf{i}(j_\mathbf{i}))\right|^2\right] \leq K_T.$$

Let $\tilde{G}^{j_\mathbf{i}}$ be obtained similar to $\tilde{G}$ but with $C_\mathbf{i}(j_\mathbf{i})$ replaced by an independent copy $C_\mathbf{i}'(j_\mathbf{i})$. It remains to bound

$$\mathbf{E}\left[\left|f(S, C_\mathbf{i}(j_\mathbf{i})) - f(S^{j_\mathbf{i}}, C_\mathbf{i}(j_\mathbf{i}))\right|^2\right] = \mathbf{E}\left[\left|R_i(\tilde{G}, t, C_\mathbf{i}(j_\mathbf{i}))\frac{\partial \hat{y}(t,x)}{\partial w_i(C_\mathbf{i}(j_\mathbf{i}))} - R_i(\tilde{G}^{j_\mathbf{i}}, t, C_\mathbf{i}(j_\mathbf{i}))\frac{\partial \hat{y}(t,x)}{\partial w_i(C_\mathbf{i}(j_\mathbf{i}))}\right|^2\right],$$

which – similar to the previous second moment bound – amounts to proving

$$\mathbf{E}\left[\left|R_i(\tilde{G}, t, C_\mathbf{i}(j_\mathbf{i})) - R_i(\tilde{G}^{j_\mathbf{i}}, t, C_\mathbf{i}(j_\mathbf{i}))\right|^4\right] \leq K_T/N^2.$$

The proof is complete upon having this bound.

By linearity from Theorem 18,

$$R_i(\tilde{G}, t, c_{i-1}, c_i) - R_i(\tilde{G}^{j_\mathbf{i}}, t, c_{i-1}, c_i) = R_i(\tilde{G} - \tilde{G}^{j_\mathbf{i}}, t, c_{i-1}, c_i),$$

and we also have:

$$-\partial_t R_i(\tilde{G} - \tilde{G}^{j_\mathbf{i}}, t, c_{i-1}, c_i)$$

$$= \mathbb{E}_Z\left[\frac{\partial \hat{y}(t,X)}{\partial w_i(c_{i-1}, c_i)} \cdot \partial_2^2 \mathcal{L}(Y, \hat{y}(t,X)) \cdot \left((\tilde{G} - \tilde{G}^{j_\mathbf{i}})^y(t,X) + \sum_{j=1}^{L} \mathbb{E}_{C'_{j-1}, C'_j}\left\{R_j(\tilde{G} - \tilde{G}^{j_\mathbf{i}}, t, C'_{j-1}, C'_j)\frac{\partial \hat{y}(t,X)}{\partial w'_j(C'_{j-1}, C'_j)}\right\}\right)\right]$$

$$+ \mathbb{E}_Z \left[ \partial_2 \mathcal{L}(Y, \hat{y}(t, X)) \cdot \left( (\tilde{G} - \tilde{G}^{j_i})_i^w(t, c_{i-1}, c_i, X) + \sum_{j=1}^{L} \mathbb{E}_{C'_{j-1}, C'_j} \left\{ R_j(\tilde{G} - \tilde{G}^{j_i}, t, C'_{j-1}, C'_j) \frac{\partial^2 \hat{y}(t, X)}{\partial w_j(C'_{j-1}, C'_j) \partial w_i(c_{i-1}, c_i)} \right\} \right) \right]$$

$$+ \mathbb{E}_Z \left[ \partial_2 \mathcal{L}(Y, \hat{y}(t, X)) \cdot \mathbb{E}_{C'_{i-1}} \left\{ R_i(\tilde{G} - \tilde{G}^{j_i}, t, C'_{i-1}, c_i) \frac{\partial^2 \hat{y}(t, X)}{\partial_* w_i(C'_{i-1}, c_i) \partial w_i(c_{i-1}, c_i)} \right\} \right]$$

$$+ \mathbb{E}_Z \left[ \partial_2 \mathcal{L}(Y, \hat{y}(t, X)) \cdot \mathbb{E}_{C'_{i+1}} \left\{ R_i(\tilde{G} - \tilde{G}^{j_i}, t, c_i, C'_{i+1}) \frac{\partial^2 \hat{y}(t, X)}{\partial_* w_{i+1}(c_i, C'_{i+1}) \partial w_i(c_{i-1}, c_i)} \right\} \right]$$

$$+ \mathbb{E}_Z \left[ \partial_2 \mathcal{L}(Y, \hat{y}(t, X)) \cdot \mathbb{E}_{C'_{i-2}} \left\{ R_i(\tilde{G} - \tilde{G}^{j_i}, t, C'_{i-2}, c_{i-1}) \frac{\partial^2 \hat{y}(t, X)}{\partial_* w_{i-1}(C'_{i-2}, c_{i-1}) \partial w_i(c_{i-1}, c_i)} \right\} \right].$$

Note that

$$\tilde{G}_j^{\tilde{H}} - \tilde{G}_j^{j_i, \tilde{H}} = 0, \qquad \forall j \leq i - 1,$$

$$(\tilde{G}_i^{\tilde{H}} - \tilde{G}_i^{j_i, \tilde{H}})(t, c_i, x) = \frac{1}{\sqrt{N}} \left( w_i(t, C_{i-1}(j_{i-1}), c_i) \varphi_{i-1} \left( \tilde{H}_{i-1}(t, C_{i-1}(j_{i-1}), x) \right) \right.$$

$$\left. - w_i(t, C'_{i-1}(j_{i-1}), c_i) \varphi_{i-1} \left( \tilde{H}_{i-1}^{j_i}(t, C'_{i-1}(j_{i-1}), x) \right) \right),$$

$$(\tilde{G}_{i+1}^{\tilde{H}} - \tilde{G}_{i+1}^{j_i, \tilde{H}})(t, c_{i+1}, x) = \frac{1}{\sqrt{N}} \left( w_{i+1}(t, C_i(j_i), c_{i+1}) \varphi_i \left( \tilde{H}_i(t, C_i(j_i), x) \right) - w_i(t, C'_i(j_i), c_{i+1}) \varphi_i \left( \tilde{H}_i^{j_i}(t, C'_i(j_i), x) \right) \right),$$

$$+ \frac{1}{\sqrt{N}} \sum_{j'_i \neq j_i} w_{i+1}(t, C_i(j'_i), c_{i+1}) \left( \varphi_i \left( \tilde{H}_i(t, C_i(j'_i), x) \right) - \varphi_i \left( \tilde{H}_i^{j_i}(t, C_i(j'_i), x) \right) \right),$$

$$(\tilde{G}_{\ell+1}^{\tilde{H}} - \tilde{G}_{\ell+1}^{j_i, \tilde{H}})(t, c_{\ell+1}, x) = \frac{1}{\sqrt{N}} \sum_{j_\ell} w_{\ell+1}(t, C_\ell(j_\ell), c_{\ell+1}) \left( \varphi_\ell \left( \tilde{H}_\ell(t, C_\ell(j_\ell), x) \right) - \varphi_\ell \left( \tilde{H}_\ell^i(t, C_\ell(j_\ell), x) \right) \right), \qquad \forall \ell \geq i + 1.$$

We bound their expected $2p$-powers. We have that for $\ell \geq i + 1$,

$$\mathbf{E}\mathbb{E}_{J_{\ell+1}} \left[ \left| (\tilde{G}_{\ell+1}^{\tilde{H}} - \tilde{G}_{\ell+1}^{j_i, \tilde{H}})(t, C_{\ell+1}(J_{\ell+1}), x) \right|^{2p} \right] \leq K_{T,p} \mathbf{E}\mathbb{E}_{J_\ell} \left[ \left| (\tilde{G}_\ell^{\tilde{H}} - \tilde{G}_\ell^{j_i, \tilde{H}})(t, C_\ell(J_\ell), x) \right|^{2p+2} \right],$$

$$\mathbf{E}\mathbb{E}_{C_{\ell+1}} \left[ \left| (\tilde{G}_{\ell+1}^{\tilde{H}} - \tilde{G}_{\ell+1}^{j_i, \tilde{H}})(t, C_{\ell+1}, x) \right|^{2p} \right] \leq K_{T,p} \mathbf{E}\mathbb{E}_{J_\ell} \left[ \left| (\tilde{G}_\ell^{\tilde{H}} - \tilde{G}_\ell^{j_i, \tilde{H}})(t, C_\ell(J_\ell), x) \right|^{2p+2} \right].$$

For $\ell = i$, by Lemma 11,

$$\mathbf{E}\mathbb{E}_{J_{i+1}} \left[ \left| (\tilde{G}_{i+1}^{\tilde{H}} - \tilde{G}_{i+1}^{j_i, \tilde{H}})(t, C_{i+1}(J_{i+1}), x) \right|^{2p} \right]$$

$$\leq K_{T,p} \mathbf{E}\mathbb{E}_{J_i} \left[ \left| (\tilde{G}_i^{\tilde{H}} - \tilde{G}_i^{j_i, \tilde{H}})(t, C_i(J_i), x) \right|^{2p+2} \right]$$

$$+ K_{T,p} \mathbf{E}\mathbb{E}_{J_{i+1}} \left[ \left| \frac{1}{\sqrt{N}} \left( w_{i+1}(t, C_i(j_i), C_{i+1}(J_{i+1})) \varphi_i \left( \tilde{H}_i(t, C_i(j_i), x) \right) - w_i(t, C'_i(j_i), C_{i+1}(J_{i+1})) \varphi_i \left( \tilde{H}_i^{j_i}(t, C'_i(j_i), x) \right) \right) \right|^{2p} \right]$$

$$\leq K_{T,p} \mathbf{E}\mathbb{E}_{J_i} \left[ \left| (\tilde{G}_i^{\tilde{H}} - \tilde{G}_i^{j_i, \tilde{H}})(t, C_i(J_i), x) \right|^{2p+2} \right] + K_{T,p} N^{-p}.$$

Similarly,

$$\mathbf{E}\mathbb{E}_{C_{i+1}} \left[ \left| (\tilde{G}_{i+1}^{\tilde{H}} - \tilde{G}_{i+1}^{j_i, \tilde{H}})(t, C_{i+1}, x) \right|^{2p} \right] \leq K K_{T,p} \mathbf{E}\mathbb{E}_{J_i} \left[ \left| (\tilde{G}_i^{\tilde{H}} - \tilde{G}_i^{j_i, \tilde{H}})(t, C_i(J_i), x) \right|^{2p+2} \right] + K_{T,p} N^{-p}.$$

Finally at $\ell = i - 1$, by the same argument:

$$\mathbf{E}\mathbb{E}_{J_i} \left[ \left| (\tilde{G}_i^{\tilde{H}} - \tilde{G}_i^{j_i, \tilde{H}})(t, C_i(J_i), x) \right|^{2p} \right] \leq K_{T,p} N^{-p},$$

$$\mathbf{E}\mathbb{E}_{C_i} \left[ \left| (\tilde{G}_i^{\tilde{H}} - \tilde{G}_i^{j_i, \tilde{H}})(t, C_i(C_i), x) \right|^{2p} \right] \leq K_{T,p} N^{-p}.$$

As such, all these expected $2p$-powers are bounded by $K_{T,p}N^{-p}$. We derive similar bounds for $\tilde{G}^{\partial\tilde{H}} - \tilde{G}^{j_{\mathbf{i}},\partial\tilde{H}}$, $\tilde{G}^w - \tilde{G}^{j_{\mathbf{i}},w}$.

The estimates in Theorem 21 imply that

$$\mathbb{E}_{C'_{j-1},C'_j}\left\{\left|R_j(\tilde{G}-\tilde{G}^{j_{\mathbf{i}}},t,C'_{j-1},C'_j)\right|^2\right\} \le \inf_B\left\{\exp(K_T B)\|\tilde{G}-\tilde{G}^{j_{\mathbf{i}}}\|_{T,2}^2 + \exp(-K_T B^2)\|\tilde{G}-\tilde{G}^{j_{\mathbf{i}}}\|_{T,4}^2\right\} \le K_T N^{-1}.$$

Similarly, letting

$$M_1(c_i) = \sup_{t\le T}\max(\|w_i(t,c_i)\|_2, \|w_{i+1}(t,c_i)\|_2, |w_i(t,C_{i-1}(j_{i-1}),c_i)|, |w_i(t,C_{i-1}(j'_{i-1}),c_i)|),$$

$$M_2(c_{i-1}) = \sup_{t\le T}\max(\|w_i(t,c_{i-1})\|_2, \|w_{i-1}(t,c_{i-1})\|_2, |w_i(t,c_{i-1},C_i(j_i))|, |w_i(t,c_{i-1},C_i(j'_i))|),$$

we have

$$\mathbb{E}_{C'_{i-1}}\left\{R_i(\tilde{G}-\tilde{G}^{j_{\mathbf{i}}},t,C'_{i-1},c_i)^2\right\} \le N^{-1}\exp(K_T M_1(c_i)),$$

$$\mathbb{E}_{C'_{i+1}}\left\{R_{i+1}(\tilde{G}-\tilde{G}^{j_{\mathbf{i}}},t,c_i,C'_{i+1})^2\right\} \le N^{-1}\exp(K_T M_2(c_i)),$$

$$\mathbb{E}_{C'_{i-2}}\left\{R_{i-1}(\tilde{G}-\tilde{G}^{j_{\mathbf{i}}},t,C'_{i-2},c_{i-1})^2\right\} \le N^{-1}\exp(K_T M_1(c_{i-1})).$$

All of these estimates give a final bound:

$$\partial_t\left|R_i(\tilde{G}-\tilde{G}^{j_{\mathbf{i}}},t,c_{i-1},c_i)\right| \le N^{-1/2}\left[K_T + \exp(K_T M_1(c_i)) + \exp(K_T M_2(c_i)) + \exp(K_T M_1(c_{i-1}))\right].$$

In particular, since it is initialized at zero, we have:

$$\left|R_i(\tilde{G}-\tilde{G}^{j_{\mathbf{i}}},t,c_{i-1},c_i)\right|^4 \le N^{-2}\left[K_T + \exp(K_T M_1(c_i)) + \exp(K_T M_2(c_i)) + \exp(K_T M_1(c_{i-1}))\right].$$

Lemma 11 implies that $\mathbf{E}\left[\exp(K_T M_1(C_i(j_i)))\right] \le K_T$ and similarly for other terms. We thus get:

$$\mathbf{E}\left[\left|R_i(\tilde{G},t,C_{\mathbf{i}}(j_{\mathbf{i}})) - R_i(\tilde{G}^{j_{\mathbf{i}}},t,C_{\mathbf{i}}(j_{\mathbf{i}}))\right|^4\right] \le K_T/N^2,$$

as desired. $\qquad\square$

**Lemma 32.** *We have for $t \le T$ and any $B > 1$,*

$$\mathbb{E}_J\left[\mathbf{R}_i(t,J_{i-1},J_i)\frac{\partial\hat{y}(t,x)}{\partial w_i(C_{i-1}(J_{i-1}),C_i(J_i))}\right]$$

$$= K_T\mathbb{E}_J\left[\left(\mathbf{R}_i(t,J_{i-1},J_i) - R_i(\tilde{G},t,C_{i-1}(J_{i-1}),C_i(J_i))\right)^2\right]^{1/2} + \mathbb{E}_C\left[R_i(\tilde{G},t,C_{i-1},C_i)\frac{\partial\hat{y}(t,x)}{\partial w_i(C_{i-1},C_i)}\right] + \mathbf{O}_T(N^{-1/2}).$$

*Proof.* We have:

$$\left|\mathbb{E}_J\left[\left(\mathbf{R}_i(t,J_{i-1},J_i) - R_i(\tilde{G},t,C_{i-1}(J_{i-1}),C_i(J_i))\right)\frac{\partial\hat{y}(t,x)}{\partial w_i(C_{i-1}(J_{i-1}),C_i(J_i))}\right]\right|^2$$

$$\le \mathbb{E}_J\left[\left(\mathbf{R}_i(t,J_{i-1},J_i) - R_i(\tilde{G},t,C_{i-1}(J_{i-1}),C_i(J_i))\right)^2\right]\mathbb{E}_J\left[\left|\frac{\partial\hat{y}(t,x)}{\partial w_i(C_{i-1}(J_{i-1}),C_i(J_i))}\right|^2\right].$$

For $i = L$:

$$\mathbb{E}_J\left[\left|\frac{\partial\hat{y}(t,x)}{\partial w_i(C_{i-1}(J_{i-1}),C_i(J_i))}\right|^2\right] \le K$$

and so the conclusion follows from Lemma 31.

For $i < L$, we have for any $B > 1$:

$$\mathbb{E}_J\left[\left|\frac{\partial\hat{y}(t,x)}{\partial w_i(C_{i-1}(J_{i-1}),C_i(J_i))}\right|^2\right] \le K\mathbb{E}_J\mathbb{E}_{C_{i+1}}\left[|w_{i+1}(C_i(J_i),C_{i+1})|^2\right]$$

$$\leq KB + K\mathbb{E}_J\mathbb{E}_{C_{i+1}}\left[\left|w_{i+1}(C_i(J_i),C_{i+1})\right|^2\right]\mathbb{I}\left(\mathbb{E}_J\mathbb{E}_{C_{i+1}}\left[\left|w_{i+1}(C_i(J_i),C_{i+1})\right|^2\right]\geq B\right).$$

Therefore, by Lemmas 11, 27 and Theorem 21:

$$\mathbf{E}\left[\left|\mathbb{E}_J\left[\left(\mathbf{R}_i(t,J_{i-1},J_i)-R_i(\tilde{G},t,C_{i-1}(J_{i-1}),C_i(J_i))\right)\frac{\partial\hat{y}(t,x)}{\partial w_i(C_{i-1}(J_{i-1}),C_i(J_i))}\right]\right|^2\right]$$

$$\leq KB\mathbf{E}\mathbb{E}_J\left[\left(\mathbf{R}_i(t,J_{i-1},J_i)-R_i(\tilde{G},t,C_{i-1}(J_{i-1}),C_i(J_i))\right)^2\right]$$

$$+K_T\mathbf{E}\left[\mathbb{I}\left(\mathbb{E}_J\mathbb{E}_{C_{i+1}}\left[\left|w_{i+1}(C_i(J_i),C_{i+1})\right|^2\right]\geq B^2\right)\right]^{1/4}\mathbf{E}\mathbb{E}_J\left[\left(\mathbf{R}_i(t,J_{i-1},J_i)\right)^4+\left(R_i(\tilde{G},t,C_{i-1}(J_{i-1}),C_i(J_i))\right)^4\right]^{1/2}$$

$$\leq KB\mathbf{E}\mathbb{E}_J\left[\left(\mathbf{R}_i(t,J_{i-1},J_i)-R_i(\tilde{G},t,C_{i-1}(J_{i-1}),C_i(J_i))\right)^2\right]+e^{-K_TBN},$$

for a suitable finite constant $B=B(T)$. The conclusion again follows from Lemma 31. $\qquad\square$

**Lemma 33.** *We have for $t\leq T$:*

$$\sqrt{N}\left(\frac{\partial\hat{\mathbf{y}}(t,x)}{\partial\mathbf{w}(j_{i-1},j_i)}-\frac{\partial\tilde{y}(t,x)}{\partial w(C_{i-1}(j_{i-1}),C_i(j_i))}\right)$$

$$=\mathbb{E}_C\left[\sum_{r=1}^{L}R_r\left(\tilde{G},t,C_{r-1},C_r\right)\frac{\partial^2\hat{y}(t,x)}{\partial w_r(C_{r-1},C_r)\,\partial w_i(C_{i-1}(j_{i-1}),C_i(j_i))}\right]$$

$$+\mathbb{E}_C\left[R_{i-1}\left(\tilde{G},t,C_{i-2},C_{i-1}(j_{i-1})\right)\frac{\partial^2\hat{y}(t,x)}{\partial_*w_{i-1}(C_{i-2},C_{i-1}(j_{i-1}))\,\partial w_i(C_{i-1}(j_{i-1}),C_i(j_i))}\right]$$

$$+\mathbb{E}_C\left[R_i\left(\tilde{G},t,C_{i-1},C_i(j_i)\right)\frac{\partial^2\hat{y}(t,x)}{\partial_*w_i(C_{i-1},C_i(j_i))\,\partial w_i(C_{i-1}(j_{i-1}),C_i(j_i))}\right]$$

$$+\mathbb{E}_C\left[R_{i+1}\left(\tilde{G},t,C_i(j_i),C_{i+1}\right)\frac{\partial^2\hat{y}(t,x)}{\partial_*w_{i+1}(C_i(j_i),C_{i+1})\,\partial w_i(C_{i-1}(j_{i-1}),C_i(j_i))}\right]$$

$$+K_T\mathbb{E}_J\left[\left(\mathbf{R}_i(t,J_{i-1},J_i)-R_i(\tilde{G},t,C_{i-1}(J_{i-1}),C_i(J_i))\right)^2\right]^{1/2}$$

$$+\mathbf{O}_T(N^{-1/2}).$$

*Proof.* By Lemma 28, we have:

$$\mathbf{E}\mathbb{E}_J\left[\left(\sqrt{N}\left(\frac{\partial\hat{\mathbf{y}}(t,x)}{\partial\mathbf{w}(J_{i-1},J_i)}-\frac{\partial\tilde{y}(t,x)}{\partial w(C_{i-1}(J_{i-1}),C_i(J_i))}\right)\right.\right.$$

$$-\sum_{j=1}^{L}\mathbb{E}_{J'}\left[\mathbf{R}_j(t,J'_{j-1},J'_j)\frac{\partial^2\hat{y}(t,x)}{\partial w_j(C_{j-1}(J'_{j-1}),C_j(J'_j))\partial w_i(C_{i-1}(J_{i-1}),C_i(J_i))}\right]$$

$$-\mathbb{E}_{J'}\left[\mathbf{R}_{i-1}\left(t,J'_{i-2},J_{i-1}\right)\frac{\partial^2\hat{y}(t,x)}{\partial_*w_{i-1}\left(C_{i-2}(J'_{i-2}),C_{i-1}(J_{i-1})\right)\partial w_i\left(C_{i-1}(J_{i-1}),C_i(J_i)\right)}\right]$$

$$-\mathbb{E}_{J'}\left[\mathbf{R}_i\left(t,J'_{i-1},J_i\right)\frac{\partial^2\hat{y}(t,x)}{\partial_*w_i\left(C_{i-1}(J'_{i-1}),C_i(J_i)\right)\partial w_i\left(C_{i-1}(J_{i-1}),C_i(J_i)\right)}\right]$$

$$\left.\left.-\mathbb{E}_{J'}\left[\mathbf{R}_{i+1}\left(t,J_i,J'_{i+1}\right)\frac{\partial^2\hat{y}(t,x)}{\partial_*w_{i+1}\left(C_i(J_i),C_{i+1}(J'_{i+1})\right)\partial w_i\left(C_{i-1}(J_{i-1}),C_i(J_i)\right)}\right]\right)^2\right]$$

$$\leq K_TN^{-1}.$$

Following the argument in Lemmas 31 and 32, we obtain:

$$\mathbb{E}_{J'}\left[\mathbf{R}_j(t,J'_{j-1},J'_j)\frac{\partial^2\hat{y}(t,x)}{\partial w_j(C_{j-1}(J'_{j-1}),C_j(J'_j))\partial w_i(C_{i-1}(J_{i-1}),C_i(J_i))}\right]$$

$$= \mathbb{E}_{C'} \left[ R_j(\tilde{G}, t, C'_{j-1}, C'_j) \frac{\partial^2 \hat{y}(t, x)}{\partial w_j(C'_{j-1}, C'_j) \partial w_i(C_{i-1}(J_{i-1}), C_i(J_i))} \right]$$

$$+ K_T \mathbb{E}_J \left[ \left( \mathbf{R}_i(t, J_{i-1}, J_i) - R_i(\tilde{G}, t, C_{i-1}(J_{i-1}), C_i(J_i)) \right)^2 \right]^{1/2} + \mathbf{O}_T(N^{-1/2}),$$

and similarly for the rest of the terms. □

## D.4 Proof of Theorem 5

*Proof of Theorem 5.* Recall that

$$\partial_t \mathbf{R}_i(t, j_{i-1}, j_i) = \sqrt{N} \left( \partial_t \mathbf{w}_i(t, j_{i-1}, j_i) - \partial_t w_i(t, C_{i-1}(j_{i-1}), C_i(j_i)) \right)$$

$$= -\mathbb{E}_Z \left[ \sqrt{N} \left( \frac{\partial \hat{\mathbf{y}}(t, X)}{\partial \mathbf{w}_i(j_{i-1}, j_i)} - \frac{\partial \hat{y}(t, X)}{\partial w_i(C_{i-1}(j_{i-1}), C_i(j_i))} \right) \partial_2 \mathcal{L}(Y, \hat{y}(t, X)) \right]$$

$$- \mathbb{E}_Z \left[ \sqrt{N} \left( \partial_2 \mathcal{L}(Y, \hat{\mathbf{y}}(t, X)) - \partial_2 \mathcal{L}(Y, \hat{y}(t, X)) \right) \frac{\partial \hat{y}(t, X)}{\partial w_i(C_{i-1}(j_{i-1}), C_i(j_i))} \right]$$

$$- \mathbb{E}_Z \left[ \sqrt{N} \left( \partial_2 \mathcal{L}(Y, \hat{\mathbf{y}}(t, X)) - \partial_2 \mathcal{L}(Y, \hat{y}(t, X)) \right) \left( \frac{\partial \hat{\mathbf{y}}(t, X)}{\partial \mathbf{w}_i(j_{i-1}, j_i)} - \frac{\partial \hat{y}(t, X)}{\partial w_i(C_{i-1}(j_{i-1}), C_i(j_i))} \right) \right]$$

$$= -\mathbb{E}_Z \left[ \sqrt{N} \left( \frac{\partial \hat{\mathbf{y}}(t, X)}{\partial \mathbf{w}_i(j_{i-1}, j_i)} - \frac{\partial \tilde{y}(t, X)}{\partial w_i(C_{i-1}(j_{i-1}), C_i(j_i))} \right) \partial_2 \mathcal{L}(Y, \hat{y}(t, X)) \right]$$

$$- \mathbb{E}_Z \left[ \tilde{G}_i^w (t, C_{i-1}(j_{i-1}), C_i(j_i), X) \partial_2 \mathcal{L}(Y, \hat{y}(t, X)) \right]$$

$$- \mathbb{E}_Z \left[ \sqrt{N} \left( \partial_2 \mathcal{L}(Y, \hat{\mathbf{y}}(t, X)) - \partial_2 \mathcal{L}(Y, \hat{y}(t, X)) \right) \frac{\partial \hat{y}(t, X)}{\partial w_i(C_{i-1}(j_{i-1}), C_i(j_i))} \right]$$

$$- \mathbb{E}_Z \left[ \sqrt{N} \left( \partial_2 \mathcal{L}(Y, \hat{\mathbf{y}}(t, X)) - \partial_2 \mathcal{L}(Y, \hat{y}(t, X)) \right) \left( \frac{\partial \hat{\mathbf{y}}(t, X)}{\partial \mathbf{w}_i(j_{i-1}, j_i)} - \frac{\partial \hat{y}(t, X)}{\partial w_i(C_{i-1}(j_{i-1}), C_i(j_i))} \right) \right].$$

Comparing with Eq. (6):

$$\partial_t \mathbf{R}_i(t, j_{i-1}, j_i) - \partial_t R_i(\tilde{G}, t, C_{i-1}(j_{i-1}), C_i(j_i)) = -\mathbb{E}_Z \left[ A_1(Z, j_{i-1}, j_i) + A_2(Z, j_{i-1}, j_i) + A_3(Z, j_{i-1}, j_i) \right],$$

in which

$$A_1(z, j_{i-1}, j_i) = \partial_2 \mathcal{L}(Y, \hat{y}(t, x)) \Bigg\{ \sqrt{N} \left( \frac{\partial \hat{\mathbf{y}}(t, x)}{\partial \mathbf{w}_i(j_{i-1}, j_i)} - \frac{\partial \tilde{y}(t, x)}{\partial w_i(C_{i-1}(j_{i-1}), C_i(j_i))} \right)$$

$$- \mathbb{E}_C \left[ \sum_{r=1}^{L} R_r \left( \tilde{G}, t, C_{r-1}, C_r \right) \frac{\partial^2 \hat{y}(t, X)}{\partial w_r(C_{r-1}, C_r) \partial w_i(C_{i-1}(j_{i-1}), C_i(j_i))} \right]$$

$$- \mathbb{E}_C \left[ R_{i-1} \left( \tilde{G}, t, C_{i-2}, C_{i-1}(j_{i-1}) \right) \frac{\partial^2 \hat{y}(t, X)}{\partial_* w_{i-1}(C_{i-2}, C_{i-1}(j_{i-1})) \partial w_i(C_{i-1}(j_{i-1}), C_i(j_i))} \right]$$

$$- \mathbb{E}_C \left[ R_i \left( \tilde{G}, t, C_{i-1}, C_i(j_i) \right) \frac{\partial^2 \hat{y}(t, X)}{\partial_* w_i(C_{i-1}, C_i(j_i)) \partial w_i(C_{i-1}(j_{i-1}), C_i(j_i))} \right]$$

$$- \mathbb{E}_C \left[ R_{i+1} \left( \tilde{G}, t, C_i(j_i), C_{i+1} \right) \frac{\partial^2 \hat{y}(t, X)}{\partial_* w_{i+1}(C_i(j_i), C_{i+1}) \partial w_i(C_{i-1}(j_{i-1}), C_i(j_i))} \right] \Bigg\},$$

$$A_2(z, j_{i-1}, j_i) = \frac{\partial \hat{y}(t, x)}{\partial w_i(C_{i-1}(j_{i-1}), C_i(j_i))} A_{2,1}(z),$$

$$A_{2,1}(z) = \sqrt{N} \left( \partial_2 \mathcal{L}(y, \hat{\mathbf{y}}(t, x)) - \partial_2 \mathcal{L}(y, \hat{y}(t, x)) \right) - \left( \sum_{i=1}^{L} \mathbb{E}_C \left[ R_i(\tilde{G}, t, C_{i-1}, C_i) \frac{\partial \hat{y}(t, x)}{\partial w_i(C_{i-1}, C_i)} \right] + \tilde{G}^y(t, x) \right) \partial_2^2 \mathcal{L}(y, \hat{y}(t, x))$$

$$A_3(z, j_{i-1}, j_i) = \sqrt{N} \left( \partial_2 \mathcal{L}(y, \hat{\mathbf{y}}(t, x)) - \partial_2 \mathcal{L}(y, \hat{y}(t, x)) \right) \left( \frac{\partial \hat{\mathbf{y}}(t, x)}{\partial \mathbf{w}_i(j_{i-1}, j_i)} - \frac{\partial \hat{y}(t, x)}{\partial w_i(C_{i-1}(j_{i-1}), C_i(j_i))} \right).$$

To analyze $A_2$, we have from Lemmas 29, 30 and 32:

$$A_{2,1}(z) = K_T \mathbb{E}_J \left[ \left( \mathbf{R}_i(t, J_{i-1}, J_i) - R_i(\tilde{G}, t, C_{i-1}(J_{i-1}), C_i(J_i)) \right)^2 \right]^{1/2} + \mathbf{O}_T(N^{-1/2}).$$

Furthermore, for any $B > 1$, by Lemma 11, Theorems 18 and 21,

$$\mathbf{E}\mathbb{E}_{Z,J}\left[\left|A_2\left(Z, J_{i-1}, J_i\right)\right|^2\right] \leq K_T \mathbf{E}\mathbb{E}_{Z,J}\left[\left|A_2\left(Z\right)\right|^2 \mathbb{E}_{C_{i+1}}\left[\left|w_{i+1}(C_i(J_i), C_{i+1})\right|^2\right]\right]$$

$$\leq K_T B^2 \mathbf{E}\mathbb{E}_Z\left[\left|A_{2,1}\left(Z\right)\right|^2\right] + K_T \mathbf{E}\mathbb{E}_Z\left[\left|A_{2,1}\left(Z\right)\right|^4\right]^{1/2} \mathbf{E}\left[\mathbb{I}\left(\mathbb{E}_{J_i}\mathbb{E}_{C_{i+1}}\left[\left|w_{i+1}(C_i(J_i), C_{i+1})\right|^2\right] \geq B^2\right)\right]^{1/2}$$

$$\leq K_T B^2 \mathbf{E}\mathbb{E}_Z\left[\left|A_{2,1}\left(Z\right)\right|^2\right] + e^{-K_T B N}.$$

Hence with a suitable choice of $B$,

$$\mathbf{E}\mathbb{E}_{Z,J}\left[\left|A_2\left(Z, J_{i-1}, J_i\right)\right|^2\right] \leq K_T \mathbb{E}_J\left[\left(\mathbf{R}_i(t, J_{i-1}, J_i) - R_i(\tilde{G}, t, C_{i-1}(J_{i-1}), C_i(J_i))\right)^2\right] + K_T N^{-1}.$$

The treatment of $A_1$ is similar via Lemma 33:

$$\mathbf{E}\mathbb{E}_{Z,J}\left[\left|A_1\left(Z, J_{i-1}, J_i\right)\right|^2\right] \leq K_T \mathbb{E}_J\left[\left(\mathbf{R}_i(t, J_{i-1}, J_i) - R_i(\tilde{G}, t, C_{i-1}(J_{i-1}), C_i(J_i))\right)^2\right] + K_T N^{-1}.$$

Finally for $A_3$, by Lemmas 28,

$$A_3 = \boldsymbol{O}_T(N^{-1/2}).$$

Therefore,

$$\partial_t \mathbf{E}\mathbb{E}_J\left[\left|\mathbf{R}_i(t, J_{i-1}, J_i) - R_i(\tilde{G}, t, C_{i-1}(J_{i-1}), C_i(J_i))\right|^2\right] \leq \mathbf{E}\mathbb{E}_J\left[\left|\partial_t \mathbf{R}_i(t, J_{i-1}, J_i) - \partial_t R_i(\tilde{G}, t, C_{i-1}(J_{i-1}), C_i(J_i))\right|^2\right]$$

$$\leq K_T \mathbb{E}_J\left[\left(\mathbf{R}_i(t, J_{i-1}, J_i) - R_i(\tilde{G}, t, C_{i-1}(J_{i-1}), C_i(J_i))\right)^2\right] + K_T N^{-1},$$

By Gronwall's inequality, we have

$$\mathbf{E}\mathbb{E}_J\left[\left(\mathbf{R}_i(t, J_{i-1}, J_i) - R_i(\tilde{G}, t, C_{i-1}(J_{i-1}), C_i(J_i))\right)^2\right] \leq \frac{K_T}{N}.$$

This completes the proof. $\qquad\square$

# E    CLT for the output fluctuation: Proof of Theorem 6

## E.1    Joint convergence in moment: Proof of Proposition 8

*Proof of Proposition 8.* In the following, let $\tilde{R}$ denote $R(\tilde{G}, \cdot)$ and – with an abuse of notations – $R$ denote $R(G, \cdot)$. By Theorem 21 — and recalling the norms defined in its statement — there exists a sequence in $B$ of processes $\tilde{R}^B$ with $\tilde{R}^B(t) = \mathfrak{L}_t^B(\tilde{G})$ where $\mathfrak{L}_t$ is a linear map for all $t \leq T$ with $\|\tilde{R}^B\|_{T,2} \leq \exp(K_T B)\|\tilde{G}\|_{T,2}$ and $\|\tilde{R}^B - \tilde{R}\|_{T,2} \leq \|\tilde{G}\|_{T,2+\epsilon}\exp(-c\epsilon B^2)$ for some $c > 0$. We have:

$$\mathbf{E}\left[\prod_j \langle f_j, \tilde{G}_{i(j)}^{\alpha_j}\rangle_{t_j}^{\beta_j} \prod_{j'} \langle h_{j'}, R_{i(j)}\rangle_{t_{j'}}^{\beta_{j'}}\right]$$

$$= \mathbf{E}\left[\prod_j \langle f_j, \tilde{G}_{i(j)}^{\alpha_j}\rangle_{t_j}^{\beta_j} \prod_{j'} \langle h_{j'}, \tilde{R}_{i(j)}^B + (\tilde{R}_{i(j')}^B - \tilde{R}_{i(j')})\rangle_{t_{j'}}^{\beta_{j'}}\right]$$

$$= \mathbf{E}\left[\prod_j \langle f_j, \tilde{G}_{i(j)}^{\alpha_j}\rangle_{t_j}^{\beta_j} \prod_{j'} \left(\langle h_{j'}, \tilde{R}_{i(j)}^B\rangle_{t_{j'}} + \|h_{j'}\|_{t_{j'}}\|\tilde{G}\|_{T,2+\epsilon}\exp(-c\epsilon B^2)O_{j'}(1)\right)^{\beta_{j'}}\right]$$

$$= \mathbf{E}\left[\prod_j \langle f_j, \tilde{G}_{i(j)}^{\alpha_j}\rangle_{t_j}^{\beta_j} \prod_{j'} \langle h_{j'}, \tilde{R}_{i(j)}^B\rangle_{t_{j'}}^{\beta_{j'}}\right] + O_D(1)\exp(-c\epsilon B^2)\prod_j \|f_j\|_{t_j}^{\beta_j}\|\tilde{G}_{i(j)}^{\alpha_j}\|_{t_j}^{\beta_j} \prod_{j'} \|h_{j'}\|_{t_{j'}}^{\beta_{j'}}\|\tilde{G}\|_{T,2+\epsilon}^{\beta_{j'}}$$

$$= \mathbf{E}\left[\prod_j \langle f_j, \tilde{G}_{i(j)}^{\alpha_j}\rangle_{t_j}^{\beta_j} \prod_{j'} \langle (\mathfrak{L}_t^B)^*(h_{j'}), \tilde{G}_{i(j')}\rangle_{t_{j'}}^{\beta_{j'}}\right] + O_D(1)\exp(-c\epsilon B^2)\prod_j \|f_j\|_{t_j}^{\beta_j}\|\tilde{G}_{i(j)}^{\alpha_j}\|_{t_j}^{\beta_j} \prod_{j'} \|h_{j'}\|_{t_{j'}}^{\beta_{j'}}\|\tilde{G}\|_{T,2+\epsilon}^{\beta_{j'}},$$

where $(\mathfrak{L}_t^B)^*$ is the adjoint of $\mathfrak{L}_t^B$, which has operator norm at most $\exp(K_T B)$. A similar bound applies to $\mathbf{E}\left[\prod_j \langle f_j, G_{i(j)}^{\alpha_j}\rangle_{t_j}^{\beta_j} \prod_{j'} \langle h_{j'}, R_{i(j')}\rangle_{t_{j'}}^{\beta_{j'}}\right]$. Since $\tilde{G}$ converges $G$-polynomially in moment to $G$, we have:

$$\left| \mathbf{E}\left[\prod_j \langle f_j, \tilde{G}_{i(j)}^{\alpha_j}\rangle_{t_j}^{\beta_j} \prod_{j'} \langle h_{j'}, \tilde{R}_{i(j)}^B\rangle_{t_{j'}}^{\beta_{j'}}\right] - \mathbf{E}\left[\prod_j \langle f_j, G_{i(j)}^{\alpha_j}\rangle_{t_j}^{\beta_j} \prod_{j'} \langle h_{j'}, R_{i(j)}^B\rangle_{t_{j'}}^{\beta_{j'}}\right]\right|$$

$$\leq N^{-1/8} O_D(\max_j \|f_j\|_{t_j}^D, \max_{j'} \|(\mathfrak{L}_t^B)^*(h_{j'})\|_{t_{j'}}^D)$$

$$\leq \exp(K_T B) N^{-1/8} O_D(\max_j \|f_j\|_{t_j}^D, \max_{j'} \|h_{j'}\|_{t_{j'}}^D).$$

This also shows

$$\|\tilde{G}_{i(j)}^{\alpha_j}\|_{t_j}^{\beta_j} \leq \|G_{i(j)}^{\alpha_j}\|_{t_j}^{\beta_j} + N^{-1/8} O_D(\max_j \|f_j\|_{t_j}^D, \max_{j'} \|h_{j'}\|_{t_{j'}}^D).$$

Then choosing $B = c_0 \sqrt{\log N}$ for a suitable constant $c_0$, for sufficiently large $N$, we have:

$$\left| \mathbf{E}\left[\prod_j \langle f_j, \tilde{G}_{i(j)}^{\alpha_j}\rangle_{t_j}^{\beta_j} \prod_{j'} \langle h_{j'}, \tilde{R}_{i(j)}\rangle_{t_{j'}}^{\beta_{j'}}\right] - \mathbf{E}\left[\prod_j \langle f_j, G_{i(j)}^{\alpha_j}\rangle_{t_j}^{\beta_j} \prod_{j'} \langle h_{j'}, R_{i(j)}\rangle_{t_{j'}}^{\beta_{j'}}\right]\right|$$

$$\leq \exp(K_T B) N^{-1/8} O_D(\max_j \|f_j\|_{t_j}^D, \max_{j'} \|h_{j'}\|_{t_{j'}}^D) + O_D(1) \exp(-c\epsilon B^2) \prod_j \|f_j\|_{t_j}^{\beta_j} \|G_{i(j)}^{\alpha_j}\|_{t_j}^{\beta_j} \prod_{j'} \|h_{j'}\|_{t_{j'}}^{\beta_{j'}} \|G\|_{T,2+\epsilon}^{\beta_{j'}}$$

$$\leq N^{-1/8+o(1)} O_D(\max_j \|f_j\|_{t_j}^D, \max_{j'} \|h_{j'}\|_{t_{j'}}^D).$$

In particular, $(R(\tilde{G}, \cdot), \tilde{G})$ converges $G$-polynomial-moment and $R$-linear-moment to $(R(G, \cdot), G)$. $\qquad\square$

### E.2 CLT for the output function: Proof of Theorem 6

*Proof of Theorem 6.* By Lemma 30, Lemma 32 and Theorem 5, we have:

$$\sqrt{N}(\hat{\mathbf{y}}(t,x) - \tilde{y}(t,x)) = \sum_{i=1}^L \mathbb{E}_C\left[R_i(\tilde{G}, t, C_{i-1}, C_i)\frac{\partial \hat{y}(t,x)}{\partial w_i(C_{i-1}, C_i)}\right] + \mathbf{O}_T(1/\sqrt{N}).$$

This implies that

$$\underbrace{\sqrt{N}(\hat{\mathbf{y}}(t,x) - \hat{y}(t,x))}_{\equiv U(z)} = \underbrace{\sum_{i=1}^L \mathbb{E}_C\left[R_i(\tilde{G}, t, C_{i-1}, C_i)\frac{\partial \hat{y}(t,x)}{\partial w_i(C_{i-1}, C_i)}\right] + \tilde{G}^y(t,x)}_{\equiv V(\tilde{G}, z)} + \mathbf{O}_T(1/\sqrt{N}).$$

For $U = U(Z)$ and $V = V(\tilde{G}, Z)$, we have for $m \geq 1$ and 1-bounded $h$,

$$|\mathbf{E}\left[\langle h, U^m\rangle_Z - \langle h, V^m\rangle_Z\right]| \leq \mathbf{E}\mathbb{E}_Z\left[|U^m - V^m|\right]$$

$$\leq K_m \mathbf{E}\mathbb{E}_Z\left[\left(|U|^{m-1} + |V|^{m-1}\right)|U - V|\right]$$

$$\leq K_m \mathbf{E}\mathbb{E}_Z\left[|U|^{2m-2} + |V|^{2m-2}\right]^{1/2} \mathbf{E}\mathbb{E}_Z\left[|U - V|^2\right]^{1/2}.$$

Recalling Lemmas 11, 28, Theorems 21 and 18, we obtain:

$$|\mathbf{E}\left[\langle h, U^m\rangle_Z - \langle h, V^m\rangle_Z\right]| \leq K_{T,m} N^{-1/2}.$$

Finally using Proposition 8 and Theorem 18, we obtain:

$$|\mathbf{E}\left[\langle h, U^m\rangle_Z - \langle h, V(\underline{G}, Z)^m\rangle_Z\right]| \leq K_{T,m} N^{-1/8+o(1)}.$$

This also proves the desired weak convergence. $\qquad\square$

# F Asymptotic variance of the output fluctuation: Proof of Theorems 9 and 10

## F.1 Variance decomposition of the limiting output fluctuation $\underline{G}^y$

The goal of this section is to show that $\underline{G}^y(t, \cdot)$ lies in the span of $\frac{\partial \hat{y}(t, \cdot)}{\partial w_i(C_{i-1}, C_i)}$. This is done by decomposing the covariance structure of $\underline{G}^y$ in a suitable layer-wise fashion.

**Proposition 34.** *Let $f$ be any square-integrable function of $X$ for which*

$$\sum_i \mathbb{E}_C \left[ \mathbb{E}_X \left[ \frac{\partial \hat{y}(t, X)}{\partial w_i(C_{i-1}, C_i)} f(X) \right]^2 \right] = 0.$$

*Then almost surely,*

$$\mathbb{E}_X \left[ \underline{G}^y(t, X) f(X) \right]^2 = 0.$$

*Proof.* Let us drop the notational dependency on $t$. Let us also write

$$\varphi_i (H_i(x, c_i)) \equiv \varphi_i (x, c_i), \qquad \varphi_i' (H_i(x, c_i)) \equiv \varphi_i' (x, c_i).$$

Note that

$$\mathbf{E} \left[ \mathbb{E}_X \left[ \underline{G}^y(X) f(X) \right]^2 \right] = \mathbb{E}_{X, X'} \left[ \mathbf{E}[\underline{G}^y(X) \underline{G}^y(X')] f(X) f(X') \right].$$

We recall from the statement of Theorem 18:

$$\underline{G}^y(x) = \varphi_L'(x) \underline{G}_L^{\tilde{H}}(x),$$

$$\underline{G}_i^{\tilde{H}}(c_i, x) = \underline{G}_i^H(c_i, x) + \mathbb{E}_{C_{i-1}} \left[ w_i(C_{i-1}, c_i) \varphi_{i-1}'(x, C_{i-1}) \underline{G}_{i-1}^{\tilde{H}}(x, C_{i-1}) \right],$$

$$\underline{G}_1^{\tilde{H}}(c_1, x) = 0,$$

which thus yield

$$\underline{G}^y(x) = \sum_{i=1}^L \varphi_L'(x) \mathbb{E}_C \left[ \left( \prod_{j=i+1}^L w_j(C_{j-1}, C_j) \varphi_{j-1}'(x, C_{j-1}) \right) \underline{G}_i^H(x, C_i) \right],$$

where we take $\prod_{j=L+1}^L = 1$. We also recall:

$$\mathbf{E}[\underline{G}_i^H(x, c_i) \underline{G}_i^H(x', c_i')] = \mathbb{C}_{C_{i-1}} \left[ w_i(C_{i-1}, c_i) \varphi_{i-1}(x, C_{i-1}) ; \ w_i(C_{i-1}, c_i') \varphi_{i-1}(x', C_{i-1}) \right],$$

and for $i \neq j$,

$$\mathbf{E}[\underline{G}_i^H(x, c_i) \underline{G}_j^H(x', c_j')] = 0.$$

Combining these facts together, we obtain:

$$\mathbf{E}[\underline{G}^y(x) \underline{G}^y(x')]$$

$$= \sum_{i=1}^L \varphi_L'(x) \varphi_L'(x') \mathbb{E}_{C, C'} \left[ \prod_{j=i+1}^L w_j(C_{j-1}, C_j) \varphi_{j-1}'(x, C_{j-1}) w_j(C_{j-1}', C_j') \varphi_{j-1}'(x, C_{j-1}') \mathbb{E} \left[ \underline{G}_i^H(x, C_i) \underline{G}_i^H(x, C_i') \right] \right]$$

$$= \sum_{i=1}^L \varphi_L'(x) \varphi_L'(x') \mathbb{E}_{C_i, \ldots, C_L, C_i', \ldots, C_L'} \left[ \prod_{j=i+1}^L w_j(C_{j-1}, C_j) \varphi_{j-1}'(x, C_{j-1}) w_j(C_{j-1}', C_j') \varphi_{j-1}'(x, C_{j-1}') \right.$$

$$\times \left( \mathbb{E}_{C_{i-1}} \left[ w_i(C_{i-1}, C_i) w_i(C_{i-1}, C_i') \varphi_{i-1}(x, C_{i-1}) \varphi_{i-1}(x', C_{i-1}) \right] \right.$$

$$\left. \left. - \mathbb{E}_{C_{i-1}} \left[ w_i(C_{i-1}, C_i) \varphi_{i-1}(x, C_{i-1}) \right] \mathbb{E}_{C_{i-1}} \left[ w_i(C_{i-1}, C_i') \varphi_{i-1}(x', C_{i-1}) \right] \right) \right]$$

$$= \sum_{i=1}^L \left\{ \mathbb{E}_{C_{i-1}} \left[ \varphi_L'(x) \mathbb{E}_{C_i, \ldots, C_L} \left[ \prod_{j=i+1}^L w_j(C_{j-1}, C_j) \varphi_{j-1}'(x, C_{j-1}) \cdot w_i(C_{i-1}, C_i) \varphi_{i-1}(x, C_{i-1}) \right] \right. \right.$$

$$\left. \left. \times \varphi_L'(x') \mathbb{E}_{C_i', \ldots, C_L'} \left[ \prod_{j=i+1}^L w_j(C_{j-1}', C_j') \varphi_{j-1}'(x', C_{j-1}') \cdot w_i(C_{i-1}, C_i') \varphi_{i-1}(x', C_{i-1}) \right] \right] \right]$$

$$- \mathbb{E}_{C_{i-1}} \left[ \varphi'_L(x) \mathbb{E}_{C_i, \ldots, C_L} \left[ \prod_{j=i+1}^{L} w_j(C_{j-1}, C_j) \varphi'_{j-1}(x, C_{j-1}) \cdot w_i(C_{i-1}, C_i) \varphi_{i-1}(x, C_{i-1}) \right] \right]$$

$$\times \mathbb{E}_{C'_{i-1}} \left[ \varphi'_L(x') \mathbb{E}_{C'_i, \ldots, C'_L} \left[ \prod_{j=i+1}^{L} w_j(C'_{j-1}, C'_j) \varphi'_{j-1}(x', C'_{j-1}) \cdot w_i(C'_{i-1}, C'_i) \varphi_{i-1}(x', C'_{i-1}) \right] \right] \Bigg\}$$

$$= \sum_{i=1}^{L} \left\{ \mathbb{E}_{C_{i-1}} \left[ \mathbb{E}_{C_i} \left[ w_i(C_{i-1}, C_i) \frac{\partial \hat{y}(x)}{\partial w_i(C_{i-1}, C_i)} \right] \mathbb{E}_{C'_i} \left[ w_i(C_{i-1}, C'_i) \frac{\partial \hat{y}(x')}{\partial w_i(C_{i-1}, C'_i)} \right] \right] \right.$$

$$\left. - \mathbb{E}_{C_{i-1}} \left[ \mathbb{E}_{C_i} \left[ w_i(C_{i-1}, C_i) \frac{\partial \hat{y}(x)}{\partial w_i(C_{i-1}, C_i)} \right] \right] \mathbb{E}_{C'_{i-1}} \left[ \mathbb{E}_{C'_i} \left[ w_i(C'_{i-1}, C'_i) \frac{\partial \hat{y}(x')}{\partial w_i(C'_{i-1}, C'_i)} \right] \right] \right\}.$$

In particular,

$$\mathbb{E}_{X, X'} \left[ \mathbf{E}[\underline{G}^y(X) \underline{G}^y(X')] f(X) f(X') \right]$$

$$= \sum_{i=1}^{L} \left\{ \mathbb{E}_{C_{i-1}} \left[ \mathbb{E}_{X, C_i} \left[ w_i(C_{i-1}, C_i) \frac{\partial \hat{y}(X)}{\partial w_i(C_{i-1}, C_i)} f(X) \right]^2 \right] - \left( \mathbb{E}_{X, C_{i-1}, C_i} \left[ w_i(C_{i-1}, C_i) \frac{\partial \hat{y}(X)}{\partial w_i(C_{i-1}, C_i)} f(X) \right] \right)^2 \right\}.$$

Of course $\mathbb{E}_{X, X'} \left[ \mathbf{E}[\underline{G}^y(X) \underline{G}^y(X')] f(X) f(X') \right] \geq 0$, but we also have:

$$\mathbb{E}_{C_{i-1}} \left[ \mathbb{E}_{X, C_i} \left[ w_i(C_{i-1}, C_i) \frac{\partial \hat{y}(X)}{\partial w_i(C_{i-1}, C_i)} f(X) \right]^2 \right] \leq \mathbb{E}_{C_{i-1}} \left[ \mathbb{E}_{C_i} \left[ |w_i(C_{i-1}, C_i)|^2 \right] \mathbb{E}_{C_i} \left[ \mathbb{E}_X \left[ \frac{\partial \hat{y}(X)}{\partial w_i(C_{i-1}, C_i)} f(X) \right]^2 \right] \right] = 0.$$

Therefore it must be that

$$\mathbf{E} \left[ \mathbb{E}_X \left[ \underline{G}^y(X) f(X) \right]^2 \right] = \mathbb{E}_{X, X'} \left[ \mathbf{E}[\underline{G}^y(X) \underline{G}^y(X')] f(X) f(X') \right] = 0.$$

□

### F.2  Variance reduction under GD: Proof of Theorem 9

We prove the variance reduction effect of GD in the case of idealized initialization.

*Proof of Theorem 9.* By Theorem 6, the fluctuation $\sqrt{N}(\hat{\mathbf{y}}(t, x) - \hat{y}(t, x))$ converges in polynomial-moment to

$$\hat{G}(t, x) := \sum_{i=1}^{L} \mathbb{E}_C \left[ R_i(\underline{G}, t, C_{i-1}, C_i) \frac{\partial \hat{y}(t, x)}{\partial w_i(C_{i-1}, C_i)} \right] + \underline{G}^y(t, x),$$

where $\underline{G}$ is the limiting Gaussian process defined in Theorem 18. Furthermore, by Theorem 19,

$$\partial_t \hat{G}(t, x) = \sum_{i=1}^{L} \mathbb{E}_C \left[ \partial_t R_i(\underline{G}, t, C_{i-1}, C_i) \frac{\partial \hat{y}(t, x)}{\partial w_i(C_{i-1}, C_i)} + R_i(\underline{G}, t, C_{i-1}, C_i) \partial_t \left( \frac{\partial \hat{y}(t, x)}{\partial w_i(C_{i-1}, C_i)} \right) \right] + \partial_t \underline{G}^y(t, x).$$

By the assumption on the initialization, $\partial_t \left( \frac{\partial \hat{y}(t, x)}{\partial w_i(C_{i-1}, C_i)} \right) = 0$ and $\partial_t \underline{G}^y = 0$. Then using Eq. (6) for $\partial_t R_i(\underline{G}, t, c_{i-1}, c_i)$ and recalling that $\mathbb{E}_Z \left[ \partial_2 \mathcal{L}(Y, \hat{y}(0, X)) | X \right] = 0$, we obtain:

$$\partial_t \hat{G}(t, x) = - \sum_{i=1}^{L} \mathbb{E}_{Z, C} \left[ \frac{\partial \hat{y}(t, x)}{\partial w_i(C_{i-1}, C_i)} \frac{\partial \hat{y}(t, X)}{\partial w_i(C_{i-1}, C_i)} \cdot \partial_2^2 \mathcal{L}(Y, \hat{y}(t, X)) \right.$$

$$\times \left. \left( \underline{G}^y(t, X) + \sum_{j=1}^{L} \mathbb{E}_{C'_{j-1}, C'_j} \left[ R_j(\underline{G}, t, C'_{j-1}, C'_j) \frac{\partial \hat{y}(t, X)}{\partial w'_j(C'_{j-1}, C'_j)} \right] \right) \right]$$

$$= - \partial_2^2 \mathcal{L} \sum_{i=1}^{L} \mathbb{E}_{Z, C} \left[ \frac{\partial \hat{y}(t, x)}{\partial w_i(C_{i-1}, C_i)} \frac{\partial \hat{y}(t, X)}{\partial w_i(C_{i-1}, C_i)} \cdot \hat{G}(t, X) \right],$$

where we denote $\partial_2^2 \mathcal{L} = \mathbb{E}_Z \left[ \partial_2^2 \mathcal{L}(Y, \hat{y}(0, X)) | X \right] > 0$ a finite constant. Hence,

$$\partial_t V^*(t) = \partial_t \mathbf{E} \mathbb{E}_X \left[ \left| \hat{G}(t, X) \right|^2 \right]$$

$$= -2\partial_2^2 \mathcal{L} \mathbb{E} \mathbb{E}_{Z'} \left[ \hat{G}(t, X') \left( \sum_{i=1}^{L} \mathbb{E}_{Z,C} \left[ \frac{\partial \hat{y}(t, X')}{\partial w_i(C_{i-1}, C_i)} \frac{\partial \hat{y}(t, X)}{\partial w_i(C_{i-1}, C_i)} \cdot \hat{G}(t, X) \right] \right) \right]$$

$$= -2\partial_2^2 \mathcal{L} \sum_{i=1}^{L} \mathbb{E} \mathbb{E}_C \left[ \mathbb{E}_Z \left[ \frac{\partial \hat{y}(t, X)}{\partial w_i(C_{i-1}, C_i)} \hat{G}(t, X) \right]^2 \right],$$

which is non-negative. This completes the proof of the first part.

Next, since we initialize the MF limit at a stationary point, we have

$$\hat{y}(t, x) = \hat{y}(x), \qquad \frac{\partial \hat{y}(t, x)}{\partial w_i(c_{i-1}, c_i)} = \frac{\partial \hat{y}(x)}{\partial w_i(c_{i-1}, c_i)},$$

independent of time $t$. Let $\mathfrak{A} : L^2(\mathcal{P}) \to L^2(\mathcal{P})$ be the linear operator defined by

$$(\mathfrak{A}f)(x) = -\sum_i \mathbb{E}_{Z,C} \left[ \frac{\partial \hat{y}(x)}{\partial w_i(C_{i-1}, C_i)} \frac{\partial \hat{y}(X)}{\partial w_i(C_{i-1}, C_i)} f(X) \right].$$

As shown – and assuming $\partial_2^2 \mathcal{L} = 1$ for simplicity – we can write:

$$\partial_t \hat{G}(t, x) = (\mathfrak{A}\hat{G}(t, \cdot))(x).$$

Note that $\mathfrak{A}$ is self-adjoint:

$$\mathbb{E}\left[ g(X)(\mathfrak{A}f)(X) \right] = -\mathbb{E}_{Z,Z',C} \left[ g(X)f(X') \sum_i \frac{\partial \hat{y}(X)}{\partial w_i(C_{i-1}, C_i)} \frac{\partial \hat{y}(X')}{\partial w_i(C_{i-1}, C_i)} \right] = \mathbb{E}\left[ (\mathfrak{A}g)(X)f(X) \right],$$

and $\mathfrak{A}$ is compact, as it is a finite sum of integral operators with Hilbert-Schmidt kernels. By the spectral theorem, the orthogonal complement of $\ker(\mathfrak{A})$ has a countable orthonormal basis consisting of eigenfunctions of $\mathfrak{A}$. Note that $\ker(\mathfrak{A})$ is a closed subspace of $L^2(\mathcal{P})$ which is separable, and hence it is also separable. Similarly, the orthogonal complement of $\ker(\mathfrak{A})$ is a separable subspace of $L^2(\mathcal{P})$. For any eigenfunction $\phi$ of $\mathfrak{A}$ with eigenvalue $\lambda$, we have

$$(\mathfrak{A}\phi)(x) = -\sum_i \frac{\partial \hat{y}(x)}{\partial w_i(C_{i-1}, C_i)} \mathbb{E}_{Z,C} \left[ \frac{\partial \hat{y}(X)}{\partial w_i(C_{i-1}, C_i)} \phi(X) \right] = \lambda \phi(x).$$

Thus,

$$\lambda = \mathbb{E}[\phi(x)(\mathfrak{A}\phi)(x)] = -\sum_i \mathbb{E}_C \left[ \mathbb{E}_X \left[ \frac{\partial \hat{y}(X)}{\partial w_i(C_{i-1}, C_i)} \phi(X) \right]^2 \right].$$

Hence, all eigenvalues of $\mathfrak{A}$ are non-positive. Note that $\mathfrak{A}f = 0$ if and only if

$$\sum_i \mathbb{E}_C \left[ \mathbb{E}_X \left[ \frac{\partial \hat{y}(X)}{\partial w_i(C_{i-1}, C_i)} f(X) \right]^2 \right] = 0.$$

By Proposition 34, almost surely, $\underline{G}^y \in \ker(\mathfrak{A})^\perp$ (noting that $\ker(\mathfrak{A})$ is separable). Hence so is $\hat{G}(t, \cdot)$. For $f \in \ker(\mathfrak{A})^\perp$, the solution to the equation $\partial_t f_t(x) = (\mathfrak{A}f_t)(x)$ is given as

$$f_t(x) = \exp(t\mathfrak{A})f_0(x).$$

Let $f_0 = \sum_j \tau_j \phi_j$ where $\phi_j$ are eigenfunctions of $\mathfrak{A}$ in $\mathcal{H}$ with eigenvalue $\lambda_j < 0$, we have

$$f_t(x) = \sum_j \exp(t\lambda_j)\tau_j \phi_j(x).$$

In particular, since $\sum_j \tau_j^2 = \|f_0\|_{L^2(\mathcal{P})}^2$ is finite, as $t \to \infty$,

$$\mathbb{E}_X \left[ |f_t(X)|^2 \right] = \sum_j \exp(2t\lambda_j)\tau_j^2 \to 0.$$

This proves the second part of the theorem. $\qquad \square$

### F.3 Variance reduction under GD: Proof of Theorem 10

We prove the variance reduction effect of GD in the case of sufficiently fast convergence to global optima. In particular, we recall from Theorem 10 that we assume for some $\eta > 0$,

$$\int_0^\infty s^{2+\eta} \mathbb{E}[\partial_2 \mathcal{L}(Y, \hat{y}(s, X))^2] ds < \infty. \tag{8}$$

The dependency on time will be important in this section, therefore we remind that $K_T$ denotes a constant depending on time, and $K$ denotes a constant that does not.

**Lemma 35.** *If $\int_0^\infty \left( \mathbb{E}[\partial_2 \mathcal{L}(Y, \hat{y}(t, X))^2] \right)^{1/2} dt < \infty$, then for any $p \geq 1$, $\sup_{t \geq 0} \sum_i \mathbb{E}_C[w_i(t, C_{i-1}, C_i)^{2p}] \leq (Kp)^p$.*

*Proof.* We have

$$\partial_t w_i(t, C_{i-1}, C_i) = \mathbb{E}_Z \left[ \partial_2 \mathcal{L}(Y, \hat{y}(t, X)) \frac{\partial \hat{y}(t, X)}{\partial w_i(C_{i-1}, C_i)} \right],$$

and hence

$$\mathbb{E}_C \left[ |\partial_t w_i(t, C_{i-1}, C_i)|^{2p} \right] \leq \mathbb{E}_C \left[ \mathbb{E}_Z \left[ \partial_2 \mathcal{L}(Y, \hat{y}(t, X))^2 \right]^p \mathbb{E}_Z \left[ \left( \frac{\partial \hat{y}(t, X)}{\partial w_i(C_{i-1}, C_i)} \right)^2 \right]^p \right]$$

$$\leq \mathbb{E}_Z \left[ \partial_2 \mathcal{L}(Y, \hat{y}(t, X))^2 \right]^p \mathbb{E}_{C,Z} \left[ \left( \frac{\partial \hat{y}(t, X)}{\partial w_i(C_{i-1}, C_i)} \right)^{2p} \right]$$

$$\leq \mathbb{E}_Z \left[ \partial_2 \mathcal{L}(Y, \hat{y}(t, X))^2 \right]^p \cdot K \mathbb{E}_{C_{i+1}, C_i} \left[ w_{i+1}(t, C_i, C_{i+1})^{2p} \right] \prod_{j \geq i+2} \mathbb{E}_{C_j, C_{j-1}} \left[ w_j(t, C_{j-1}, C_j)^2 \right]^p.$$

Let us consider $i > 1$; the case $i = 1$ is similar. We then have:

$$\partial_t \left( \mathbb{E}[w_i(t, C_{i-1}, C_i)^{2p}] \right) = 2p \mathbb{E}[w_i^{2p-1}(t, C_{i-1}, C_i) \partial_t w_i(t, C_{i-1}, C_i)]$$

$$\leq 2pK \left( \mathbb{E}[w_i(t, C_{i-1}, C_i)^{2p}] \right)^{(2p-1)/(2p)} \cdot \left( \mathbb{E}[\partial_t w_i(t, C_{i-1}, C_i)^{2p}] \right)^{1/(2p)}$$

$$\leq 2pK \left( \mathbb{E}[w_i(t, C_{i-1}, C_i)^{2p}] \right)^{(2p-1)/(2p)} \left( \mathbb{E}[\partial_2 \mathcal{L}(Y, \hat{y}(t, X))^2] \right)^{1/2}$$

$$\times \mathbb{E}_{C_{i+1}, C_i} \left[ w_{i+1}(t, C_i, C_{i+1})^{2p} \right]^{1/2p} \prod_{j \geq i+2} \mathbb{E}_{C_j, C_{j-1}} \left[ w_j(t, C_{j-1}, C_j)^2 \right]^{1/2}.$$

In particular, for $i = L$,

$$\partial_t \left( \mathbb{E}[w_L(t, C_{L-1}, C_L)^{2p}] \right) = 2pK \left( \mathbb{E}[w_L(t, C_{L-1}, C_L)^{2p}] \right)^{(2p-1)/(2p)} \left( \mathbb{E}[\partial_2 \mathcal{L}(Y, \hat{y}(t, X))^2] \right)^{1/2},$$

which implies that

$$\mathbb{E}[w_L(t, C_{L-1}, C_L)^{2p}]^{1/(2p)} \leq Kp^{1/2} + K \int_0^t \left( \mathbb{E}[\partial_2 \mathcal{L}(Y, \hat{y}(s, X))^2] \right)^{1/2} ds,$$

and therefore,

$$\sup_{t \geq 0} \mathbb{E}[w_L(t, C_{L-1}, C_L)^{2p}]^{1/(2p)} \leq Kp^{1/2} + K.$$

(Here we note that $\mathbb{E}[w_L(0, C_{L-1}, C_L)^{2p}]^{1/(2p)} \leq Kp^{1/2}$ by the sub-Gaussian initialization assumption.)

Next, by induction, assuming that

$$\sup_{i > h} \sup_t \mathbb{E}[w_i(t, C_{i-1}, C_i)^{2p}]^{1/(2p)} < Kp^{1/2} + K,$$

we have

$$\partial_t \left( \mathbb{E}[w_h(t, C_{h-1}, C_h)^{2p}] \right) \leq 2pK \left( \mathbb{E}[w_h(t, C_{h-1}, C_h)^{2p}] \right)^{(2p-1)/(2p)} \left( \mathbb{E}[\partial_2 \mathcal{L}(Y, \hat{y}(t, X))^2] \right)^{1/2}$$

$$\times \mathbb{E}_{C_{h+1}, C_h} \left[ w_{h+1}(t, C_h, C_{h+1})^{2p} \right]^{1/2p} \prod_{j \geq h+2} \mathbb{E}_{C_j, C_{j-1}} \left[ w_j(t, C_{j-1}, C_j)^{2p} \right]$$

$$\leq 2pK \left( \mathbb{E}[w_h(t, C_{h-1}, C_h)^{2p}] \right)^{(2p-1)/(2p)} \left( \mathbb{E}[\partial_2 \mathcal{L}(Y, \hat{y}(t, X))^2] \right)^{1/2} \mathbb{E}_{C_{h+1}, C_h} \left[ w_{h+1}(t, C_h, C_{h+1})^{2p} \right]^{1/2p}.$$

Thus, as above,

$$\mathbb{E}[w_h(t, C_{h-1}, C_h)^{2p}]^{1/(2p)} \leq Kp^{1/2} + (Kp^{1/2} + K) \int_0^t \left( \mathbb{E}[\partial_2 \mathcal{L}(Y, \hat{y}(s, X))^2] \right)^{1/2} ds,$$

so we get

$$\sup_{t\geq0} \mathbb{E}[w_h(t, C_{h-1}, C_h)^{2p}]^{1/(2p)} \leq Kp^{1/2} + K.$$

This proves the claim. $\square$

Under the assumption (8), the MF limit $w_i(t, \cdot, \cdot)$ converges in $L^2$ to a limit $\overline{w}_i$, as shown in the corollary below.

**Corollary 36.** *Under the setting of Lemma 35, we have*

$$\mathbb{E}\left[|w_t(t, C_{i-1}, C_i) - \overline{w}_i(C_{i-1}, C_i)|^{2k}\right] \leq K_\delta^k \int_t^\infty s^{(2k-1)(1+\delta)} \mathbb{E}_Z\left[\partial_2 \mathcal{L}(Y, \hat{y}(s, X))^2\right]^k ds.$$

*Proof.* We have from the previous lemma that

$$\mathbb{E}\left[|\partial_t w_i(t, C_{i-1}, C_i)|\right] \leq \mathbb{E}\left[|\partial_t w_i(t, C_{i-1}, C_i)|^{2p}\right]^{1/(2p)} \leq K_p \mathbb{E}_Z\left[\partial_2 \mathcal{L}(Y, \hat{y}(t, X))^2\right]^{1/2}.$$

By Holder's inequality,

$$|w_t(t, C_{i-1}, C_i) - \overline{w}_i(C_{i-1}, C_i)|^{2k} \leq \left(\int_t^\infty s^{-1-\delta} ds\right)^{2k-1} \left(\int_t^\infty s^{(2k-1)(1+\delta)} |\partial_t w_i(s, C_{i-1}, C_i)|^{2k} ds\right)$$

$$\leq K_\delta^k \left(\int_t^\infty s^{(2k-1)(1+\delta)} |\partial_t w_i(s, C_{i-1}, C_i)|^{2k} ds\right).$$

Thus,

$$\mathbb{E}\left[|w_t(t, C_{i-1}, C_i) - \overline{w}_i(C_{i-1}, C_i)|^{2k}\right] \leq K_\delta^k \int_t^\infty s^{(2k-1)(1+\delta)} \mathbb{E}_{C_{i-1}, C_i}\left[|\partial_t w_i(s, C_{i-1}, C_i)|^{2k}\right] ds$$

$$\leq K_\delta^k \int_t^\infty s^{(2k-1)(1+\delta)} \mathbb{E}_Z\left[\partial_2 \mathcal{L}(Y, \hat{y}(s, X))^2\right]^k ds.$$

$\square$

This suggests that one should have convergence to a limit as $t \to \infty$ at the fluctuation level. Should we have that, we would obtain Theorem 10 as a consequence of Theorem 9. The caveat is that the convergence must satisfy certain *uniform-in-time* properties. This is the bulk of the work.

Now we let $\overline{G}, \overline{w}, \overline{R}$ be obtained by plugging the infinite-time limit $\overline{w} = \{\overline{w}_i\}$ into $w = \{w_i(t, \cdot, \cdot)\}$. Recall that $\underline{G}$ is the limiting Gaussian process defined in Theorem 18. We then have:

$$\partial_t R_i(\underline{G}, t, c_{i-1}, c_i) = -\mathbb{E}_Z\left[\frac{\partial \hat{y}(t, X)}{\partial w_i(c_{i-1}, c_i)} \cdot \partial_2^2 \mathcal{L}(Y, \hat{y}(t, X)) \cdot \left(\underline{G}^y(t, X) + \sum_{j=1}^L \mathbb{E}_{C'_{j-1}, C'_j}\left\{R_j(\underline{G}, t, C'_{j-1}, C'_j)\frac{\partial \hat{y}(t, X)}{\partial w_j(C'_{j-1}, C'_j)}\right\}\right)\right]$$

$$- \mathbb{E}_Z\left[\partial_2 \mathcal{L}(Y, \hat{y}(t, X)) \cdot \left(\underline{G}_i^w(t, c_{i-1}, c_i, X) + \sum_{j=1}^L \mathbb{E}_{C'_{j-1}, C'_j}\left\{R_j(\underline{G}, t, C'_{j-1}, C'_j)\frac{\partial^2 \hat{y}(t, X)}{\partial w_j(C'_{j-1}, C'_j)\partial w_i(c_{i-1}, c_i)}\right\}\right)\right]$$

$$- \mathbb{E}_Z\left[\partial_2 \mathcal{L}(Y, \hat{y}(t, X)) \cdot \mathbb{E}_{C'_{i-1}}\left\{R_i(\underline{G}, t, C'_{i-1}, c_i)\frac{\partial^2 \hat{y}(t, X)}{\partial_* w_i(C'_{i-1}, c_i)\partial w_i(c_{i-1}, c_i)}\right\}\right]$$

$$- \mathbb{E}_Z\left[\partial_2 \mathcal{L}(Y, \hat{y}(t, X)) \cdot \mathbb{E}_{C'_{i+1}}\left\{R_i(\underline{G}, t, c_i, C'_{i+1})\frac{\partial^2 \hat{y}(t, X)}{\partial_* w_{i+1}(c_i, C'_{i+1})\partial w_i(c_{i-1}, c_i)}\right\}\right]$$

$$- \mathbb{E}_Z\left[\partial_2 \mathcal{L}(Y, \hat{y}(t, X)) \cdot \mathbb{E}_{C'_{i-2}}\left\{R_i(\underline{G}, t, C'_{i-2}, c_{i-1})\frac{\partial^2 \hat{y}(t, X)}{\partial_* w_{i-1}(C'_{i-2}, c_{i-1})\partial w_i(c_{i-1}, c_i)}\right\}\right],$$

$$\partial_t \overline{R}_i(\overline{G}, t, c_{i-1}, c_i) = -\mathbb{E}_Z\left[\frac{\partial \hat{y}(X)}{\partial \overline{w}_i(c_{i-1}, c_i)} \cdot \partial_2^2 \mathcal{L} \cdot \left(\overline{G}^y(t, X) + \sum_{j=1}^L \mathbb{E}_{C'_{j-1}, C'_j}\left\{\overline{R}_j(\overline{G}, t, C'_{j-1}, C'_j)\frac{\partial \hat{y}(X)}{\partial \overline{w}_j(C'_{j-1}, C'_j)}\right\}\right)\right],$$

where $\partial_2^2 \mathcal{L} > 0$ denotes the positive constant at global optima as in Assumption 2. Whenever the context is clear, we write $R$ for $R(\underline{G}, \cdot)$. We also recall from Theorem 6 that the limiting output fluctuation $\hat{G}$ is described via $R(\underline{G}, \cdot)$.

Let us rewrite these dynamics in the following form:

$$\partial_t R = \mathfrak{A}_t R + H_t, \qquad \partial_t \overline{R} = \overline{\mathfrak{A}} \overline{R} + \overline{H},$$

which implies

$$\partial_t (R - \overline{R}) = (\mathfrak{A}_t - \overline{\mathfrak{A}})R + (H_t - \overline{H}) + \overline{\mathfrak{A}}(R - \overline{R}),$$

$$(R - \overline{R})(s) = \int_s^\infty \exp(-t\overline{\mathfrak{A}})((\mathfrak{A}_t - \overline{\mathfrak{A}})R + (H_t - \overline{H}))dt,$$

for linear operators $\mathfrak{A}_t$ and $\overline{\mathfrak{A}}$ defined by

$$\mathfrak{A}_t S(c_{i-1}, c_i) = -\mathbb{E}_Z \left[ \frac{\partial \hat{y}(t,X)}{\partial w_i(c_{i-1},c_i)} \cdot \partial_2 \mathcal{L}(Y, \hat{y}(t,X)) \cdot \left( \sum_{j=1}^L \mathbb{E}_{C'_{j-1},C'_j} \left\{ S_j(C'_{j-1}, C'_j) \frac{\partial \hat{y}(t,X)}{\partial w'_j(C'_{j-1},C'_j)} \right\} \right) \right]$$

$$- \mathbb{E}_Z \left[ \partial_2 \mathcal{L}(Y, \hat{y}(t,X)) \cdot \left( \sum_{j=1}^L \mathbb{E}_{C'_{j-1},C'_j} \left\{ S_j(C'_{j-1}, C'_j) \frac{\partial^2 \hat{y}(t,X)}{\partial w_j(C'_{j-1},C'_j)\partial w_i(c_{i-1},c_i)} \right\} \right) \right]$$

$$- \mathbb{E}_Z \left[ \partial_2 \mathcal{L}(Y, \hat{y}(t,X)) \cdot \mathbb{E}_{C'_{i-1}} \left\{ S_i(C'_{i-1}, c_i) \frac{\partial^2 \hat{y}(t,X)}{\partial_* w_i(C'_{i-1},c_i)\partial w_i(c_{i-1},c_i)} \right\} \right]$$

$$- \mathbb{E}_Z \left[ \partial_2 \mathcal{L}(Y, \hat{y}(t,X)) \cdot \mathbb{E}_{C'_{i+1}} \left\{ S_i(c_i, C'_{i+1}) \frac{\partial^2 \hat{y}(t,X)}{\partial_* w_{i+1}(c_i,C'_{i+1})\partial w_i(c_{i-1},c_i)} \right\} \right]$$

$$- \mathbb{E}_Z \left[ \partial_2 \mathcal{L}(Y, \hat{y}(t,X)) \cdot \mathbb{E}_{C'_{i-2}} \left\{ S_i(C'_{i-2}, c_{i-1}) \frac{\partial^2 \hat{y}(t,X)}{\partial_* w_{i-1}(C'_{i-2},c_{i-1})\partial w_i(c_{i-1},c_i)} \right\} \right],$$

$$\overline{\mathfrak{A}} S(c_{i-1}, c_i) = -\mathbb{E}_Z \left[ \frac{\partial \hat{y}(X)}{\partial \overline{w}_i(c_{i-1},c_i)} \cdot \partial_2^2 \mathcal{L} \cdot \left( \sum_{j=1}^L \mathbb{E}_{C'_{j-1},C'_j} \left\{ S(C'_{j-1}, C'_j) \frac{\partial \hat{y}(X)}{\partial \overline{w}'_j(C'_{j-1},C'_j)} \right\} \right) \right],$$

$$H_t(c_{i-1}, c_i) = -\mathbb{E}_Z \left[ \frac{\partial \hat{y}(t,X)}{\partial w_i(c_{i-1},c_i)} \cdot \partial_2^2 \mathcal{L}(Y, \hat{y}(t,X)) \cdot \underline{G}^y(t,X) \right] - \mathbb{E}_Z \left[ \partial_2 \mathcal{L}(Y, \hat{y}(t,X)) \cdot \underline{G}_i^w(t,c_{i-1},c_i,X) \right],$$

$$\overline{H}(c_{i-1}, c_i) = -\mathbb{E}_Z \left[ \frac{\partial \hat{y}(X)}{\partial \overline{w}_i(c_{i-1},c_i)} \cdot \partial_2^2 \mathcal{L} \cdot \overline{G}^y(X) \right].$$

**Lemma 37.** *For any $\delta > 0$ and $k \geq 1$, we have the estimate*

$$\mathbf{E}\|H_t - \overline{H}\|_{2k}^{2k} := \mathbf{E}\left\{ \sum_{i=1}^L \mathbb{E}_C \left[ \left( \mathbb{E}_Z \left[ \frac{\partial \hat{y}(t,X)}{\partial w_i(C_{i-1},C_i)} \cdot \partial_2^2 \mathcal{L}(Y, \hat{y}(t,X)) \cdot \underline{G}^y(t,X) \right] - \mathbb{E}_Z \left[ \frac{\partial \hat{y}(X)}{\partial \overline{w}_i(C_{i-1},C_i)} \cdot \partial_2^2 \mathcal{L} \cdot \overline{G}^y(X) \right] \right)^{2k} \right]$$

$$+ \sum_{i=1}^L \mathbb{E}_C \left[ \left( \mathbb{E}_Z \left[ \partial_2 \mathcal{L}(Y, \hat{y}(t,X)) \cdot \underline{G}_i^w(t,c_{i-1},c_i,X) \right] \right)^{2k} \right] \right\}$$

$$\leq K_{k,\delta} \int_t^\infty s^{(2k-1)(1+\delta)} \mathbb{E}_Z \left[ \partial_2 \mathcal{L}(Y, \hat{y}(s,X))^2 \right]^k ds.$$

*Proof.* We have

$$\mathbb{E}_{C_{i-1},C_i} \left[ \left( \mathbb{E}_Z \left[ \frac{\partial \hat{y}(t,X)}{\partial w_i(C_{i-1},C_i)} \cdot \partial_2^2 \mathcal{L}(Y, \hat{y}(t,X)) \cdot \underline{G}^y(t,X) \right] - \mathbb{E}_Z \left[ \frac{\partial \hat{y}(X)}{\partial \overline{w}_i(C_{i-1},C_i)} \cdot \partial_2^2 \mathcal{L} \cdot \overline{G}^y(X) \right] \right)^{2k} \right]$$

$$\leq K_k \mathbb{E}_{C_{i-1},C_i} \left[ \mathbb{E}_Z \left[ \left( \frac{\partial \hat{y}(t,X)}{\partial w_i(C_{i-1},C_i)} \cdot \partial_2^2 \mathcal{L}(Y, \hat{y}(t,X)) - \frac{\partial \hat{y}(X)}{\partial \overline{w}_i(C_{i-1},C_i)} \cdot \partial_2^2 \mathcal{L} \right) \cdot \underline{G}^y(t,X) \right]^{2k} \right.$$

$$\left. + \mathbb{E}_Z \left[ \frac{\partial \hat{y}(X)}{\partial \overline{w}_i(C_{i-1},C_i)} \cdot \partial_2^2 \mathcal{L} \cdot \left( \underline{G}^y(t,X) - \overline{G}^y(X) \right) \right]^{2k} \right]$$

$$\leq K_k \mathbb{E}_Z \left[ \underline{G}^y(t,X)^2 \right]^k \mathbb{E}_{C_{i-1},C_i} \left[ \mathbb{E}_Z \left[ \left( \frac{\partial \hat{y}(t,X)}{\partial w_i(C_{i-1},C_i)} \cdot \partial_2^2 \mathcal{L}(Y,\hat{y}(t,X)) - \frac{\partial \hat{y}(X)}{\partial \overline{w}_i(C_{i-1},C_i)} \cdot \partial_2^2 \mathcal{L} \right)^2 \right]^k \right]$$

$$+ K_k \mathbb{E}_Z \left[ \left( \underline{G}^y(t,X) - \overline{G}^y(X) \right)^2 \right].$$

Note that $\sup_t \mathbb{E}_Z \left[ \underline{G}^y(t,X)^2 \right]^k < \infty$ since $W(t)$ is uniformly bounded in $L^{2p}$ for any $p \geq 1$ by Lemma 35. Furthermore,

$$\mathbb{E}_{C_{i-1},C_i} \left[ \mathbb{E}_Z \left[ \left( \frac{\partial \hat{y}(t,X)}{\partial w_i(C_{i-1},C_i)} \cdot \partial_2^2 \mathcal{L}(Y,\hat{y}(t,X)) - \frac{\partial \hat{y}(X)}{\partial \overline{w}_i(C_{i-1},C_i)} \cdot \partial_2^2 \mathcal{L} \right)^2 \right]^k \right]$$

$$\leq K_k \mathbb{E}_{C_{i-1},C_i,Z} \left[ \left( \frac{\partial \hat{y}(t,X)}{\partial w_i(C_{i-1},C_i)} - \frac{\partial \hat{y}(X)}{\partial \overline{w}_i(C_{i-1},C_i)} \right)^{2k} \right] + \mathbb{E}_{C_{i-1},C_i} \left[ \mathbb{E}_Z \left[ \left( \frac{\partial \hat{y}(t,X)}{\partial w_i(C_{i-1},C_i)} \cdot \left( \partial_2^2 \mathcal{L}(Y,\hat{y}(t,X)) - \partial_2^2 \mathcal{L} \right) \right)^2 \right]^k \right]$$

$$\leq K_k \mathbb{E}_{C_{i-1},C_i,Z} \left[ \left( \frac{\partial \hat{y}(t,X)}{\partial w_i(C_{i-1},C_i)} - \frac{\partial \hat{y}(X)}{\partial \overline{w}_i(C_{i-1},C_i)} \right)^{2k} \right] + K_k \mathbb{E}_Z \left[ \left( \partial_2^2 \mathcal{L}(Y,\hat{y}(t,X)) - \partial_2^2 \mathcal{L} \right)^{2k} \right]$$

$$\leq K_k \mathbb{E}_{C_{i-1},C_i,Z} \left[ \left( \frac{\partial \hat{y}(t,X)}{\partial w_i(C_{i-1},C_i)} - \frac{\partial \hat{y}(X)}{\partial \overline{w}_i(C_{i-1},C_i)} \right)^{2k} \right] + K_k \mathbb{E}_Z \left[ \left( \hat{y}(t,X) - \overline{\hat{y}}(X) \right)^{2k} \right].$$

Using uniform boundedness of $W(t)$ in $t$, we have

$$\mathbb{E}_{C_{i-1},C_i,Z} \left[ \left( \frac{\partial \hat{y}(t,X)}{\partial w_i(C_{i-1},C_i)} - \frac{\partial \hat{y}(X)}{\partial \overline{w}_i(C_{i-1},C_i)} \right)^{2k} \right] \leq K_k \sum_j \mathbb{E}_{C_{j-1},C_j} \left[ (w_j(t,C_{j-1},C_j) - \overline{w}_j(C_{j-1},C_j))^{2k} \right],$$

and

$$\mathbb{E}_Z \left[ \left( \hat{y}(t,X) - \overline{\hat{y}}(X) \right)^{2k} \right] \leq K_k \sum_j \mathbb{E}_{C_{j-1},C_j} \left[ (w_j(t,C_{j-1},C_j) - \overline{w}_j(C_{j-1},C_j))^{2k} \right].$$

By Lemma 36,

$$\mathbb{E}_{C_{i-1},C_i} \left[ \mathbb{E}_Z \left[ \left( \frac{\partial \hat{y}(t,X)}{\partial w_i(C_{i-1},C_i)} \cdot \partial_2^2 \mathcal{L}(Y,\hat{y}(t,X)) - \frac{\partial \hat{y}(X)}{\partial \overline{w}_i(C_{i-1},C_i)} \cdot \partial_2^2 \mathcal{L} \right)^2 \right]^k \right]$$

$$\leq K_\delta \int_t^\infty s^{(2k-1)(1+\delta)} \mathbb{E}_Z \left[ \partial_2 \mathcal{L}(Y,\hat{y}(s,X))^2 \right]^k ds.$$

Next, we have

$$\mathbf{E}\mathbb{E}_Z \left[ \left( \underline{G}^y(t,X) - \overline{G}^y(X) \right)^2 \right]^k \leq K_{k,\delta} \int_t^\infty s^{(2k-1)(1+\delta)} \mathbf{E}\mathbb{E}_Z \left[ (\partial_t \underline{G}^y(s,X))^{2k} \right] ds.$$

Using the covariance structure of $\underline{G}^{\partial_t H}$ and $\underline{G}^{\partial_t(\partial H)}$, we similarly deduce that

$$\mathbf{E}\mathbb{E}_Z \left[ (\partial_t \underline{G}^y(t,X))^{2k} \right] \leq K_k \mathbb{E}_Z \left[ \partial_2 \mathcal{L}(Y,\hat{y}(t,X))^2 \right]^k.$$

Furthermore,

$$\sum_{i=1}^L \mathbf{E}\mathbb{E}_{C_{i-1},C_i} \left[ (\mathbb{E}_Z \left[ \partial_2 \mathcal{L}(Y,\hat{y}(t,X)) \cdot \underline{G}_i^w(t,c_{i-1},c_i,X) \right])^{2k} \right] \leq K_k \mathbb{E}_Z \left[ \partial_2 \mathcal{L}(Y,\hat{y}(t,X))^2 \right]^k,$$

by Lemma 35. Hence,

$$\mathbf{E} \left[ \|H_t - \overline{H}\|_{2k}^{2k} \right] \leq K_{k,\delta} \int_t^\infty s^{(2k-1)(1+\delta)} \mathbb{E}_Z \left[ \partial_2 \mathcal{L}(Y,\hat{y}(s,X))^2 \right]^k ds.$$

$\square$

Note that under Assumption (8), $\mathbb{E}_Z \left[ \partial_2 \mathcal{L}(Y,\hat{y}(s,X))^2 \right] \to 0$ as $s \to \infty$, and hence

$$\int_0^\infty s^{(2k-1)(1+\delta)} \mathbb{E}_Z \left[ \partial_2 \mathcal{L}(Y,\hat{y}(s,X))^2 \right]^k ds \leq K_k + \int_0^\infty s^{(2k-1)(1+\delta)} \mathbb{E}_Z \left[ \partial_2 \mathcal{L}(Y,\hat{y}(s,X))^2 \right] ds.$$

Thus, under the assumption, we have that for $k = 1 + \epsilon$ with $\epsilon$ sufficiently small, and taking $\delta$ sufficiently small, we conclude that $\mathbf{E}\|H_t - \overline{H}\|_{2k}^{2k} \to 0$ as $t \to \infty$ for all $k = k(\eta)$ sufficiently close to 1.

**Lemma 38.** *We have* $\sup_t \|\overline{R}(t)\|_2^2 / \|\overline{H}\|_2^2 < \infty$.

*Proof.* We have that $\overline{R}$ is the solution to

$$\partial_t \overline{R} = \overline{\mathfrak{A}} \overline{R} + \overline{H}.$$

In the proof of Theorem 9, we observe that $\overline{\mathfrak{A}}$ is self-adjoint and has nonpositive eigenvalues. Hence,

$$\overline{R}/\|\overline{H}\|_2 = \exp(t\overline{\mathfrak{A}})\overline{H}/\|\overline{H}\|_2$$

is bounded uniformly in time. $\qquad\square$

**Lemma 39.** *We have the following estimate for $\delta$ sufficiently small (depending on $\eta$):*

$$\mathbf{E}\|(\mathfrak{A}_t - \overline{\mathfrak{A}})R\|_2^2 \le K_\delta \int_t^\infty s^{1+\delta} \mathbb{E}_Z \left[\partial_2 \mathcal{L}(Y, \hat{y}(s, X))^2\right] ds + K_\delta \mathbb{E}_Z \left[\partial_2 \mathcal{L}(Y, \hat{y}(t, X))^2\right].$$

*Proof.* We first follow the proof of Theorem 21 to give uniform-in-time estimates for $R(t)$ under the assumption (8). For a threshold $B$, define the linear operator

$$\mathfrak{A}_t^B S(c_{i-1}, c_i) = -\mathbb{E}_Z \left[ \frac{\partial \hat{y}(t, X)}{\partial w_i(c_{i-1}, c_i)} \cdot \partial_2^2 \mathcal{L}(Y, \hat{y}(t, X)) \cdot \left( \sum_{j=1}^L \mathbb{E}_{C'_{j-1}, C'_j} \left\{ S_j(C'_{j-1}, C'_j) \frac{\partial \hat{y}(t, X)}{\partial w_j(C'_{j-1}, C'_j)} \right\} \right) \right]$$
$$+ \tilde{\mathfrak{A}}_t^B S(c_{i-1}, c_i),$$

$$\tilde{\mathfrak{A}}_t^B S(c_{i-1}, c_i) = -\left\{ \mathbb{E}_Z \left[ \partial_2 \mathcal{L}(Y, \hat{y}(t, X)) \cdot \left( \sum_{j=1}^L \mathbb{E}_{C'_{j-1}, C'_j} \left\{ S_j(C'_{j-1}, C'_j) \frac{\partial^2 \hat{y}(t, X)}{\partial w_j(C'_{j-1}, C'_j) \partial w_i(c_{i-1}, c_i)} \right\} \right) \right] \right.$$

$$- \mathbb{E}_Z \left[ \partial_2 \mathcal{L}(Y, \hat{y}(t, X)) \cdot \mathbb{E}_{C'_{i-1}} \left\{ S_i(C'_{i-1}, c_i) \frac{\partial^2 \hat{y}(t, X)}{\partial_* w_i(C'_{i-1}, c_i) \partial w_i(c_{i-1}, c_i)} \right\} \right]$$

$$- \mathbb{E}_Z \left[ \partial_2 \mathcal{L}(Y, \hat{y}(t, X)) \cdot \mathbb{E}_{C'_{i+1}} \left\{ S_i(c_i, C'_{i+1}) \frac{\partial^2 \hat{y}(t, X)}{\partial_* w_{i+1}(c_i, C'_{i+1}) \partial w_i(c_{i-1}, c_i)} \right\} \right]$$

$$\left. - \mathbb{E}_Z \left[ \partial_2 \mathcal{L}(Y, \hat{y}(t, X)) \cdot \mathbb{E}_{C'_{i-2}} \left\{ S_i(C'_{i-2}, c_{i-1}) \frac{\partial^2 \hat{y}(t, X)}{\partial_* w_{i-1}(C'_{i-2}, c_{i-1}) \partial w_i(c_{i-1}, c_i)} \right\} \right] \right\} \mathbb{B}^B(t, c_{i-1}, c_i),$$

and

$$H_t(c_{i-1}, c_i) = -\mathbb{E}_Z \left[ \frac{\partial \hat{y}(t, X)}{\partial w_i(c_{i-1}, c_i)} \cdot \partial_2^2 \mathcal{L}(Y, \hat{y}(t, X)) \cdot G^y(t, X) \right] - \mathbb{E}_Z \left[\partial_2 \mathcal{L}(Y, \hat{y}(t, X)) \cdot G_i^w(t, c_{i-1}, c_i, X) \right],$$

$$\partial_t R^B(\underline{G}, t, \cdot, \cdot) = \mathfrak{A}_t^B (R^B(\underline{G}, t)) + H(\underline{G}, t).$$

Following Theorem 21, we have the bound

$$\left| \tilde{\mathfrak{A}}_t^B (R^B(\underline{G}, t))(c_{i-1}, c_i) \right| \le K \Big( \|R^B(t)\| \|w_i(t, c_i)\| \|w_{i+1}(t, c_i)\| + \|R_i^B(t, c_i)\| \|w_{i+1}(t, c_i)\|$$
$$+ \|R_{i+1}^B(t, c_i)\| + \|R_i^B(t, c_{i-1})\| \|w_i(t, c_i)\| \Big) \cdot \|\partial_2 \mathcal{L}(Y, \hat{y}(t, X))\| \mathbb{B}^B(t, c_{i-1}, c_i).$$

Thus

$$\mathbb{E}_{C_{i-1}, C_i} \left[ \left| \tilde{\mathfrak{A}}_t^B (R^B(\underline{G}, t))(C_{i-1}, C_i) \right|^2 \right] \le K \|R^B(t)\|^2 (1 + B^2) \|\partial_2 \mathcal{L}(Y, \hat{y}(t, X))\|^2,$$

where we have used uniform-in-time boundedness of $W(t)$ from Lemma 35. We then have

$$\partial_t \|R^B(t)\|_2^2$$
$$\le -\sum_{i=1}^L \mathbb{E}_{Z, C_{i-1}, C_i} \left[ R_i^B(\underline{G}, t, C_{i-1}, C_i) \frac{\partial \hat{y}(t, X)}{\partial w_i(C_{i-1}, C_i)} \cdot \partial_2^2 \mathcal{L}(Y, y) \cdot \left( \sum_{j=1}^L \mathbb{E}_{C'_{j-1}, C'_j} \left\{ R_j^B(\underline{G}, t, C'_{j-1}, C'_j) \frac{\partial \hat{y}(t, X)}{\partial w'_j(C'_{j-1}, C'_j)} \right\} \right) \right]$$
$$+ \|R^B(t)\| \cdot K \|R^B(t)\| (1 + B) \|\partial_2 \mathcal{L}(Y, \hat{y}(t, X))\| + \|H(\underline{G}, t)\|^2$$

$$\leq -\partial_2^2 \mathcal{L} \left( \sum_{j=1}^{L} \mathbb{E}_{C'_{j-1}, C'_j} \left\{ R_j^B(\underline{G}, t, C'_{j-1}, C'_j) \frac{\partial \hat{y}(t, X)}{\partial w'_j(C'_{j-1}, C'_j)} \right\} \right)^2 + K \|w(t)\|_K^{K_L} \|\partial_2^2 \mathcal{L}(Y, \hat{y}(t, X)) - \partial_2^2 \mathcal{L}\| \|R^B(t)\|_2^2$$

$$+ K \|R^B(t)\|^2 (1 + B) \|\partial_2 \mathcal{L}(Y, \hat{y}(t, X))\| + \|H(\underline{G}, t)\|^2.$$

Thus,

$$\partial_t \|R^B(\underline{G}, t)\|_2^2 \leq K(1 + B) \|R^B(t)\|^2 \|\partial_2 \mathcal{L}(Y, \hat{y}(t, X))\| + K \|w(t)\|_K^{K_L} \|\partial_2^2 \mathcal{L}(Y, \hat{y}(t, X)) - \partial_2^2 \mathcal{L}\|_2^2 + \|H(\underline{G}, t)\|^2.$$

Using that

$$\int_0^\infty \|\partial_2^2 \mathcal{L}(Y, \hat{y}(t, X)) - \partial_2^2 \mathcal{L}\| dt < \infty,$$

$$\mathbf{E} \int_0^\infty \|H(\underline{G}, t)\|^2 dt < \infty,$$

$$\int_0^\infty \|\partial_2 \mathcal{L}(Y, \hat{y}(t, X))\| dt < \infty,$$

as well as Lemma 35, we can then conclude

$$\sup_t \mathbf{E} \|R^B(\underline{G}, t)\|_2^2 \leq \exp(K(1 + B)).$$

Next, we compare $R^B$ and $R^{B'}$ as in Theorem 21 for $B' > B$. We have the bound, recalling the definition of $\mathbb{D}_{B,B'}$ in Theorem 21,

$$\partial_t \mathbf{E} \|(R^B - R^{B'})(t)\|_2^2$$

$$\leq -\sum_{i=1}^{L} \mathbf{E} \mathbb{E}_{Z, C_{i-1}, C_i} \left[ (R_i^B - R_i^{B'})(\underline{G}, t, C_{i-1}, C_i) \frac{\partial \hat{y}(t, X)}{\partial w_i(C_{i-1}, C_i)} \cdot \partial_2^2 \mathcal{L}(Y, y) \right.$$

$$\left. \times \left( \sum_{j=1}^{L} \mathbb{E}_{C'_{j-1}, C'_j} \left\{ (R_j^B - R_j^{B'})(\underline{G}, t, C'_{j-1}, C'_j) \frac{\partial \hat{y}(t, X)}{\partial w'_j(C'_{j-1}, C'_j)} \right\} \right) \right]$$

$$+ K \mathbf{E} \|R^B(t)\|^2 (1 + B) \|\partial_2 \mathcal{L}(Y, \hat{y}(t, X))\|$$

$$+ K_\delta \sum_{i=1}^{L} \mathbf{E} \mathbb{E}_{C_{i-1}, C_i} \left[ \mathbb{D}_{B,B'}(t, C_{i-1}, C_i) \exp(K(1 + B')) \int_0^t (\|\underline{G}_{i+1}(s, C_i)\|^2 + \|\underline{G}_i(s, C_i)\|^2) ds \right] \|\partial_2 \mathcal{L}(Y, \hat{y}(t, X))\|$$

$$\leq -\partial_2^2 \mathcal{L} \mathbf{E} \left[ \left( \sum_{j=1}^{L} \mathbb{E}_{C'_{j-1}, C'_j} \left\{ (R_j^B - R_j^{B'})(\underline{G}, t, C'_{j-1}, C'_j) \frac{\partial \hat{y}(t, X)}{\partial w'_j(C'_{j-1}, C'_j)} \right\} \right)^2 \right]$$

$$+ K \|w(t)\|_K^{K_L} \|\partial_2^2 \mathcal{L}(Y, \hat{y}(t, X)) - \partial_2^2 \mathcal{L}\|_2 \mathbf{E} \|R^B - R^{B'}\|^2$$

$$+ K \mathbf{E} \|(R^B - R^{B'})(t)\|^2 (1 + B) \|\partial_2 \mathcal{L}(Y, \hat{y}(t, X))\| + \exp(K(1 + B') - cB^2) \|\partial_2 \mathcal{L}(Y, \hat{y}(t, X))\|.$$

Again using our convergence assumption, we can conclude that

$$\sup_t \mathbf{E} \|(R^B - R^{B'})(t)\|^2 \leq \exp(K(1 + B)) \exp(K(1 + B') - cB^2).$$

In particular, $R^B$ converges (uniformly in time $t$) in $L^2$ to the limit $R$, whose norm is uniformly bounded in time.

An identical argument shows that upon having finite moment $\|H_t\|^{2k+\delta}$, we have that $R^B$ converges (uniformly in time) in $L^{2k}$ to a limit $R$, whose $L^{2k}$ norm is uniformly bounded in time. This is guaranteed for all $k$ sufficiently close to 1 (in terms of $\eta$) by the remark following Lemma 37.

From Holder's Inequality, we have the following estimate for sufficiently small $\delta$:

$$\|\tilde{\mathfrak{A}}_t R(t)\|_2^2 \leq K_\delta \|R(t)\|_{2+\delta}^2 \|\partial_2 \mathcal{L}(Y, \hat{y}(t, X))\|^2,$$

and

$$\|((\mathfrak{A}_t - \tilde{A}_t) - \overline{\mathfrak{A}})R(t)\|_2^2 \leq K_\delta \|R(t)\|_2^2 \int_t^\infty s^{1+\delta} \mathbb{E}_Z \left[ \partial_2 \mathcal{L}(Y, \hat{y}(s, X))^2 \right] ds.$$

Thus,

$$\mathbf{E}\|(\mathfrak{A}_t - \overline{\mathfrak{A}})R(t)\|_2^2 \le K_\delta \int_t^\infty s^{1+\delta}\mathbb{E}_Z\left[\partial_2\mathcal{L}(Y,\hat{y}(s,X))^2\right]ds + \mathbb{E}_Z\left[\partial_2\mathcal{L}(Y,\hat{y}(t,X))^2\right].$$

$\square$

**Lemma 40.** *We have for $\delta$ sufficiently small in $\eta$ that*

$$\mathbf{E}\|R(t) - \overline{R}\|_2^2 \le K_\delta\left(\int_t^\infty (s^{2+\delta} - t^{2+\delta} + 1)\mathbb{E}_Z\left[\partial_2\mathcal{L}(Y,\hat{y}(s,X))^2\right]ds\right).$$

*Proof.* We have

$$\partial_t\mathbf{E}\|R(t) - \overline{R}\|_2^2 = 2\mathbf{E}\langle R(t) - \overline{R}, (\mathfrak{A}_t - \overline{\mathfrak{A}})R(t) + (H_t - \overline{H}) + \overline{\mathfrak{A}}(R(t) - \overline{R})\rangle.$$

We now bound

$$\mathbf{E}\langle R(t) - \overline{R}, H_t - \overline{H}\rangle \le K_\delta\left(\int_t^\infty s^{1+\delta}\mathbb{E}_Z\left[\partial_2\mathcal{L}(Y,\hat{y}(s,X))^2\right]ds\right)^{1/2}\cdot(\mathbf{E}\|R(t) - \overline{R}\|_2^2)^{1/2},$$

We also have

$$\langle R(t) - \overline{R}, \overline{\mathfrak{A}}(R(t) - \overline{R})\rangle \le 0.$$

From Lemma 39,

$$\mathbf{E}\left[\langle R(t) - \overline{R}, (\mathfrak{A}_t - \overline{\mathfrak{A}})\overline{R}\rangle\right] \le K\mathbf{E}\left[\|R(t) - \overline{R}\|_2^2\right]^{1/2}\mathbf{E}\left[\|(\mathfrak{A}_t - \overline{\mathfrak{A}})\overline{R}\|_2^2\right]^{1/2}$$

$$\le K\mathbf{E}\left[\|R(t) - \overline{R}\|_2^2\right]^{1/2}\left(K_\delta\int_t^\infty s^{1+\delta}\mathbb{E}_Z\left[\partial_2\mathcal{L}(Y,\hat{y}(s,X))^2\right]ds + K_\delta\mathbb{E}_Z\left[\partial_2\mathcal{L}(Y,\hat{y}(t,X))^2\right]\right)^{1/2}.$$

Thus,

$$\partial_t\mathbf{E}\|R(t) - \overline{R}\|_2^2 \le K_\delta\left(\mathbb{E}_Z\left[\partial_2\mathcal{L}(Y,\hat{y}(t,X))^2\right] + \int_t^\infty s^{1+\delta}\mathbb{E}_Z\left[\partial_2\mathcal{L}(Y,\hat{y}(s,X))^2\right]ds\right)^{1/2}(\mathbf{E}\|R(t) - \overline{R}\|_2^2)^{1/2}.$$

In particular,

$$\partial_t(\mathbf{E}\|R(t) - \overline{R}\|_2^2)^{1/2} \le K_\delta\left(\mathbb{E}_Z\left[\partial_2\mathcal{L}(Y,\hat{y}(t,X))^2\right] + \int_t^\infty s^{1+\delta}\mathbb{E}_Z\left[\partial_2\mathcal{L}(Y,\hat{y}(s,X))^2\right]ds\right).$$

Hence,

$$\mathbf{E}\|R(t) - \overline{R}\|_2^2 \le K_\delta\left(\int_t^\infty (s^{2+\delta} - t^{2+\delta} + 1)\mathbb{E}\left[\partial_2\mathcal{L}(Y,\hat{y}(s,X))^2\right]ds\right).$$

$\square$

We can now finish the proof of Theorem 10.

*Proof of Theorem 10.* We have from Theorem 6:

$$\lim_{N\to\infty}\mathbf{E}\mathbb{E}_X\left[\left(\sqrt{N}(\hat{\mathbf{y}}(t,X) - \hat{y}(t,X))\right)^2\right] = \mathbf{E}\mathbb{E}_Z\left[\left(\sum_{i=1}^L\mathbb{E}_C\left[R_i(\underline{G},t,C_{i-1},C_i)\frac{\partial\hat{y}(t,X)}{\partial w_i(C_{i-1},C_i)}\right] + \underline{G}^y(t,X)\right)^2\right].$$

For $\delta > 0$ sufficiently small in $\eta$,

$$\left|\mathbf{E}\mathbb{E}_Z\left[\left(\sum_{i=1}^L\mathbb{E}_C\left[R_i(\underline{G},t,C_{i-1},C_i)\frac{\partial\hat{y}(t,X)}{\partial w_i(C_{i-1},C_i)}\right] + \underline{G}^y(t,X)\right)^2\right]\right.$$

$$\left. - \mathbf{E}\mathbb{E}_Z\left[\left(\sum_{i=1}^L\mathbb{E}_C\left[\overline{R}_i(\overline{G},C_{i-1},C_i)\frac{\partial\overline{\hat{y}}(X)}{\partial w_i(C_{i-1},C_i)}\right] + \overline{G}^y(X)\right)^2\right]\right|$$

$$\le K\mathbf{E}\|R(t) - \overline{R}\|_2^2 + K\|w(t) - \overline{w}\|_2^2$$

$$\leq K_\delta \left( \int_t^\infty (s^{2+\delta} - t^{2+\delta} + 1)\mathbb{E}_Z \left[ \partial_2 \mathcal{L}(Y, \hat{y}(s,X))^2 \right] ds \right).$$

The conclusion immediately follows from the assumption

$$\int_0^\infty s^{2+\eta}\mathbb{E}[\partial_2 \mathcal{L}(Y, \hat{y}(s,X))^2]ds < \infty,$$

together with Theorem 9. $\qquad\square$

## G  Experimental details

We give a full description of the experimental setup in Section 5.

**Overall setup.**  We consider a training set of 100 randomly chosen MNIST images of digits 0, 4, 5 and 9, where we encode the label $y = +1$ if the image is digit 5 or 9 and $y = -1$ otherwise. This small scale allows us to run full-batch GD in reasonable time and avoid stochastic algorithms which create extra fluctuations that are not the focus of our study, as mentioned in Section 1.

We use a neural network of 3 layers and $\tanh$ activations in the hidden layers (i.e. $\varphi_1 = \varphi_2 = \tanh$ and $\varphi_3 = \mathrm{identity}$). We include the bias in the first layer's weight, and there is no bias elsewhere. The network has constant widths $N$ which we vary in the plots of Fig. 1. We train it with full-batch (discrete-time) GD, a Huber loss $\mathcal{L}(y, y') = \mathrm{Huber}(y - y')$ and a constant learning rate $7 \times 10^{-3}$. This learning rate is sufficiently small so that we obtain smooth evolution plots in Fig. 1(a), suggesting a behavior close to the continuous-time GD. We also define

$$\mathrm{Huber}(u) = \begin{cases} u^2/2, & |u| \leq 1, \\ |u| - 1/2, & \text{otherwise.} \end{cases}$$

All the training is run with one NVIDIA Tesla P100 GPU.

**Initialization.**  In Fig. 1(a)-(c), we initialize the network randomly as follows. We generate $\mathbf{w}_1(0, \cdot, \cdot) \sim \mathcal{N}(0, 1/785)$, $\mathbf{w}_2(0, \cdot, \cdot) \sim \mathcal{N}(0.1, 1)$ and $\mathbf{w}_3(0, \cdot, \cdot) \sim \mathcal{N}(0.1, 1)$ all independently. In Fig. 1(e), we first train a big network which has width $N^* = 5000$ and has the same configuration as described for Fig. 1(a)-(c). We stop the training of this width-$N^*$ network at iteration $10^5$, which ensures that it has found a global optimum. Then we consider a network which has width $N \in \{50, 100, 200, 400\}$ and is initialized as follows:

- Let $\mathbf{w}_i^*(T_e, \cdot, \cdot)$ be the $i$-th layer's weight of the width-$N^*$ network at $T_e$ corresponding to iteration $10^5$.
- For each $i = 1, 2$, we draw at random $\ell_{i,1}, ..., \ell_{i,N}$ from $[N^*]$ independently. Let $\ell_{0,\cdot} = \ell_{3,\cdot} = 1$.
- We set the initial weight $\mathbf{w}_i(0, j_{i-1}, j_i) = \mathbf{w}_i^*(T_e, \ell_{i-1,j_{i-1}}, \ell_{i,j_i})$ for the width-$N$ network.

In other words, the width-$N$ network at initialization is a result of subsampling the neurons of the width-$N^*$ network at time $T_e$. The width-$N^*$ network is essentially an approximation of the MF limit. Ideally when $N^* \to \infty$ then $N \to \infty$, following [17], we have that the width-$N$ network is initialized at a global optimum. Here this holds approximately at finite widths, but is sufficient for our purpose.

**Estimation for the plots.**  For Fig. 1(a), we run once for each $N$ and plot the training loss over time.

For Fig. 1(b), we draw at random a single test image $x_{\text{test}}$ (which is not among the 100 training images and happens to be a digit 5 in this plot). For each $N$, we train the network till time $T_b$, which corresponds to iteration $10^3$, and we repeat $M = 1000$ times with $M$ different randomization seeds. Let $\hat{\mathbf{y}}_N^k(T_b, x_{\text{test}})$ be the prediction of $x_{\text{test}}$ of the $k$-th repeat, for $k \in [M]$. Let

$$\hat{\mathbf{r}}_N^k(T_b, x_{\text{test}}) = \sqrt{N}\left( \hat{\mathbf{y}}_N^k(T_b, x_{\text{test}}) - \frac{1}{M} \sum_{k'=1}^M \hat{\mathbf{y}}_N^{k'}(T_b, x_{\text{test}}) \right).$$

Then we plot the histogram of $\left\{ \hat{\mathbf{r}}_N^k(T_b, x_{\text{test}}) \right\}_{k \in [M]}$.

For Fig. 1(c), we follow a procedure similar to that of Fig. 1(b), but with $M = 10$. For each $N$ and each $x$ from the training image set, we obtain $\hat{\mathbf{y}}_N^k(t, x)$ for $k \in [M]$. Then we compute:

$$v_N(t) = \frac{1}{100M} \sum_{k=1}^M \sum_{x \in \text{training set}} |\hat{\mathbf{r}}_N^k(t, x)|^2.$$

We take $v_N(t)$ as an estimate for the output variance $\mathbb{E}\mathbb{E}_Z \left[ \left| \sqrt{N}(\hat{\mathbf{y}}(t, X) - \hat{y}(t, X)) \right|^2 \right]$.

For Fig. 1(e), we follow the same procedure as Fig. 1(c), except that we train a single width-$N^*$ network that is used for all $M$ repeats and all $N \in \{50, 100, 200, 400\}$.

For Fig. 1(d), we recall the initialization procedure for Fig. 1(e). Here instead of using one value of the terminal time $T_e$, we shall let it vary. Following this procedure, for each $T_e$ and each $N$, we obtain an untrained width-$N$ network. For each $T_e$, we repeat the procedure for $M = 10$ times, but we fix the same width-$N^*$ network (terminated at time $T_e$) for all $M$ repeats. Let $\tilde{\mathbf{y}}_N^k(T_e, x)$ be the prediction of the untrained width-$N$ network at the training image $x$, in the $k$-th repeat with the terminal time $T_e$. Using these predictions, one computes:

$$\tilde{v}_N(T_e) = \frac{1}{100M} \sum_{k=1}^{M} \sum_{x \in \text{training set}} |\tilde{\mathbf{r}}_N^k(T_e, x)|^2,$$

$$\tilde{\mathbf{r}}_N^k(T_e, x) = \sqrt{N} \left( \tilde{\mathbf{y}}_N^k(T_e, x) - \frac{1}{M} \sum_{k'=1}^{M} \tilde{\mathbf{y}}_N^{k'}(T_e, x) \right).$$

We take $\tilde{v}_N(T_e)$ as an estimate for $\mathbf{E}\mathbb{E}_Z[|\tilde{G}^y(T_e, X)|^2]$.

**Reproducibility.** The experiments can be repeated using the codes in this link: `https://github.com/npminh12/nn-clt`