# OpenReview forum: "Limiting fluctuation and trajectorial stability of multilayer neural networks with mean field training"
_NeurIPS.cc/2021/Conference — NeurIPS 2021 Poster_

### Official Review · Reviewer_D7BN · 2021-07-15

**Rating:** 6
**Confidence:** 3

**Summary:**

This work focuses on the random fluctuation of the multilayer network in the mean-field limit. Leveraging on the neuronal embedding framework introduced in previous work, the authors derived a second-order mean-field limit to captures the limiting fluctuation distribution. In the large-width regime, it is shown that the gradient descent helps to reduce and eventually eliminating the variance when the learning dynamics converge sufficiently fast to a global optimum.

**Limitations And Societal Impact:**

The authors adequately addressed the limitations and potential negative societal impact of their work.

**Main Review:**

Originality: It seems that the technique introduced in this work is new and makes a novel combination of previous techniques. The related work is adequately cited and discussed. For example, the authors make a detailed comparison to the related work in terms of networks and optimization settings [1,2].

Quality: This submission is technically sound. It seems a complete piece of work in progress. Besides, the authors careful and honest about evaluating both the strengths and weaknesses of their work.

Clarity: This submission is clearly written with good organization.

Significance: My main concern is the significance of the result. It extends a similar phenomenon (gradient descent helps to reduce and eventually eliminating the variance) previously shown for shallow networks [3]. Compared with the previous work [3], the loss function setting of this work is more general. However, how the result could be used in the practice or understanding deep learning is unclear to me.

Even though this work focuses on the theoretical perspective, it would be better if the authors could provide some simple (perhaps on toy model/dataset) but insightful numerical verifications. For example, [3] complemented their results with numerical experiments.

[1] Phan-Minh Nguyen and Huy Tuan Pham, A rigorous framework for the mean field limit of multilayer neural networks, arXiv preprint arXiv:2001.11443 (2020).

[2] Justin Sirignano and Konstantinos Spiliopoulos, Mean field analysis of neural networks: A central limit theorem, Stochastic Processes and their Applications 130 (2020), no. 3, 1820–1852.

[3] Zhengdao Chen, GrantMRotskoff, Joan Bruna, and Eric Vanden-Eijnden, A dynamical central limit theorem for shallow neural networks, arXiv preprint arXiv:2008.09623 (2020).

**Time Spent Reviewing:**

3

---

> ### Author Response · Authors · 2021-08-10
> **Response to review**
>
> We thank the reviewer for finding the work novel and clearly written! We add some numerical illustration, as detailed in the general comment above (https://openreview.net/forum?id=RWioR2WoZNs&noteId=0tsV10knavE).
>
> Regarding significance, we think that it is imperative to understand multilayer nets, which we know are quite different from shallow nets in the mean field regime. More generally, the mean field scaling provides an opportunity to understand neural nets that are not oversimplified like the NTK scaling, at the expense of mathematical difficulties. One such difficult task is to understand what happens near the end of training. This task is arguably important, and increasingly difficult for the multilayer case. Yet it is expected that the landscape near convergence is typically complex and chaotic, and the infinite-width first-order approximation may no longer be excellent. As such, we think it is important to study a second-order approximation via the (up-scaled) fluctuation. In fact, in our opinion, the finding that gradient descent reduces the output variance (Theorems 9 and 10) is quite encouraging: it suggests that GD training helps stabilize the network despite the complex loss landscape. Even though this phenomenon holds for shallow nets, it is not obvious at all whether it holds for multilayer nets. Thankfully it does hold indeed as we prove, though the argument becomes quite intricate owing to depth and cross-layer interactions.

---

> > ### Comment · Reviewer_D7BN · 2021-08-29
> > **Thanks for your response**
> >
> > I thank the authors for your responses which have addressed most of my concerns. I am happy to see the new simulation the authors provided. I think this work is technically sound and provides some insights to the optimization of over-parameterized multilayer neural networks in the mean-field regime.

---

> ### Author Response · Authors · 2021-08-27
> **Response to reviewer**
>
> In the case you have any questions after reading our responses, we would love to have an opportunity to address them, so feel free to let us know!

---

### Official Review · Reviewer_EQhg · 2021-07-17

**Rating:** 6
**Confidence:** 3

**Summary:**

The paper studies the fluctuations of a wide but finite multilayer network about the mean field theory limit. The first main result of the paper establishes the form of these fluctuations as a sum of two components, one given by a Gaussian process and another by what the authors call the second-order mean field limit. Additionally, the paper shows that in the large-width limit and near the global optimum, the gradient descent reduces the variance of the network output.

**Limitations And Societal Impact:**

-

**Main Review:**

The paper addresses the important topic of network fluctuation about the infinite-width limit. The paper proposes a novel solution in the multilayer case and also shows that the (previously observed) output variance reduction effect extends to this case. The paper is well written.

The two (mutually connected) main issues with the paper are a rather unwieldy form of the main result and the absence of its independent validation. The obtained structure of the fluctuations is quite complex and is only fully described in the appendix. Because of this complexity, it is not obvious how useful is this theorem. The paper is purely theoretical and does not include any numerical experiments or applications. Moreover, the main text contains only a couple of very brief sketches of the arguments, with 50 pages of detailed proofs deferred to the appendix. I have not checked these detailed proofs, so it's hard for me to judge if the results are correct. Accordingly, I think that a journal publication with a comprehensive review process would be more appropriate for this paper. Also, the paper would benefit substantially from some experimental confirmation of its findings - at least the variance reduction effect that should be easy to demonstrate.

**Time Spent Reviewing:**

4

---

> ### Author Response · Authors · 2021-08-10
> **Response to review**
>
> We are happy that the reviewer finds the topic important, our proposed solution novel and our writing clear! We will add some numerical illustration, the details of which are discussed in the general comment above (https://openreview.net/forum?id=RWioR2WoZNs&noteId=0tsV10knavE).  As for applications, even though the form of the ODEs in Section 3 admittedly seems difficult to analyze, our proofs for Section 4 utilize this to derive useful properties; in particular, Theorem 6 (the structure of the fluctuation) is actually the crucial step in Theorems 9 and 10 on the variance reduction effect of gradient descent. We think the finding from those theorems that the network is stabilized by gradient descent near convergence is interesting from the theoretical perspective and an initial step for further understanding. We admit, though, that applying the theory to change the practice is extremely difficult at the time.

---

> > ### Comment · Reviewer_EQhg · 2021-08-31
> > **Some final thoughts**
> >
> > Thank you for your reply. The experiments definitely improve the paper. I believe this is quite a solid work. I still feel, however, that it is a little too theoretical and complicated for this venue. It takes a serious effort to digest the results. A couple of suggestions that might help make the paper more accessible.
> >
> > * The "neuronal embedding" initialization seems very different from the standard initialization by independend Gaussians. There's some complex dependence between the weights hidden in the neuron indexing sets $\Omega_i,$ the distributions $P_i$, and the initialization maps $\omega_i^0$. It's not clear how to reasonably choose all these. It makes sense to discuss a little this choice, and the relation of this setting to the usual initialization.
> >
> > * There are very bulky and similarly-looking systems of equations describing first a finite network, then the MF limit, and then some kind of intermediate quantities obtained by combining finite networks with MF weights. It actually takes some effort to understand what exactly differs between these equations and what is the same. More detailed comments, drawing the reader's attention to the essential points, would help.
> >
> > * I don't feel that I have really understood some of the key points of the paper, in particular the concept "second-order MF limit" and its interpretation as "MF interactions at the fluctuation level". It would be nice to see more expanded and down-to-earth explanations. The exact meaning of "sampled neurons" is not clear: on page 4 it seems to just refer to the finite network, but in line 255 it is stated that the difference between the neural network and the sampled neurons is captured by $R$.
> >
> > * It would help to see some simplest examples illustrating the main fluctuation decomposition result (e.g., for a one-layer network, the quadratic loss, some simple $\Omega$ and $P$,...).

---

> > > ### Author Response · Authors · 2021-09-02
> > > **Re: Some final thoughts**
> > >
> > >
> > > Many thanks for the suggestions to improve the paper! We very much appreciate your efforts to read carefully the paper.
> > >
> > > Regarding the initialization, the work [Nguyen and Pham 2020] gives a construction of the neuronal embedding $(\Omega, \mathcal{F}, P, ( w_i^0 )_{i=1,…,L})$ for i.i.d. initializations (in which all weight entries are generated independently of each other), thus accommodating all such initializations. As pointed out in the same work and its companion note [Pham and Nguyen 2020], the framework is not limited to i.i.d. initializations; in particular, they show via the neuronal embedding framework that certain sophisticated non-i.i.d. initializations could be well beneficial.
> > >
> > > One thing to note is that these initializations follow the mean field scaling, and hence they are not exactly the same as the usual initialization in practice. In terms of infinite-width scaling, [Yang and Hu 2021] proves that infinite-width neural nets with standard scalings (and hence standard initializations) still behave NTK-like. In general, it is our belief that all infinite-width scalings which retain non-NTK feature learning are proxies of practical finite-width neural nets. This gives a strong motivation for considering the mean field limit, and studying the finite-width correction such as the fluctuation studied in our paper.
> > >
> > > In our revision, we will add some brief details for greater clarity, as the reviewer suggests. In particular:
> > >
> > > - In Section 2 (which gives an introductory background from the work [Nguyen and Pham 2020]), the multilayer network and the MF limit are two separate dynamical systems. Their connection is only formalized via the neuronal embedding and the sampling of neurons, described on page 4. Equation (2) is one main result from [Nguyen and Pham 2020] that states this connection in mathematical terms. The quantities in the equations are indeed corresponding to each other, and we will highlight these in the revision.
> > >
> > > - The sampled neurons, mentioned in Sections 2-3 and line 255, refer to samples $C_i(j_i)$. These samples induce from the MF limit quantities such as $w_i(t, C_{i-1}(j_{i-1}), C_i(j_i))$, as opposed to the neural net quantity $\mathbf{w_i}(t, j_{i-1}, j_i)$. Equation (2) reflects the approximation relation between these quantities.
> > >
> > > - The second-order MF limit is the process $R_i$ capturing the fluctuation $\mathbf{R}_i$ in the limit. Its interpretation as giving rise to the MF interaction at the fluctuation level can be seen from the derivation on page 8. In particular, this process can be seen as replacing the empirical average that involves neural net fluctuations $\mathbf{R}_i$ via a law-of-large-numbers averaging (Eq. (4)), hence MF interaction.
> > >
> > > - We hope to illustrate the two components in the fluctuation decomposition via the numerical illustration, in particular Figures (c) and (d) in our general comment (https://openreview.net/forum?id=RWioR2WoZNs&noteId=0tsV10knavE). Unfortunately we do not know of a solvable toy yet meaningful setup to illustrate the fluctuation decomposition. As such, we rely on the generality of the result to prove properties like Theorems 9 and 10 and use the numerical illustration for visual understanding.
> > >
> > >
> > > References:
> > >
> > > Phan-Minh Nguyen and Huy Tuan Pham, A rigorous framework for the mean field limit of multilayer neural networks, 2020.
> > >
> > > Huy Tuan Pham and Phan-Minh Nguyen, A note on the global convergence of multilayer neural networks in the mean field regime, 2020.
> > >
> > > Greg Yang and Edward, Hu Feature Learning in Infinite-Width Neural Networks, 2021.

---

> ### Author Response · Authors · 2021-08-27
> **Response to reviewer**
>
> In the case you have any questions after reading our responses, we would love to have an opportunity to address them, so feel free to let us know!

---

### Official Review · Reviewer_ErCU · 2021-07-20

**Rating:** 5
**Confidence:** 1

**Summary:**

This manuscript proposes a framework for analyzing multilayer neural nets based on the mean field theory, and prooven that gradient descent (GD) in a large-width multilayer network has a variance reduction effect.
This research is important and highly novel because mean-field analysis of "multilayer" neural networks is considered to be very difficult.


**Limitations And Societal Impact:**

It was noted around Line 74 in the manuscript.

**Main Review:**

I had a strong interest in the purpose of the manuscript, but unfortunately I did not understand it.
This is because the manuscript was confusing to me.
From my view point, I think there are the lack of explanation and numerous careless mistakes.
The following is a list of the relevant points.

Lack of explanation:
1.(Line 29) I cannot understand the word "deterministic limit" means what kind of limit, because there are no explanation.

2. (Line 94) What is the meaning of $ j_i $?

3. (Line 97) Is $ J_i \sim Uinf([N_i]) $ means $ J_i \sim N_i $ ?

4. (Line101) What is the meaning of index 2 in $ \partial_2 $?

5. (Line106) There might be no definition and explanation of $ \mathcal{F} $?

6. (Line 142) There seem to be some sentences which have no information, such as the following one.
The initialization assumption is also mild.

Careless mistakes:
1. In Reference, there are no author name for reference [12], [21], and [22].
I don't think this is a response for double blind either, since other papers written by the same author have the names listed.
Is it suggested that the paper is incomplete?

2. There is "Theorem 2" in Line 179. But there is not "Theorem 1" in the manuscript.
Is it also suggested that the paper is incomplete?

3. The words "GD" and "ODEs" are suddenly omitted without formal names.

4. There are some elementary errors in English grammar, such as the following
(Line 20) "helps reducing and" -> "helps to reduce, and"


In addition, since the theoretical results are not verified numerically, the validity of the derivation process cannot be verified by me.
In case other reviewers have not verified the validity of the derivation process, the authors should add numerical experiments.


For your information, here are my attributes.
I have no experience in theoretical analysis of DNNs using mean field theory.
Also, since I was educated in the Department of Physics, I am not familiar with mathematical paper structure.
Since NeuRIPS has participants with various attributes, I think that the influence of the manuscript will be greater if the authors take care to make it readable by people from a wide range of fields.
I am happy if I can understand your nice work.

**Time Spent Reviewing:**

12hours

---

> ### Author Response · Authors · 2021-08-10
> **Response to review**
>
> We thank the reviewer for having a strong interest in the work and for the careful reading! We have added some numerical illustration, the details of which are discussed in the general comment above (https://openreview.net/forum?id=RWioR2WoZNs&noteId=0tsV10knavE). We hope it aids the conveyance of the main results of the paper to a broader range of readers.
>
> We provide some clarifications:
>
> - Line 29, “deterministic limit” refers to an infinite-width limit that is non-random.
> - Line 94, $j_i$ is an index that goes from 1 to $N_i$; thanks for pointing this out.
> - Line 97, $Unif([N_i])$ refers to the uniform distribution over the set $[N_i]$ (the set of integers from $1$ to $N_i$).
> - Line 101, $\partial_2$ refers to the partial derivative w.r.t. the second variable.
> - Line 106, ${\cal F}$ refers to the sigma-algebra in the usual triplet $(\Omega, {\cal F}, P)$ defining a probability space. It is not crucial or used later in our paper.
> - Line 142, by “the initialization assumption is also mild”, we refer to the sub-Gaussian initialization assumption in Assumption 1, which should cover many initialization schemes of interest and hence is mild.
> - No author name for references [12, 21, 22]: the author names are the same as the previous references [11, 20]. This is an automatic feature of the LaTeX reference style we use.
> - There is no Theorem 1: We do not use numbering by types (so there is Definition 1 that comes before Theorem 2; likewise there is Theorem 10, but there are not 10 theorems).
> - We will explain the abbreviations GD (gradient descent) and ODEs (ordinary differential equations).
> - Typos: thanks, we will fix them.

---

> ### Author Response · Authors · 2021-08-27
> **Response to reviewer**
>
> In the case you have any questions after reading our responses, we would love to have an opportunity to address them, so feel free to let us know!

---

### Official Review · Reviewer_PZpb · 2021-07-28

**Rating:** 6
**Confidence:** 3

**Summary:**

The paper presents theoretical results on the fluctuations in the mean-field limit. Building on the framework proposed by Nguyen and Pham, the paper goes beyond the strict mean-field limit and derives next-order results about the fluctuations of weights and network output. Specifically, the paper shows that weight fluctuations scale as `1/\sqrt{N}` (for width `N`), and derives a differential equation that describes the fluctuation. It further shows that the network output obeys a CLT-like result, placing an upper bound on the error in terms of an inverse power of the width. Finally, the paper shows that gradient descent close to a global optimum tends to send the variance in the network output to zero.

UPDATE FOLLOWING AUTHOR RESPONSE:

I would like to thank the authors for sharing the results of numerical experiments verifying key results from the paper, and for addressing my and other reviewers' concerns. In light of these results I am updating my score and recommend accepting the paper.

**Ethical Concerns:**

This paper does not present any ethical concerns.

**Limitations And Societal Impact:**

Yes.

**Main Review:**

The paper makes an interesting and valuable contribution to the study of large width networks in the mean-field regime. Making the connection between infinite-width and finite-width networks is worthwhile, and the paper makes a step in this direction by studying the fluctuations around the large width limit in detail. The results are highly technical: They rely on explicit expressions for the weight fluctuations in the mean-field limit, which are expressed in terms of a complicated ODE. This level of complexity is to be expected from second-order analysis.

The exposition is as clear as can be given the significant technical complexity. The notation is consistent, there is a section devoted to reviewing the existing framework on which these results build, and heuristic explanations are provided in several instances which aid understanding. I do suggest numbering all equations and referring to these numbers in the text to help the reader navigate the text more easily. I also suggest referring to the main theorems in the introduction by number, to help guide the reader toward the main results when reading the later sections.

My main concern is that it is not possible to verify the results of the paper from the main text alone. There are no experimental results to back up the theoretical claims, and the proofs are not in the main text. The authors include useful heuristic arguments for several results, but given the technical complexity these are not enough to be convincing arguments. (See for example equation 3, a 4-line equation that describes the gradient descent evolution of the mean-field fluctuations.) I do believe these results are valuable. I will vote to accept the paper if the authors provide empirical evidence supporting Theorem 5 (including equation 3), and the results of Section 4.

**Time Spent Reviewing:**

2

---

> ### Author Response · Authors · 2021-08-10
> **Response to review**
>
> We thank the reviewer for finding our paper interesting and valuable! We acknowledge that the technicality of the paper is heavy but also unavoidable, so we agree some numerical illustration would be helpful. We will add numerical experiments, the details of which are discussed in the general comment above (https://openreview.net/forum?id=RWioR2WoZNs&noteId=0tsV10knavE). We will also make amendments to numbering and referencing.

---

> > ### Comment · Reviewer_PZpb · 2021-08-30
> >
> > Thank you for sharing these results. I updated my review accordingly and recommend accepting the paper.

---

> ### Author Response · Authors · 2021-08-27
> **Response to reviewer**
>
> In the case you have any questions after reading our responses, we would love to have an opportunity to address them, so feel free to let us know!

---

### Author Response · Authors · 2021-08-10
**General response to reviewers**

We thank the reviewers for their careful reading and very constructive comments, especially when the paper is unavoidably heavily technical. Below we address a common comment that is on experimental verification. While we generally avoid unhelpful experiments, we agree that some numerical simulations could be useful for verification as well to quickly aid understanding. We provide some results and codes in the following link: https://anonymous.4open.science/r/nn-clt-FF2B/

In particular, this is a simple experiment with MNIST, classifying digits 0 and 4 versus 5 and 9 with 3-layer fully-connected networks of varying width $N$, equipped with Huber loss and tanh activation. We use only 100 images for training. The small scale of this experiment is necessary for us to run full-batch gradient descent in reasonable time, since the paper focuses on the mean-field fluctuation and avoids the effect of SGD (discussed in our Limitations section).

Following the figure in this link: https://anonymous.4open.science/r/nn-clt-FF2B/figure.png

- Fig (a): This is the evolution of the training loss, for $N=50, 100, 200, 400$. Note that the global optimum is found at around iteration $10^4$, and the trajectory indicates nonlinear dynamics.
- Fig (b): This is the histogram of the output fluctuation $\sqrt{N}(\hat{\bf y}(t,x_{test}) - \hat{y}(t,x_{test}))$ at a particular test sample $x_{test}$ and iteration $10^3$. The agreement among various $N$ verifies the existence of a limiting Gaussian-like behavior, predicted by Theorems 5 and 6.
- Fig (c): This is the plot of the output variance ${\bf E}\mathbb{E}_Z[|\sqrt{N}(\hat{\bf y}(t,X) - \hat{y}(t,X))|^2]$ versus training iteration. The variance decreases quickly past iteration $10^4$. This is predicted by Theorem 10.
- Fig (d): This is the plot of the Gaussian component $\tilde{G}^y$ in the output variance. Note that after iteration $10^4$, this component no longer moves since the global optimum is reached. By contrasting this plot with Fig (c), we see clearer the role of gradient descent training in reducing the variance.
- Fig (e): This is similar to Fig (c), except that we initialize at a global optimum. The variance monotonically decreases on the entire period. This is predicted by Theorem 9.

All simulations are run with a single NVIDIA Tesla P100 GPU. We will provide a full description of the experiment in the updated manuscript, and please feel free to ask for any further detail (e.g. how estimation is done to obtain the plots, etc). The experiment can be repeated by running the provided codes.

---

### Decision · Program_Chairs · 2021-09-28

**Decision:**

Accept (Poster)

**Comment:**

The paper presents theoretical results on neural network training for finite width networks, which extends prior work which assumes an infinite width limit. The theory proves (among other results) bounds on the output variance of standard feed forward networks as a function of network width when trained with GD. Reviewers agreed that the technical contribution is significant enough to warrant publication, however no reviewer had time to go over the 50 pages of proofs provided in the Appendix. Unfortunately, the ML conference system is ill equipped to handle these kinds of submissions (a reviewer for a mathematics journal may have over a year to review a highly technical submission).
During the rebuttal period, authors provided experimental confirmation of the theoretical predictions for a simple network trained on MNIST which was enough to satisfy reviewers. Although I am still hesitant to accept an almost purely theoretical work where no reviewer had time to go over the proofs, I will side with the reviewers in light of the experimental confirmation provided during the rebuttal period and recommend acceptance.


**Consistency Experiment:**

NeurIPS has a long history of experimentation. In 2014, NeurIPS ran an experiment in which 10% of submissions were reviewed by two independent committees to quantify the randomness in the review process. This year, we repeated a variant of this experiment to see how the quality of the review process has changed over time.  This paper was part of the experiment and was therefore assigned to two committees (consisting of reviewers, an Area Chair, and a Senior Area Chair) that reached independent decisions.  If both committees made the same recommendation, this recommendation was followed. If a single committee recommended acceptance, the paper was accepted (with the exception of a few cases in which the other committee identified what we considered a fatal flaw, e.g., an error in a key result).

Both committees reached the same decision: **Accept (Poster)**

The other committee assigned to the paper recommended **Accept (Poster)**.  You can find the other set of reviews, along with any follow up discussion with the authors here:
https://openreview.net/forum?id=jg9LM8QItms